# Implementing an empirical scalar constitutive relation for ice with flow-induced polycrystalline anisotropy into large-scale ice sheet models

Felicity S. Graham[1], Mathieu Morlighem[2], Roland C. Warner[3], and Adam Treverrow[3]

[1]Institute for Marine and Antarctic Studies, University of Tasmania, Private Bag 129, Hobart, Tasmania 7001, Australia
[2]Department of Earth System Science, University of California, Irvine, California, USA
[3]Antarctic Climate and Ecosystems Cooperative Research Centre, Private Bag 80, Hobart, Tasmania 7001, Australia

*Correspondence to:* Felicity S. Graham (felicity.graham@utas.edu.au)

**Abstract.** The microstructure of polycrystalline ice evolves under prolonged deformation, leading to anisotropic patterns of crystal orientations. The response of this material to applied stresses is not adequately described by the ice flow relation most commonly used in large-scale ice sheet models – the Glen flow relation. We present a preliminary assessment of the implementation in the Ice Sheet System Model (ISSM) of a computationally efficient, empirical, scalar, constitutive relation which addresses the influence of the dynamically steady-state flow-compatible induced anisotropic crystal orientation patterns that develop when ice is subjected to the same stress regime for a prolonged period – sometimes termed tertiary flow. We call this the ESTAR flow relation. The effect on ice flow dynamics is investigated by comparing idealised simulations using ESTAR and Glen flow relations, where we include in the latter an overall flow enhancement factor. For an idealised embayed ice shelf, the Glen flow relation overestimates velocities by up to 17% when using an enhancement factor equivalent to the maximum value prescribed in the ESTAR relation. Importantly, no single Glen enhancement factor can accurately capture the spatial variations in flow across the ice shelf generated by the ESTAR flow relation. For flow line studies of idealised grounded flow over varying topography or variable basal friction – both scenarios dominated at depth by bed-parallel shear – the differences between simulated velocities using ESTAR and Glen flow relations depend on the value of the enhancement factor used to calibrate the Glen flow relation. These results demonstrate the importance of describing the deformation of anisotropic ice in a physically realistic manner, and have implications for simulations of ice sheet evolution used to reconstruct paleo-ice sheet extent and predict future ice sheet contributions to sea level.

## 1 Introduction

An essential component of any ice sheet model is the constitutive relation (or flow relation), which connects ice deformation rates and applied stresses. Under prolonged deformation polycrystalline ice aggregates develop material anisotropy, patterns of preferred orientations of individual crystal $c$-axes, which we refer to as anisotropic fabrics. There is broad agreement that deformation under stresses within polar ice sheets leads to widespread development of anisotropic fabrics (e.g., Budd and Jacka, 1989; Hudleston, 2015), through a variety of physical processes (e.g., Faria et al., 2014). The development of these anisotropic

fabrics is associated with different deformation rates for different patterns of applied stresses. Laboratory deformation tests (e.g. Russell-Head and Budd, 1979; Bouchez and Duval, 1982; Jacka and Maccagnan, 1984; Treverrow et al., 2012) and field evidence (e.g., Wang and Warner, 1999; Wang et al., 2002a; Treverrow et al., 2015) indicate that the influence of anisotropy on deformation rates is significant for polar ice sheets and should be incorporated in flow relations used in large-scale ice sheet

models.

There are complex flow relations that explicitly include material anisotropy, and models that track the evolution of crystal fabrics, as discussed briefly below, but the Glen flow relation (Glen, 1952, 1953, 1955, 1958; Nye, 1953) is the prevailing description of ice deformation in large-scale ice sheet models. It is given by the following expression

$$\dot{\boldsymbol{\varepsilon}} = A(T')\tau_e^{n-1}\boldsymbol{\sigma}'. \tag{1}$$

Here, $\dot{\boldsymbol{\varepsilon}}$ is the strain rate tensor (s$^{-1}$), $\tau_e$ is the effective stress (Pa), $\boldsymbol{\sigma}'$ is the deviatoric stress tensor (Pa), and $n$ is a power law stress exponent (observations support a value of $n = 3$). $A(T')$ is a flow parameter (Pa$^{-n}$ s$^{-1}$), for which various parameterisations exist based on laboratory tests and field measurements (e.g., Budd and Jacka, 1989; Cuffey and Paterson, 2010, p. 73). The Glen flow relation is not expected to hold for anisotropic ice (Budd et al., 2013), being empirically derived under the assumption of mechanical isotropy (Nye, 1953), which necessarily restricts the possible structure of the flow relation (Glen,

1958). Hence, while the Glen flow relation captures the observed nonlinear response of ice deformation to the magnitude of the applied stresses, it cannot explain the observed dependence of steady-state strain rates on the character of the applied stress.

To account for the increased deformability associated with steady-state creep, a common adaptation of the Glen flow relation is the inclusion of a constant flow enhancement factor, $E_G$,

$$\dot{\boldsymbol{\varepsilon}} = E_G A(T')\tau_e^{n-1}\boldsymbol{\sigma}'. \tag{2}$$

Such a parameter is included in most large-scale ice sheet models (e.g., Saito and Abe-Ouchi, 2004; Greve, 2005; Huybrechts et al., 2007; Winkelmann et al., 2011), typically to increase the rates expected in the bed-parallel shear that is important in ice sheets. However, the specification of $E_G$ is typically ad hoc: $E_G$ may be selected from reported experimental values (e.g., Duval, 1981; Jacka and Maccagnan, 1984; Pimienta et al., 1987; Treverrow et al., 2012), or used as a model tuning parameter. Indeed, Greve and Blatter (2009) comment that $E_G$ is "often introduced without explicitly mentioning anisotropy". In any

case, a value of $E_G$ that does not vary spatially in connection with the fabric and flow configuration, will lead to an unrealistic spatial distribution of strain rates (Wang and Warner, 1999; Wang et al., 2002a; Treverrow et al., 2015). Previous studies have assigned regional values to $E_G$ (e.g., Ma et al., 2010) that may also vary according to the prevailing stress regime.

Budd et al. (2013) recently proposed a flow relation based on results from laboratory ice deformation experiments involving simple shear, compression, and combinations of these. These experiments reached steady-state creep rates – also referred

to as tertiary flow. As these strain rates and corresponding compatible fabrics were found to vary depending on the relative proportions of the simple shear and compression stresses, Budd et al. (2013) defined an enhancement factor $E$ as a function of the stress configuration. The laboratory experiments were satisfactorily described by a scalar relation between stresses and strain-rates (Budd et al. (2013)), which motivated the suggestion that this might extend to general stresses. We refer to the

generalised flow relation proposed by Budd et al. (2013) as ESTAR (Empirical Scalar Tertiary Anisotropy Regime), since it is based on steady-state (tertiary) creep rates describing the deformation of ice with a flow-compatible induced anisotropy and features a scalar (collinear) relationship between the strain rate and deviatoric stress tensor components. As discussed below, the ESTAR relation is a mathematically isotropic flow relation for ice with a fully developed anisotropic fabric compatible with the deformation regime. While this flow relation will not capture the all the influences of the full variety of anisotropic crystal fabrics, nor situations where the fabric and stress regime are not compatible, we suggest it should provide an improvement on the Glen flow relation.

Here, as a first step towards exploring the implications of this description of anisotropic ice in large-scale ice sheet models, we describe how to implement the ESTAR flow relation and apply the required changes to the Ice Sheet System Model (ISSM; Larour et al., 2012). ISSM is a thermomechanical finite element model that solves the full system of Stokes equations to describe ice flow. This will permit exploration of the ramifications of the ESTAR flow relation in general ice flow situations. In this initial study, we examine the effect of induced anisotropy in simple, idealised scenarios of a floating ice shelf and of grounded ice sheets, comparing flow fields simulated using ESTAR and Glen flow relations. Section 2 discusses the role of anisotropic ice in polar ice sheets. Section 3 presents a brief overview of flow relations for ice with a polycrystalline anisotropy, focussing on the experimental and theoretical basis and applicability of the ESTAR flow relation. Section 4 details the implementation of the ESTAR flow relation in ISSM while Sect. 5 verifies the implementation against an analytical solution. In Sect. 6, we compare simulations of ice flow with the ESTAR and Glen flow relations using a suite of idealised flow geometries, including prognostic (evolving) simulations based on selected experiments from the Ice Sheet Model Intercomparison Project for Higher Order Models (ISMIP-HOM; Pattyn et al., 2007). Section 7 discusses the results, and implications of the ESTAR description of the flow of ice with a flow-induced crystallographic anisotropy. Conclusions are drawn together in Sect. 8.

## 2 Anisotropy and polar ice sheets

In this section we outline the development of anisotropic fabrics in polycrystalline ice, including the tertiary flow regime and its connection with enhanced deformation rates and development of compatible anisotropy, and discuss the expected occurrence of anisotropy and tertiary flow conditions in polar ice sheets.

Individual ice crystals have a strong mechanical anisotropy, owing to high levels of deformability due to slip on the crystallographic basal plane, whose normal is the crystallographic $c$-axis. Under prolonged deformation, the microstructure of ice evolves, leading to the development of patterns of preferred $c$-axis orientations (crystal orientation fabric). While the direct evidence for anisotropic fabrics in polar ice sheets is limited to data from a small number of deep ice core sites, the long passage of ice through the ice sheet stress regime makes this inevitable. In the course of flow through a polar ice sheet each parcel of polycrystalline ice is deformed by exposure to patterns of stress which usually change gradually, the most obvious being stresses dominated by vertical compression through the upper part of the ice sheet before a smooth transition to predominantly bed-parallel simple shear below.

As discussed by Budd and Jacka (1989) the nature of the applied stresses and the rotation with the flow produce anisotropic crystal fabrics that evolve to reflect the accumulating strain history and flow. These fabrics necessarily have a compatibility with their strain history, and if we assume that fabrics usually develop within time frames that are short compared to the rates at which the ice encounters changing stress regimes (Thorsteinsson et al., 2003) they will typically be compatible with recent strain history and by extension with the current stress regime.

There are exceptional locations where this concept of compatibility is likely to break down – where the stress regime experienced by the flowing ice alters rapidly. Examples include transitions from tributary glacier or sheet flow into the shear margins of ice streams or ice shelves (e.g. Thorsteinsson et al., 2003), and for deeper layers the transition from bed-parallel shear to extensional flow at ice shelf grounding lines or at the onset of ice streams. Thorsteinsson et al. (2003) also point out that temporal changes in the stress regime, such as divide migration, can provide a more abrupt change than advection of ice through a steady distribution of stresses.

In their review, Budd and Jacka (1989) made a further conjecture about the character of the anisotropic fabrics. Comparing evidence from an array of boreholes on Law Dome, East Antarctica (the most extensive ice coring program focussed on ice dynamics rather than paleoclimate) with laboratory studies of ice deformation, they suggested that as ice passes through the varying ice sheet stress regime, it likewise passes through a succession of "steady-state" fabrics, which they termed tertiary flow. We return to this point after a brief review of the stages of deformation observed in the accommodation to a fixed stress from a laboratory perspective.

Experimental observations for pure polycrystalline ice, demonstrate that an accumulated strain of $\geq 10\%$ is required for the microstructure to evolve to a state that is compatible with the flow configuration, irrespective of its initial condition (Jacka and Maccagnan, 1984; Gao and Jacka, 1987; Li and Jacka, 1998; Treverrow et al., 2012). Specifically, laboratory experiments have demonstrated that under conditions of constant stress and temperature, deformation of polycrystalline ice with an initial statistically random distribution of crystallographic $c$-axes (isotropic fabric) passes though three stages. Initially it behaves as a mechanically isotropic material (Budd and Jacka, 1989) where the rate of deformation is not sensitive to the character of the applied stresses. At the commencement of deformation, during the primary stage of creep, the initially high strain rate rapidly decreases. A minimum strain rate is reached during secondary creep (Fig. 1). With continued strain, a tertiary stage of creep is established (typically observed at strain of $\sim 10\%$ in under laboratory conditions) with steady strain rates that are enhanced relative to the rate observed during secondary creep, and characterised by the development of statistically steady-state anisotropic microstructures that are associated with the stress regime. We describe this as tertiary anisotropy. An important feature of the experiments is that for the same stress magnitude the enhanced tertiary deformation rates under compression alone or simple shear alone are different (Budd and Jacka, 1989; Treverrow et al., 2012). Laboratory experiments also indicate that attainment of enhanced deformation rates precedes the full development of anisotropic fabrics (Jacka and Maccagnan, 1984), suggesting that strain rate does not alter much once the most obstructive grains have been removed by fabric development.

There are a variety of microdeformation and recovery processes that lead to the development of anisotropic fabrics (e.g. Faria et al., 2014); however, there is not a consensus on how the activity of specific processes may vary according to temperature and/or stress. Observations from ice cores can provide guidance on the temperature domain over which laboratory observations

remain indicative of in-situ behaviour. For the A001 ice core drilled at the summit of Law Dome, East Antarctica, a distinct small circle girdle (cone-type) fabric (the compression-compatible form) is observed at a depth of 318 m (Fig. 3a in Budd and Jacka, 1989), where the total accumulated strain is $\sim 30\%$, the temperature is $\sim -22°C$, and the in-situ flow regime is compression dominated with approximately radial symmetry in the transverse rates. Since the fabric is already well developed at this point, the actual strain required to achieve tertiary creep was probably less. Similar compression-compatible fabrics have been observed in the laboratory at higher temperatures and stresses (e.g. Jacka and Maccagnan, 1984; Treverrow et al., 2012). Accordingly, these observations suggest a conservative temperature limit of $\sim -22°C$ for extrapolating laboratory observations of fully developed tertiary creep down to in-situ conditions.

Information about the effects of anisotropy on in-situ deformation rates in polar ice sheets is limited. Analyses of the shear strain rate profiles inferred from bore hole inclination measurements on Law Dome (Russell-Head and Budd, 1979; Wang and Warner, 1999; Wang et al., 2002a; Treverrow et al., 2015) indicate enhancement in deformation rates correlated with the stress regime consistent with tertiary flow, and fabrics that match the expectations of the laboratory experiments (Donoghue and Jacka, 2009; Treverrow et al., 2016).

While the relevant temperature regime and the amount of strain that needs to be accumulated remains uncertain, tertiary creep, with the associated development of polycrystalline anisotropy, may be common in polar ice sheets, particularly in regions controlling the large scale dynamics, as discussed further in Sect. 3.4. This has the potential to provide a relatively simple description of the deformation properties of this anisotropic ice, since the stress regime becomes a guide to the enhanced flow, and motivates this study to incorporate an empirical tertiary flow relation into large scale ice sheet modelling, as discussed in the next section.

## 3 Constitutive relations for anisotropic polycrystalline ice

A range of constitutive relations have been proposed to account for polycrystalline anisotropy. They can be broadly grouped in two categories (Marshall, 2005): (1) those defined at the individual ice crystal scale, where the effects of crystallographic anisotropy are parameterised based on specific properties of individual crystals, and (2) those connected with the present work that describe the deformation of ice with flow-compatible anisotropy, through an empirical function of the stress configuration. In this section, we briefly review these two approaches, and the underlying experimental and modelling basis for the ESTAR flow relation. We then outline the expected domain where the ESTAR flow relation might apply in polar ice sheets. Lastly, we distinguish between anisotropic constitutive relations, and constitutive relations for ice with a compatible, flow-induced anisotropy. In what follows, we distinguish between the Glen enhancement factor $E_G$ and the ESTAR enhancement factor $E(\lambda_S)$, which is a function of normal deviatoric and simple shear stresses, parameterised by the shear fraction $\lambda_S$, which characterises the contribution of simple shear to the effective stress as discussed later. Where necessary, we denote a more general enhancement factor, i.e., with unspecified form, as $E$.

## 3.1 Microstructure approaches

Experiments on single crystals of ice demonstrate that deformation occurs predominantly by slip on the crystallographic basal plane (perpendicular to the $c$-axis), with the yield stress being geometrically related to the magnitude of the applied stress resolved onto the basal plane (Trickett et al., 2000) according to Schmid's Law (Schmid and Boas, 1950). In the formulation of the Glen flow relation for polycrystalline ice the assumed isotropic distribution of $c$-axes results in indifference to material rotations (relative to applied stresses) and an isotropic expression. The underlying anisotropic deformation properties of individual crystals, in conjunction with the development of crystallographic preferred orientations during deformation of polycrystalline ice to high strains (e.g. Russell-Head and Budd, 1979; Jacka and Maccagnan, 1984; Pimienta et al., 1987; Morgan et al., 1998; DiPrinzio et al., 2005; Durand et al., 2009; Budd et al., 2013; Montagnat et al., 2014), has driven the development of constitutive relations in which the connection between deviatoric stresses and resulting strain-rates is regarded as an intrinsic material property determined by the effects of microstructure on bulk deformation processes, (e.g. Lile, 1978; Lliboutry, 1993; Azuma and Goto-Azuma, 1996; Staroszczyk and Gagliardini, 1999; Thorsteinsson, 2001; Gödert, 2003; Gillet-Chaulet et al., 2005; Pettit et al., 2007; Placidi et al., 2010). See also the review by Gagliardini et al. (2009). These constitutive relations describe polycrystalline anisotropy through the geometric relationship between the crystallographic $c$-axes and the stresses driving deformation, with the role of misorientation relationships between nearest neighbour grains explicitly considered in some cases.

The complexity of resulting flow relations varies according to the extent to which a physically realistic description of microdeformation and recovery processes, or a parameterisation of these, enters into the relationship between strain rates and the stresses driving deformation. Many of these constitutive models are more complicated than a collinear flow relation and involve a tensor coupling in place of Eq. (2). A further consideration is the quantitative description of fabric that is used in flow relations. Those incorporating a discrete vector-based description of fabric based on $c$-axes (e.g. Lile, 1978; van der Veen and Whillans, 1994; Azuma and Goto-Azuma, 1996; Thorsteinsson, 2002) are appropriate only for highly localised studies and are incompatible with large-scale ice sheet modelling. Flow relations based on a continuous description of fabric, e.g., a parameterised orientation distribution function (ODF) or $c$-axis orientation tensor are also possible (e.g. Staroszczyk and Gagliardini, 1999; Gödert, 2003; Gillet-Chaulet et al., 2005; Pettit et al., 2007; Placidi et al., 2010). A central tenet of these microstructure-based constitutive relations is that they require the specific input of the anisotropic character of the material being deformed. They describe the instantaneous deformational response of any sample of ice to any pattern of applied stresses. In this regard they can be defined as anisotropic flow relations, whether they involve a tensor or a scalar connection between stresses and strain-rates (Faria, 2008).

Including any fabric-based description of material anisotropy in the flow relation for an ice sheet model requires either a prescription of anisotropy or an additional set of equations governing the fabric evolution. A complication of such an approach is the computational overhead and uncertainty associated with defining the spatial distribution of fabric within ice sheets, which is poorly constrained by observations. Sometimes, as a simplification, restricted forms of the ODF or orientation tensor are specified, which may not adequately describe all fabrics likely to be encountered in an ice sheet. To date, flow relations

utilising a fabric description that relies on fabric evolution equations or that is imposed as a function of location within the ice sheet have been restricted to regional simulations (e.g., Seddik et al., 2011; Martín and Gudmundsson, 2012; Zwinger et al., 2014).

## 3.2 Empirical approaches to tertiary flow

5 As indicated in Sect. 2 a flow relation applicable to the tertiary regime may provide a useful description of deformation, capturing important aspects of the flow of anisotropic ice in polar ice sheets.

An empirical approach to the deformation properties of ice with a tertiary polycrystalline anisotropy has developed, comprising experimental and observational studies (Li et al., 1996; Wang et al., 2002a, b), modelling (Wang and Warner, 1998, 1999; Hulbe et al., 2003; Wang et al., 2003, 2004; Breuer et al., 2006; Wang et al., 2012), and theoretical studies (Warner 10 et al., 1999). This empirical approach has focussed on the development and assessment of a flow relation for polycrystalline ice in the circumstance where the crystal fabric and the flow properties are both determined by the stress regime; specifically, the relative proportions of the simple shear and normal deviatoric stresses. For such flow relations, it is typically assumed that the spatial variation in dynamic conditions (e.g., flow configuration and temperature) occur only gradually in an ice sheet, so that the microstructure evolves to maintain compatibility with these conditions. Through most of an ice sheet we expect that 15 the rate of microstructural evolution generally exceeds the rate at which the flow configuration varies, and that the distances travelled by a parcel of ice during the time taken to develop a compatible fabric are typically small compared to the relevant ice sheet spatial scales.

A flow line model by Wang and Warner (1999) implemented an empirical enhancement function based on the stress regime, using a compression fraction, $\lambda_C = \sqrt{1 - \lambda_S^2}$, and an earlier parameterisation of tertiary enhancement, $E(\lambda_C)$, from Li et al. 20 (1996). That study of a flow line on Law Dome, East Antarctica, showed how an enhancement function improved agreement with observations of shear strain rate profiles from borehole inclination measurements, and displayed correlations with ice core crystal fabrics. Wang et al. (2002a) demonstrated that vertical variation of enhancement was required to match the shear strain rate profile from the Law Dome Summit South (DSS) borehole, and showed its correlation to stress configuration, and the connection with crystal fabric anisotropy.

25 The generalised tertiary flow relation proposed by Budd et al. (2013) represents a continuation of this strand. While more complicated parameterisations were also explored, Budd et al. (2013) found that a scalar flow relation, i.e., one maintaining the collinear relationship between the components of $\dot{\varepsilon}$ and $\sigma'$, with a functional dependence on both the second invariant of $\sigma'$ and the fraction of the deformation that was simple shear, provided a good fit to laboratory data for tertiary flow in combined compression and shear experiments. Given this scalar character, they proposed what we term the ESTAR flow relation as a 30 suitable candidate flow relation for arbitrary stress configurations, extrapolating from its applicability to the limited set of experimental stress configurations described in Li et al. (1996) and Budd et al. (2013). A scalar relation also simplifies the requirements for implementation within ice sheet models that currently use the Glen flow relation. A simplified version of the ESTAR flow relation has also been incorporated into the ice sheet model SICOPOLIS (SImulation COde for POlythermal Ice Sheets, http://www.sicopolis.net; Greve and Blatter, 2009, 2016).

### 3.3 Empirical Scalar Tertiary Anisotropy Regime (ESTAR) flow relation

Here, we summarise the generalised constitutive relation for ice in tertiary flow (the ESTAR flow relation) proposed by Budd et al. (2013). We implement this in ISSM as an alternative to the Glen flow relation as the ESTAR flow relation is more applicable to the tertiary creep of anisotropic polycrystalline ice typical in ice sheets. The ESTAR flow relation is a scalar power law formulation based on tertiary creep rates measured in laboratory ice deformation experiments under various combinations of simple shear and compression that has been generalised to arbitrary stress configurations.

The main features of Budd et al. (2013) and the ESTAR flow relation are the observation that tertiary strain rates depend on the nature of the applied stresses, and the identification of the proportion of the overall deformation stress that can be regarded as simple shear as the appropriate variable to characterise that dependency. Accordingly, determining the shear fraction, $\lambda_S$, is the main task in implementing the ESTAR flow relation. This involves the identification of a particular local plane – the local non-rotating shear plane – and the determination of the shear acting on that plane, $\tau'$, as the measure of simple shear. As indicated in Sect. 4 below Budd et al. (2013) also prescribed a further projection to remove any component of $\tau'$ parallel to the deformational vorticity. The importance of moving beyond strain rates to consider other aspects of flow – the 'movement picture' – has been recognised since at least the 1970s (e.g., Budd, 1972; Kamb, 1973; Duval, 1981; Budd et al., 2013). Duval (1981) identified the plane normal to the velocity gradient in a simple shear regime as the 'permanent shear plane' and discussed its role in the evolution of crystal fabrics. Budd et al. (2013) proposed a local definition for this plane in an arbitrary flow as the plane containing the velocity vector and the vorticity vector associated solely with deformation.

Recasting Eqs. (62) and (63) of Budd et al. (2013) to more closely resemble Eq. (2), the ESTAR flow relation is given by the following expression:

$$\dot{\varepsilon} = E(\lambda_S)A(T')\tau_e^2\sigma'. \tag{3}$$

Assuming $n = 3$ in Eq. (2), Eq. (3) differs from the Glen flow relation only by the form of the functional enhancement factor $E(\lambda_S)$, which could be regarded as providing a variable enhancement function for the Glen relation that incorporates the effect of flow-induced fabric anisotropy. $E(\lambda_S)$ in Eq. (3) is defined as

$$E(\lambda_S) = E_C + (E_S - E_C)\lambda_S^2. \tag{4}$$

Here, $E_C$ and $E_S$ are the enhancement factors above the minimum or secondary deformation rate of isotropic ice under compression alone or simple shear alone, respectively, and $\lambda_S$ is the shear fraction, which characterises the contribution of simple shear to the effective stress. The shear fraction $\lambda_S$ can be written as

$$\lambda_S = \frac{\|\tau'\|}{\tau_e}. \tag{5}$$

The collinear nature of the ESTAR flow relation Eq. (3) allows $\lambda_S$ to be written in terms of the corresponding strain rates, which is more convenient for Stokes flow modelling, as

$$\lambda_S = \frac{\dot{\varepsilon}'}{\dot{\varepsilon}_e}, \tag{6}$$

where $\dot{\varepsilon}'$ is the magnitude of the shear strain rate on the locally non-rotating shear plane, as defined in Eq. (7) below. In compression-alone scenarios, including three-dimensional uniaxial compression and two-dimensional plane compression and extension, $\lambda_S = 0$, so that $E(\lambda_S) = E_C$. Similarly, for simple shear alone, $\lambda_S = 1$ and $E(\lambda_S) = E_S$.

Analysis of tertiary creep rates for experiments conducted in simple shear-alone and compression-alone suggests that a suitable ratio of $E_S$ to $E_C$ for ice sheets is $\sim 8/3$ (Treverrow et al., 2012). The same study also suggests that $E \propto \sqrt{\tau_e}$ for tertiary creep rates determined over a range of stress magnitudes. A flow relation incorporating such a stress dependent enhancement could be achieved by employing a creep power-law stress exponent of $n = 3.5$, rather than the more commonly used $n = 3$, assuming both $E_S$ and $E_C$ are described by functions of $\sqrt{\tau_e}$. For simplicity, we have excluded the apparent stress dependence of $E_S$ and $E_C$ in our initial implementation of the ESTAR flow relation in ISSM since further work is required to verify the stress dependence of $E_S$ and $E_C$ experimentally for complex, combined stress configurations. Accordingly, we use scalar enhancement factors of $E_S = 8$ and $E_C = 3$ for the idealised scenarios examined in this study. These values may be at the higher end of the anticipated range in $E_S$ and $E_C$ for an ice sheet (see e.g., Russell-Head and Budd, 1979). However, the strength of anisotropy and its influence on ice dynamics in comparison to the enhanced Glen flow relation depends on the ratio $E_S/E_C$ and its spatial variation, i.e., the dynamically controlled distribution of $E(\lambda_S)$.

If the enhancement parameters are selected so that $E_C = E_S = E_G$, where $E_G$ is the Glen enhancement factor, the ESTAR flow relation loses its dependence on the stress regime, reducing to the Glen flow relation since $E(\lambda_S) \equiv E_G$. However, the viscous creep behaviour of polycrystalline ice is highly anisotropic and regional variations in the relative proportions of shear and normal strain rates, which are driven by variations in the distribution of the stresses responsible for deformation, mean that spatial contrasts in anisotropy are common and widespread in ice sheets. For this reason, a spatially varying enhancement factor is required for ice sheet modelling (e.g., Morgan et al., 1998; Wang and Warner, 1999; Wang et al., 2002a).

Comparisons of simulations of ice sheet dynamics using the ESTAR and Glen flow relations will be influenced by: the choice of the Glen enhancement parameter, $E_G$; the ESTAR parameters $E_C$ and $E_S$; and the spatial distribution of $\lambda_S$. The most significant differences between simulations using Glen and ESTAR flow relations are expected to arise where there are regional contrasts in $\lambda_S$. Specific regions where these conditions are likely to arise include: the progression with increasing depth in the ice sheet from a regime of normal stresses to one dominated by bed parallel shear; the contrasts between lateral margins of embayed ice shelves and ice streams and their central flows; and where there is significant relief in the bedrock topography.

A caveat is that as stated earlier, for the ESTAR flow relation to hold, the assumption of the tertiary state (i.e., crystallography and deformation rates being compatible with the instantaneous stress/deformation regime) requires that this does not change too rapidly along the flow. That is to say, for a compatible (tertiary) anisotropy to be present, the present deformation regime needs to be a suitable indicator of the recent strain history of the flowing ice.

### 3.4 Domain of applicability of tertiary creep and the ESTAR flow relation

As discussed in Sect. 2 two conditions need to be satisfied for the applicability of the tertiary creep concept and the ESTAR flow relation – activation of the appropriate microstructural processes to generate steady-state fabrics, and sufficiently gradual

changes in the stress regime experienced by the flowing ice to permit a quasi-steady transition in the fabric and corresponding deformation rate as controlled by the shear fraction $\lambda_S$.

Within an ice sheet there will be zones where the assumption of compatible tertiary flow will not apply; however, these zones will be restricted in extent (Thorsteinsson et al., 2003). We contend that the ESTAR flow relation will apply to the majority of the dynamically active regions of an ice sheet, in particular those zones where creep deformation makes a significant contribution to the overall flow. Specific zones where the assumption of tertiary creep may be inappropriate can be summarised as those where the fabric has not yet evolved compatibility with the flow, where there is a rapid transition in the flow configuration, or where creep deformation makes only a minor contribution to the overall dynamics.

Regions where rapid transitions in dynamic conditions can lead to abrupt changes in the pattern of applied stresses and a potential breakdown in tertiary flow compatibility include ice shelf grounding zones and other locations where basal traction is lost or abruptly changes, e.g., where ice flows over a subglacial lake, or with the onset of basal sliding in ice streams. The convergence zones where tributary glaciers or ice streams merge with a larger flow unit at a high angle (Thorsteinsson et al., 2003) may also lead to a transition in dynamic conditions that is problematic for the assumption of tertiary compatibility. Of course the more highly dynamic the evolving flow regime, the more rapidly a new compatible anisotropy will be established, so that the spatial interval where the flow relations are inapplicable may be limited. The effect of these localised encounters of stresses with incompatible fabrics on the overall flow is unclear; however, we note that similar difficulties exist for a Glen-type flow relation, which unlike the ESTAR flow relation does not have the benefit of being able to correctly describe enhanced flow rates throughout the remainder of the ice sheet.

Under the very cold and low stress conditions occurring in the uppermost layers of the polar ice sheets, particularly towards the interior at high elevations, any increase in the accumulated strain necessary to develop a compatible fabric may lead to a near-surface zone in which the assumption of tertiary creep is not valid. Since the development of anisotropic fabrics provides an indication of the existence of, or the approach towards tertiary flow, the observation of evolving anisotropic fabrics at modest depths, (e.g., $\sim 100-200$ m; DiPrinzio et al., 2005; Montagnat et al., 2014; Treverrow et al., 2016) allows the maximum extent of the zone where tertiary creep is not occurring to be estimated. Because the nonlinear nature of polycrystalline ice deformation rates leads to very high viscosities in low temperature and stress environments, incorrectly estimating deformation rates due to the assumption of tertiary flow in such regions may be of limited importance to simulations of ice sheet evolution.

## 3.5 The semantics of anisotropy

We conclude this section with some remarks about the seeming paradox of using an isotropic constitutive relation to describe the deformation of ice that has an anisotropic pattern of $c$-axis orientations.

Anisotropy in broad terms describes differences in physical systems associated with different directions. The various flow relations in Budd et al. (2013) involve a specific direction, namely the normal to the non-rotating shear plane that is determined by the combination of the stress regime and the flow, and also connect the strain rates with the character of the stress regime through the shear fraction $\lambda_S$.

In materials science, anisotropy is used to refer to material properties which have different values when measured along different directions. Indeed, the term is often introduced (e.g., Kocks et al., 1998) as the opposite to isotropy or indifference to rotations. Polycrystalline ice with a $c$-axis distribution that exhibits certain preferred directions clearly displays anisotropy, though the manifestation of this physically discernible feature in deformational properties requires demonstration.

Microstructural approaches to ice deformation, such as those discussed in Sect. 3.1 above, typically aim to describe the prompt response (ignoring transient primary creep) of ice with any crystal fabric to an arbitrary arrangement of applied stresses, where the emphasis on promptness covers ignoring any resultant fabric evolution. In this context it is variation in the response to applied stresses under rotations of the anisotropic material relative to the stress distribution that could be said to characterise an anisotropic flow relation.

In contrast, the applicability of the flow relation we are implementing from Budd et al. (2013) is limited to ice undergoing tertiary flow, i.e., ice with an anisotropic crystal fabric induced by prolonged deformation under the same stress regime. Indeed, the directional sense of anisotropy of the fabric, its character and the resultant mechanical properties, are all characterised by the nature of the stress regime. This is not a situation amenable to considering arbitrary rotations of a material element relative to the stresses. Accordingly, as presented completely in terms of the stresses, it is an isotropic flow relation, for material with

a flow-compatible induced anisotropy.

The general flow relation constructed by Glen (1958) was empirically formulated on the basis of isotropy, involving only the tensors of strain-rates and deviatoric stresses, and their scalar invariants. In the most general expression provided, (Eq. (4) of Glen, 1958) the flow relation was not collinear, and by including possible dependence on the cubic invariant of the stress tensor it also contained a measure of the character of the pattern of stresses as well as the magnitude (given by the second invariant).

Indeed, consideration of a possible dependence on the third (cubic) invariant of $\boldsymbol{\sigma}'$ as an explanation of the dependence of tertiary flow rates on different stress regimes is a recurrent suggestion (e.g., Baker, 1987; Morland, 2007).

As Faria (2008) points out in discussing the CAFFE flow relation, this is a different issue from whether the flow relation involves a rank-four tensor connecting strain rates and deviatoric stresses. The CAFFE and ESTAR flow relations are both scalar (collinear) relations between deviatoric stresses and strain rates, yet only the CAFFE flow relation, whose "deformability"

parameter involves both stresses and the anisotropic crystal orientations is an anisotropic constitutive relation in the sense discussed above.

## 4    Implementation of the ESTAR flow relation

The magnitude of the shear strain rate defined on the local non-rotating shear plane, $\dot{\varepsilon}'$ for Eq. (6), is central to the formulation of the ESTAR flow relation (Eqs. (3)-(4)). The full prescription, following Budd et al. (2013), involves the expression

$$\dot{\varepsilon}' = \|\dot{\boldsymbol{\varepsilon}} \cdot \mathbf{n} - (\mathbf{n} \cdot (\dot{\boldsymbol{\varepsilon}} \cdot \mathbf{n})) \mathbf{n} - (\hat{\boldsymbol{\omega}}_D \cdot (\dot{\boldsymbol{\varepsilon}} \cdot \mathbf{n})) \hat{\boldsymbol{\omega}}_D\|, \tag{7}$$

where: $\mathbf{n}$ is the unit normal to the non-rotating shear plane, $\hat{\boldsymbol{\omega}}_D$ is the unit vector parallel to that part of the vorticity vector that is associated solely with deformation ($\boldsymbol{\omega}_D$), and $\dot{\boldsymbol{\varepsilon}}$ is the strain rate tensor. The unit normal to the non-rotating shear plane, $\mathbf{n}$,

is defined as the normalised cross product of the velocity ($\mathbf{v}$) and the deformational vorticity vector ($\boldsymbol{\omega}_D$)

$$\mathbf{n} = \frac{\mathbf{v} \times \boldsymbol{\omega}_D}{\|\mathbf{v} \times \boldsymbol{\omega}_D\|}. \tag{8}$$

The last projection term in Eq. (7) was proposed in Budd et al. (2013) to prevent any shear component parallel to the deformational vorticity from contributing to the measure of simple shear.

5 The vorticity of a flow, whether viewed as the anti-symmetrised part of the velocity gradient tensor or as the usual vector $\boldsymbol{\omega} = \nabla \times \mathbf{v}$, contains motions associated with both deformation and local rigid-body rotation. The locally non-rotating shear plane is intended to be rotating with any rigid rotation portion of the flow field, so it is only vorticity associated with the deformation process that is relevant to determining the shear fraction. Accordingly we formally decompose vorticity into deformational and rotational parts:

10 $$\boldsymbol{\omega} = \nabla \times \mathbf{v} = \boldsymbol{\omega}_D + \boldsymbol{\omega}_R. \tag{9}$$

For the present implementation it is convenient to decompose the vorticity further, into vectors perpendicular and parallel to the velocity direction as follows:

$$\boldsymbol{\omega} = \boldsymbol{\omega}_D^\perp + \boldsymbol{\omega}_R^\perp + \boldsymbol{\omega}_D^\parallel + \boldsymbol{\omega}_R^\parallel. \tag{10}$$

From Eq. (8), only the perpendicular projection $\boldsymbol{\omega}_D^\perp$ of the deformational vorticity is relevant in determining the direction of 15 the normal to the non-rotating shear plane. This is fortunate since $\boldsymbol{\omega}_R^\perp$ the perpendicular projection of the rotational vorticity can be calculated directly for steady flow from the flow speed and the curvature of the local streamline, and is oriented along the binormal (the unit vector orthogonal to both the tangent vector and the normal vector) to the streamline. The decomposition of the component of vorticity parallel to the flow direction, conventionally termed swirling motion, into deformational and rotational pieces is not so straightforward, but we can use the following expression, which can be calculated using variables 20 available within an individual element of ISSM to generate a vector suitable for computing $\mathbf{n}$:

$$\tilde{\boldsymbol{\omega}}_D = \nabla \times \mathbf{v} - \frac{2\mathbf{v} \times ((\mathbf{v} \cdot \nabla)\mathbf{v})}{\|\mathbf{v}\|^2}. \tag{11}$$

This vector contains the correct perpendicular component $\boldsymbol{\omega}_D^\perp$ to compute $\mathbf{n}$ using Eq. (8), but contains all of $\boldsymbol{\omega}_D^\parallel + \boldsymbol{\omega}_R^\parallel$. We can obviously project out the component parallel to velocity to find

$$\boldsymbol{\omega}_D^\perp = \tilde{\boldsymbol{\omega}}_D - (\mathbf{v} \cdot \tilde{\boldsymbol{\omega}}_D)\frac{\mathbf{v}}{\|\mathbf{v}\|^2}. \tag{12}$$

25 In the present implementation of the ESTAR flow relation, we assume that swirling effects are small for flows with the relevant spatial scales, aspect ratios etc., which can be verified from the modelled flow-fields in our test cases, and hence $\boldsymbol{\omega}_D^\parallel$ is also expected to be small. We use the unit vector corresponding to $\boldsymbol{\omega}_D^\perp$ (i.e., $\hat{\boldsymbol{\omega}}_D$) in Eq. (7) for our computation of $\dot{\varepsilon}'$. This corresponds to extracting the component of the shear resolved on the non-rotating shear plane which is parallel to the velocity direction, which could be regarded as an alternative generalisation for the simple shear to that proposed by Budd et al. (2013).

No approximation is involved for flows that are exactly two dimensional in character, since vorticity is always orthogonal to velocity in such situations.

The description of the ESTAR flow relation above is implemented in ISSM for the full-Stokes model of flow. We also extended the implementation to ISSM versions of the higher-order three-dimensional model of Blatter (1995) and Pattyn (2003), and the shallow-shelf approximation (SSA) of MacAyeal (1989). The higher-order model is derived from the full-Stokes model by assuming that horizontal gradients in the vertical velocities are negligible ($\partial v_z/\partial x = \partial v_z/\partial y = 0$) compared with vertical gradients in the horizontal velocities when computing vertical shear, and longitudinal derivatives of vertical shear stress (bridging effects, van der Veen and Whillans, 1989) are ignored. The higher-order vertical velocities are recovered directly through incompressibility. Extending on the higher-order model assumptions, for the SSA model, vertical shear is assumed to be negligible ($\dot{\varepsilon}_{xz} = \dot{\varepsilon}_{yz} = 0$). For both the higher-order and SSA models, the approximations will affect calculations of the total vorticity and hence the magnitude of the shear strain rate on the non-rotating shear plane, Eq. (7), and $\lambda_S$, Eq. (6).

## 5    Analytical verification

We perform convergence tests in order to verify the implementation of the ESTAR flow relation within the ISSM full-Stokes and higher-order models. The objective of these tests is to compare the model results to analytical solutions for different mesh resolutions. As the mesh becomes finer, the error between the model and the analytical solution (i.e., $\sqrt{\int_\Omega (X - X_a)^2 / \int_\Omega X_a^2}$, for model solution $X$, analytical solution $X_a$, and domain $\Omega$) should decrease, with a cubic dependence on resolution for full-Stokes (quadratic for ice pressure) when using Taylor-Hood finite elements, and a quadratic dependence for higher-order using P1$\times$P1 finite elements (e.g., Ern and Guermond, 2004).

We designed our analytical solutions by considering a three-dimensional, grounded, isothermal ice slab of unit dimension lying on a flat bed topography, with cartesian coordinates $(x, y, z)$, where $z$ is vertically upward and where there is no gravitational force. The full-Stokes three-dimensional velocity field is given by

$$v_x(x, y, z) = 3z, \tag{13}$$
$$v_y(x, y, z) = 2x + y, \tag{14}$$
$$v_z(x, y, z) = -z, \tag{15}$$

and the higher-order velocity field by

$$v_x(x, y, z) = x^2, \tag{16}$$
$$v_y(x, y, z) = 3z + y. \tag{17}$$

In the case of the higher-order model, $v_z(x, y, z)$ is recovered by incompressibility. For both full-Stokes and higher-order models, we use shear and compression enhancement factors of $E_S = 3$ and $E_C = 1$, and the flow parameter $A(T') = 2/3 \, \mathrm{Pa}^{-3}\,\mathrm{s}^{-1}$. The open source mathematics software system SageMath (http://www.sagemath.org/) is used to calculate analytical solutions

for the force balance equations based on the above velocity fields:

$$f_x(x,y,z) = -\left( \frac{\partial \sigma'_{xx}}{\partial x} + \frac{\partial \sigma_{xy}}{\partial y} + \frac{\partial \sigma_{xz}}{\partial z} \right), \tag{18}$$

$$f_y(x,y,z) = -\left( \frac{\partial \sigma_{xy}}{\partial x} + \frac{\partial \sigma'_{yy}}{\partial y} + \frac{\partial \sigma_{yz}}{\partial z} \right), \tag{19}$$

$$f_z(x,y,z) = -\left( \frac{\partial \sigma_{xz}}{\partial x} + \frac{\partial \sigma_{yz}}{\partial y} + \frac{\partial \sigma'_{zz}}{\partial z} \right). \tag{20}$$

Here, the deviatoric stress fields are calculated using the ESTAR flow relation as specified in Eq. (3). When the total, rather than deformational, vorticity (i.e., without inclusion of the rigid body correction or removal of the vorticity component aligned with the flow) is used in the calculation of the ESTAR flow relation, the full-Stokes analytical solution for $(f_x, f_y, f_z)$ comprises (20 521, 9 190, 20 523) characters. By contrast, non-trivial analytical solutions for the forcing functions that are calculated from an anisotropic enhancement factor that is based on the deformational, rather than total, vorticity are at minimum 200 000

characters, well in excess of the character limits for most compilers. Accordingly, we verify our implementation of the ESTAR flow relation using the total, rather than the deformational, vorticity.

  To test the numerical implementation ISSM is forced using the analytical expressions for $f_x$, $f_y$, and $f_z$ in Eqs. (18)-(20) and the resulting three-dimensional flow field is compared with the relevant analytical specification in Eqs. (13)-(17). Since the aim is to verify correct coding of the modifications within ISSM for the ESTAR flow relation we apply the analytic velocities on

the faces as the boundary conditions. Four sets of element sizes are used for each of the full-Stokes and higher-order models, increasing in resolution from 0.2 (272 elements over 5 vertical layers) to 0.08 (4656 elements over 13 vertical layers). We find convergence powers of 2.5 ($v_x$), 3.1 ($v_y$), and 2.6 ($v_z$) for full-Stokes, respectively, and 1.4 ($v_x$) and 1.1 ($v_y$) for higher-order (Fig. 2), which are consistent with theory (e.g., Ern and Guermond, 2004) and verify our implementation.

## 6 Application of the ESTAR flow relation to idealised scenarios

The ESTAR flow relation was applied to a suite of test cases. The first case we present simulates flow in an embayed ice shelf; the second two are based on experiments from the Ice Sheet Model Intercomparison Project for Higher Order Models (ISMIP-HOM; Pattyn et al., 2007). The ISMIP-HOM experiments describe idealised scenarios of ice flow where the bed topography or basal friction vary. ISSM has already been validated against the ISMIP-HOM experiments (Larour et al., 2012).

  The ISMIP-HOM experiments were diagnostic. In contrast, we have taken the same geometries and boundary conditions,

which are already familiar to the modelling community, but allowed the velocity, surface, and thickness fields to evolve to steady-state, as defined in the corresponding sections below. We choose to present steady-state results from prognostic simulations based on ISMIP-HOM experiments (supplemented by prescribing zero accumulation or loss of ice) using the Glen and ESTAR flow relations because in this situation the ice sheets, the flow fields, and stress regimes are steady in time. This is more in keeping with the assumptions underlying the ESTAR flow relation than a simple diagnostic experiment for a prescribed ge-

ometry. It is also of interest to see the differences in the dynamic evolution of the systems resulting from the different material constitutive relations. While our focus is on the differences between the results for the ESTAR and Glen flow relations, the

latter results provide a direct extension to the original ISMIP-HOM experiments presented in Pattyn et al. (2008), which may be of some interest, rather than being directly comparable with them. We append a "p" for prognostic to experiment names, where appropriate. The ice sheet is isothermal, as in the original ISMIP-HOM experiments.

As mentioned above, we use shear and compression enhancement factors of $E_S = 8$ and $E_C = 3$, respectively (Treverrow et al., 2012). For each experiment, we performed simulations using a range of Glen enhancement factors (1, 3, 5, and 8), but since these idealised experimental systems have simple scaling properties under global changes in flow rates, we present only results for $E_G = 8$ since that proved the most directly relevant value. The ISMIP-HOM experiments used the original parameter values (Pattyn et al., 2007) unless otherwise indicated.

## 6.1 Flow through an embayed ice shelf

The first prognostic experiment simulates three-dimensional flow through a rectangular embayed ice shelf. The experiment was carried out for model domains with transverse spans $x \in [0, L]$, for $L = 20$, 60, and 100 km and along-flow dimension $y \in [0, 100]$ km. The initial ice thickness decreases uniformly from 1000 m at the grounded zone to 300, 600, and 850 m at the ice front for the $L = 20$, 60, and 100 km cases, respectively. The main features of the anisotropic effects are similar regardless of aspect ratio. This is principally because wider embayed ice shelves are flatter so that the influence of simple shear stresses on the dynamics is not particularly sensitive to aspect ratio. Accordingly, we focus our discussion on one transverse length scale: $L = 20$ km. The plan view mesh is extruded ten quadratically-spaced layers in the vertical. A no-slip boundary condition is applied along the $x = 0$ and $x = L$ side boundaries. At the inflow boundary, the $y$-component of velocity is set by

$$V(x) = V_0 e^{-\left[\frac{5(x - x_{\mathrm{mid}})}{2L}\right]^8}, \tag{21}$$

$$v_y(x, 0) = V(x) - V(0), \tag{22}$$

where $V_0 = 100$ m yr$^{-1}$ and $x_{\mathrm{mid}} = L/2$. This ensures that $v_y(x, 0)$ satisfies the no-slip boundary condition on the margins. As is standard, ocean water pressure is applied at the evolving ice-ocean interface where tangential (traction) stresses vanish. It is assumed that there is no surface or basal melting or accumulation over the ice shelf domain. The flow parameter $A(T') = 1.74 \times 10^{-25}$ Pa$^{-3}$ s$^{-1}$, is set using the Budd and Jacka (1989) value for an isothermal ice shelf of $-20$ °C. We consider the case where the Glen enhancement factor is equal to the ESTAR shear enhancement factor, i.e., $E_G = E_S = 8$.

We run the higher-order ice flow model for each of the ESTAR and Glen flow relations to steady-state, which we define to be reached when the mean velocity change over the surface mesh points is less than $1 \times 10^{-2}$ m yr$^{-1}$ between two consecutive time steps (of $\triangle t = 2$ yr).

The Glen and ESTAR higher-order steady-state surface velocity magnitudes are compared in Fig. 3. The patterns of ice flow are similar: in each case the ice velocity increases as it flows through the ice shelf, reaching its maximum at the ice front. Over most of the domain the velocities are in close agreement, reflecting the dominance of the shear flow. However, the Glen velocities are up to 17% larger than the ESTAR velocities at the ice front, where the flow field is predominantly tensile in accordance with the ice front boundary conditions. The differences in velocities can be attributed to differences in flow enhancement factors for simple shear and compression. Near the centre line of the ice shelf and across the ice front, where

longitudinal and vertical normal stresses dominate, the Glen enhancement is as much as $8/3$ times larger than the corresponding ESTAR enhancement.

The steady-state thickness patterns for each flow relation, and their ratio are shown in Figs. 3d-f. In both cases, the equilibrated ice shelf is thicker along the centre line and thinner towards the side margins where ice flow is slower, and thicknesses agree within 5% over much of the domain. However, the Glen ice shelf is consistently thinner than the ESTAR ice shelf, particularly along the centre line where the Glen velocity is enhanced relative to the ESTAR case, and it is up to 20% thinner at the ice front.

The ESTAR strain rate components are presented in Fig. 4. As expected, shear strain rate in the $x-y$ plane is very high near the lateral boundaries (Fig. 4d). However, it dominates the effective strain rate (and hence $\lambda_S$) well beyond those margins (Fig. 5), before decreasing towards the centre line, where it identically vanishes. Towards the ice-ocean front, each of the normal strain rates – $\dot{\varepsilon}_{xx}$, $\dot{\varepsilon}_{yy}$, and $\dot{\varepsilon}_{zz}$ – increase in magnitude, reaching their absolute maxima at the front. The (approximately) longitudinal $\dot{\varepsilon}_{yy}$ is the dominant normal strain rate component and is extensional towards the front. Due to the confined geometry, towards the front $\dot{\varepsilon}_{yy}$ is largely balanced by $\dot{\varepsilon}_{zz}$, which drives ice shelf thinning. Transverse normal strain rate $\dot{\varepsilon}_{xx}$ plays a lesser role at the ice-ocean front than the other normal strain rates. It is extensional along the front as the streamlines diverge, but changes sign to compressive towards the corners. The patterns in the component strain rates, including the dominance of normal strain rates in the centre of the ice shelf and at the ice-ocean front, are evident in the strain rate on the non-rotating shear plane ($\dot{\varepsilon}'$) and the effective strain rate ($\dot{\varepsilon}_e$), the ratio of which sets the magnitude of $\lambda_S$ (Fig. 5). In the embayed ice shelf simulation using the ESTAR flow relation, the vanishing of basal traction and the depth independent nature of the inflow velocity lead to an almost 2D flow field with local non-rotating shear planes essentially vertical where they can be defined – there being neither $x-y$ plane shear nor vorticity along the centre line. We note that $\dot{\varepsilon}_{yy}$ decreases in magnitude with depth at the ice-ocean front, coincident with ice front tilting (Weertman, 1957), which also gives rise to some local shear deformation in the $y-z$ plane.

Computation times for serial and parallel simulations with the higher-order model for the embayed ice shelf, using each flow relation and for increasing number of processors, are summarised in Table 1. For parallel simulations, the ESTAR flow relation computation times are on average approximately 6% longer than the corresponding times for the Glen relation, but the parallel efficiency of the ESTAR flow relation is marginally greater than that of the Glen relation for each parallel simulation.

To check the computational demands of the full ISSM model with the ESTAR flow relation, the full-Stokes ice flow model was computed for one model year (i.e., steady state had not yet been reached) and the results compared with the higher-order simulation results for the same model period (results not shown). Use of the ESTAR flow relation increased walltimes by $< 3\%$, since the flow relation is an even smaller part of the full-Stokes computation. At the ice front, the higher-order velocities are everywhere within 5% of the full-Stokes velocities, with the maximum differences occurring near the lateral boundaries. Across the shelf, the higher-order component velocities accord well with the full-Stokes velocities. The magnitude and spatial patterns of $\dot{\varepsilon}'$, $\dot{\varepsilon}_e$, and $\lambda_S$ also agree well between the full-Stokes and higher-order models.

## 6.2 ISMIP-HOM experiment B: two-dimensional flow over a bumpy bed

ISMIP-HOM Experiment B (ISMIPB) describes two-dimensional flow ($x$ horizontal, $z$ vertical) over a bed topography that varies sinusoidally, according to the following equation

$$z_b(x) = z_s(x) - 1000 + 500 \sin\left(\frac{2\pi x}{L}\right), \tag{23}$$

where $z_s(x) = -x \tan\alpha$, for a mean bed slope of $\alpha = 0.5°$, and $L$ controls the scale of the bedrock undulation. We take $z_s(x)$ as the initial surface so that the mean initial ice thickness is 1000 m. Prognostic experiments require specification of the local mass balance; here, there is no surface or basal melting or accumulation at any point in the domain. We present results for $L = 20$ km and $L = 5$ km to explore the influence of different longitudinal stresses. We consider the case when the Glen enhancement factor is equal to the ESTAR shear enhancement factor ($E_G = E_S = 8$), which is the most relevant

case as the dynamics are driven by bed-parallel shear. The flow parameter is fixed at $3.96 \times 10^{-25}$ Pa$^{-3}$ s$^{-1}$, corresponding to an ice temperature of approximately $-14°$ C (Budd and Jacka, 1989). We have reduced the original flow parameter by a factor of 8 (i.e., equal to $E_S$) to ensure the Glen flow relation solution corresponds to the original ISMIP-HOM experiments. Periodic boundary conditions are applied at the vertical edges of the domain and a no-slip boundary condition is applied at the base. In this and the following two-dimensional test case, the normal to the non-rotating shear plane is simply the direction

perpendicular to the velocity and there is no uncertainty about the vorticity (which has only one non-vanishing component) being perpendicular to the velocity. In each case we used the full-Stokes version of ISSM to carry out a *prognostic* simulation of the ISMIPB experiment (the original ISMIPB was diagnostic). Prognostic steady-state is regarded as reached when the mean velocity change over the surface mesh points is less than $1 \times 10^{-2}$ m yr$^{-1}$ between two consecutive time steps of $\triangle t = 1$ yr for this and the following ISMIP-HOM experiment.

20       The ESTAR and Glen full-Stokes prognostic steady-state horizontal velocities ($v_x$) for ISMIPBp for $L = 20$ km are shown in Fig. 6a and b. The ESTAR velocities are marginally slower than the Glen velocities throughout the domain, regionally by as much as 6% (Fig. 6c). While in a real-world situation, a local difference of 6% may not be significant to overall flow, unless the Glen enhancement factor is approximately $E_S$ there would be a significant and widespread difference in velocities. One major contrast occurs in the near-surface layers either side of the topographic bump (Fig. 6c) where normal stresses dominate

($\lambda_S < 1$; Fig. 6e) and $E(\lambda_S)$ tends to $E_C < E_G$, as $\lambda_S$ tends to 0. This reduces the shear deformation in the upper-layers for ESTAR compared to Glen, leading to slightly lower horizontal velocities near the surface. We will discuss the relevance of this in Sect. 7. Another major velocity contrast occurs in the lowest part of the ice sheet directly above the topographic depression. Since deformation here is clearly shear dominated ($\lambda_S = 1$ for essentially the whole column), the differences must arise from a varying but consistently lower shear stress profile in the ESTAR case, reflecting indirectly the distributed effect of the stiffer

ice in the upper layers where $E(\lambda_S) < E_G$. In contrast, the closest agreement between the two velocity distributions is in the basal region over the topographic high.

The steady-state surface elevations for the Glen and ESTAR flow relations are everywhere within 1 m (Fig. 6d). The differences between initial and final elevations show that the Glen steady-state surface is slightly lower than the ESTAR surface

between 0-5 km and 15-20 km and slightly higher between 5 and 15 km. This reflects differing contrasts in the horizontal velocities for the two flow relations. Greater differences are seen in the next example.

Figure 6 also shows the full-Stokes $\dot{\varepsilon}_{xz}$ and $\dot{\varepsilon}_{xx}$ strain rates for the simulation using the ESTAR flow relation: effectively the shear and normal strain rates. The dominance of high values of $\lambda_S$ indicates that bed-parallel simple shear is the main driver of ice flow in ISMIPBp with the expected transition through the ice column from compression/extension-dominated flow near the surface to shear-dominated flow near the non-slip bed. Note that while the component strain rates are presented in the background cartesian frame, $\lambda_S$ denotes the relative importance of simple shear on local non-rotating shear planes. Peaks in $\lambda_S$ appear directly over the topographical bump and depression, extending further into the surface layers than in surrounding regions. The locations of the peaks in $\lambda_S$ correspond to the transitions between tensile and compressive longitudinal stresses, centred on "transition curves" (Fig. 6f), along which normal strain rates are identically zero.

In order to examine the dynamics giving rise to the high shear-dominance peaks in Fig. 6e, we consider the following exact form of $\lambda_S^2$ (for these two-dimensional flow fields) expressible using the cartesian frame strain rate components

$$\lambda_S^2 = \frac{\alpha \dot{\varepsilon}_{xx}^2 + \beta \dot{\varepsilon}_{xz}^2 + \gamma \dot{\varepsilon}_{xx}\dot{\varepsilon}_{xz}}{\dot{\varepsilon}_{xx}^2 + \dot{\varepsilon}_{xz}^2}, \tag{24}$$

for some spatially varying coefficients $\alpha$, $\beta$, and $\gamma$. Since there is no surface accumulation, velocities and hence local non-rotating shear planes at the ice sheet surface are parallel to the surface. The traction free surface boundary condition implies that the numerator ($\dot{\varepsilon}'^2$) in Eq. (24), and accordingly $\lambda_S$, vanishes at the surface, except that if $\dot{\varepsilon}_e$ also vanishes, $\lambda_S$ is technically undefined. Our implementation sets $\lambda_S = 0$ for vanishing $\dot{\varepsilon}'$ in such situations. It is apparent from Eq. (24) that along the transition curves, i.e., where $\dot{\varepsilon}_{xx} = \dot{\varepsilon}_{zz} = 0$, $\lambda_S^2 = \beta$, independent of (non-zero) $\dot{\varepsilon}_{xz}$ strain rate. One can show that $\beta \to (1 - S_x^2)^2$ towards the surface (i.e., for surface slope in the $x$-direction $S_x$) along the transition curve, in order to satisfy the surface boundary condition. This indicates that $\lambda_S$ would be finite along the transition curves all the way to the surface, except that we enforced its vanishing there. For these locations, the Glen and ESTAR viscosities corresponding to Eqs. (2)-(3) would tend to infinity as $\dot{\varepsilon}_e$ vanished approaching the surface, but are limited to a maximum value in the ISSM implementation.

Away from the transition curves $\lambda_S \to 0$ as we approach the surface, associated with vanishing shear on the non-rotating shear plane and a corresponding dominance of normal deformations. We return to the near-surface peaks in $\lambda_S$ in the discussion. We also investigated the impact of reducing the horizontal extent to $L = 5$ km. In this steeper bed scenario (Fig. 7), the ESTAR velocities are at least 11% slower than the Glen velocities in the surface layers, as much as 20% slower around the topographic bump, and up to 25% slower in the topographic depression (Fig. 7c). The reductions in the ESTAR velocity magnitudes are a consequence of the increasing importance of longitudinal stresses in the stress balance equations for the smaller aspect ratio (Fig. 7f), and also in some areas the lower strain rates, which lead to correspondingly stiffer ice. The patterns in the steady-state surface elevations for the $L = 5$ km case (Fig. 7d) are consistent with the $L = 20$ km case (Fig. 6d), although here the differences between the steady-state surface elevations for each flow relation are greater ($\pm 1.25$ m).

## 6.3 ISMIP-HOM experiment D: two-dimensional flow over a sticky spot

ISMIP-HOM experiment D (ISMIPD) describes a two-dimensional domain over which the basal friction coefficient $\chi$ varies sinusoidally in the horizontal direction. A Paterson-type friction law (Paterson, 1994, p.151) of the following form is used

$$\tau_b = -\chi^2 v_b, \tag{25}$$

where $\tau_b$ is the basal stress, $v_b$ the basal velocity, and the friction coefficient, $\chi^2$ (Pa yr m$^{-1}$), varies according to the equation

$$\chi^2 = 1000 + 1000 \sin\left(\frac{2\pi}{L}x\right). \tag{26}$$

The bed topography and the initial ice surface are inclined planes with a slope of $0.1°$, and the initial thickness is 1000 m throughout the domain. There is no surface or basal melting or accumulation at any point in the domain. As in the preceding ISMIPBp experiment, the control of the final deformation flow in the ISMIPDp experiment is bed-parallel shear, so we consider the case $E_G = E_S = 8$, for a flow parameter of $A(T') = 3.96 \times 10^{-25}$ Pa$^{-3}$ s$^{-1}$. Periodic boundary conditions are applied at the edges of the domain. We present results for steady-state solutions for two different horizontal scales: $L = 20$ km and $L = 5$ km. Once again, we employ the full-Stokes solver in ISSM for *prognostic* simulations evolved to steady-state (as defined in Sect. 6.2).

The steady-state Glen and ESTAR results when $L = 20$ km are shown in Fig. 8. In both cases, the fastest velocities develop in the upper part of the column over the sticky spot (centred around 5 km). This is required by continuity, given the lower basal velocities over the sticky spot, to balance the block flow over the slippery section ($\chi = 0$ at 15 km). The total steady-state ESTAR velocities are everywhere within 1% of the corresponding Glen velocities (see Fig. 8c), with the maximum difference occurring in the near-surface layers over the sticky spot, where the ice is stiffer in the ESTAR case (Fig. 8f). To explore what differences might be associated with the departure from block flow we calculated "deformational velocities" from $v_x$ by subtracting the basal motion, and examined the ratio between ESTAR and Glen cases (Fig. 8d). The most notable feature of this ratio is a localised $\sim$50% decrease in deformational velocity for ESTAR compared to Glen through the entire column, peaking directly over the point where the basal stress vanishes. This major band of difference coincides with the band of higher viscosity for ESTAR, relative to Glen (see Fig. 8f), over the slippery region.

The ISMIPDp steady-state surface elevations for each flow relation are shown in Fig. 8e. The Glen steady-state surface is lower than ESTAR over the sticky spot and higher than the ESTAR surface over the slippery spot. The absolute differences between the two steady-state surfaces are everywhere less than $0.42$ m. The evolution of the ice thickness and surface by approximately 5 m above and below the initial linear profile produces a marked change in the surface slope, given the $0.1°$ slope of the bedrock ramp, which corresponds to a fall of 34.9 m over the 20 km domain.

The ESTAR component strain rates and shear fraction $\lambda_S$ are illustrated in Fig. 8g-i for the $L = 20$ km case. Transitions exist in the deformation regime from shear-dominated to compression/extension-dominated and back over a few kilometres around the slippery spot (Fig. 8g). These transitions extend to the bed, becoming perpendicular to the ice flow direction and reflect the low shear over the slippery region. This is in contrast to the ISMIPBp experiments, where the shear deformation regime dominated except near the free surface. In Sect. 7 we discuss the implications of these abrupt transitions for the assumptions

that underpin the ESTAR flow relation. Shear dominates much of the ice column ($\lambda_S > 0.5$) throughout the rest of the domain (over the region where values of the basal friction coefficient exceed 10% of the maximum value). The shear strain rates are greatest in the region of maximum stickiness at approximately 5 km, and a steep profile in the shear strain rate is present there. Naturally, transitions between tensile and compressive flows occur around the sticky spot and the transition curves (vanishing normal strain rates) resemble the ISMIPBp cases (Fig. 6f). Here the transition curves reach the surface as the friction coefficient increases and just downstream of its maximum value, leading to near-surface peaks of high values $\lambda_S$ analogous to ISMIPBp. It should be noted that the strain rates here are very small compared to the ISMIPBp experiments. The viscosity ratio in Fig. 8f reveals that there are also lower strain rates for the ESTAR case in the compression dominated regions, as the ratios there are higher than the factor of 1.39 that would be produced by the influence of $E(\lambda_S)$ alone.

Steady-state simulation results for the smaller aspect ratio ($L = 5$ km) are presented in Fig. 9. The $v_x$ velocity ratio (Fig. 9c) shows very little difference between results for the two flow relations. However, there is a significantly different picture in the ratio of deformation velocities seen in Fig. 9d compared to Fig. 8d: the largest differences are now limited to the lower portion of the ice column and for much of the region over the slippery bed the ratio is almost unity. In the $L = 5$ km case, we see that a slight reverse slope develops downstream of the slippery spot for both flow relations, while the highest point occurs approaching the peak of the basal drag coefficient. Again, the steady-state surface elevations differences (Fig. 9e) show similar patterns in the $L = 5$ km case as in the $L = 20$ km case, with smaller differences between the two final surface elevations (i.e., 0.28 m). The deviations from the initial profile are approximately half that observed for the $L = 20$ km, but the fall in the bed (and the initial profile) over 5 km is only 8.73 m.

The pattern of deformation regimes mapped by $\lambda_S$ for $L = 5$ km (Fig. 9g) is also more complex than that seen in the preceding $L = 20$ km case (Fig. 8g). The general structure of the normal strain rates is similar to previous experiments, but here the persistence of a band of shear (Fig. 9i) above the slippery spot at intermediate depths prevents the establishment of a vertical block of flow dominated by normal stresses. The shear profile above the sticky spot is much weaker in the upper layers. Accordingly, $\lambda_S$ reveals a band of unevenly shear-dominated deformation which is continuous across the periodic domain. Once again, shear dominated peaks extend towards the surface in association with the vanishing of the normal strain rates.

The spatial variations in the viscosity ratio (Fig. 9f) depart significantly from those of $\lambda_S$, reflecting more strikingly than for $L = 20$ km (Fig. 8f) the combined influence of the pattern of enhancement (controlled by $E(\lambda_S)$) and the effect of different strain rates, with values both above and below the range (1.0-1.39) directly controlled by $E(\lambda_S)/E_G$.

## 7    Discussion

In this study we conducted various ice flow simulations, comparing the ESTAR and Glen flow relations. The ESTAR flow relation incorporates the observed differences between tertiary deformation rates for shear dominated and normal stress dominated stress regimes.

Our simulations of embayed ice shelf flow showed that no single Glen enhancement factor ($E_G$) can reproduce the anisotropic flow characteristic of the various stress regimes encountered. Significantly, while a choice of $E_G = E_S$ was necessary to repro-

duce the same overall velocities and ice thicknesses, which are largely controlled by lateral shear, this overestimated velocities near the ice front by up to 17% compared to the results using the ESTAR flow relation. This is a consequence of softer ice in the Glen case for the zone near the ice front, where extensional longitudinal stresses dominate. The steady-state Glen ice shelf was accordingly up to 20% thinner than the ESTAR ice shelf in this region. Even with this thinner ice influencing the ocean pressure boundary condition at the ice front, the softer ice in the Glen case meant that the longitudinal strain rates there were higher.

These results highlight one of the key failures of the Glen flow relation: an inability to account for complex, spatially varying stress regimes in its prescription of ice flow. The addition of an enhancement factor $E_G$ to the Glen flow relation permits some compensation for the flow enhancement associated with microstructural development (i.e., rescaling the minimum creep rate data conventionally used in prescribing the Glen flow relation, e.g., Table 3.3, Cuffey and Paterson, 2010). However, such a modification does nothing to allow for the stress configuration dependent aspects of ice deformation rates, associated with the development of anisotropic crystal fabrics, that are characteristic of tertiary creep. The improvement offered by the ESTAR flow relation is that the specification of the pattern and degree of enhancement is physically based, varying spatially as a function of the stress configuration. This is achieved without the complication of a detailed treatment of microstructural information.

The modified ISMIP-HOM experiments B and D simulated scenarios in which the dominant control of flow was bed-parallel simple shear. In the prognostic runs with the larger aspect ratio ($L = 20\,\text{km}$), only small differences were apparent between the Glen and ESTAR velocities ($< 6\%$ for ISMIPBp and $< 1\%$ for ISMIPDp), again provided the Glen enhancement factor was chosen equal to the ESTAR shear enhancement factor ($E_G = E_S$).

For more rapidly varying bed topography in ISMIPBp, with $L = 5$ km, the differences in velocity for the two flow relations reached $\sim 25\%$, with surface variations of $\sim 11\%$. For ISMIPDp, which explored variations in basal friction, the $L = 5$ km experiments still showed $< 1\%$ differences in velocities with very low strain rates, so that although a complex pattern of deformation regimes emerged, there was little effect on flow from the choice of flow relation.

These results suggest that if major bed topography varied only on scales much longer than the ice thickness, close agreement between simulations using the ESTAR and Glen flow relations might be achieved more generally by choosing the tertiary shear enhancement factor as the Glen enhancement factor ($E_G = E_S$). This might provide a physical rationale to replace the *ad hoc* enhancement factors typically used in large-scale grounded ice sheet modelling with the value appropriate to flow dominated by simple shear. However, larger differences between velocities and vertical shear profiles emerged for the more rapid bedrock variation, where the importance of including longitudinal stresses in the momentum balance is already recognised (Pattyn et al., 2008), suggesting that adopting the ESTAR flow relation would be preferable.

Our idealised test cases also provide some insights into the validity of the tertiary flow assumption underlying the ESTAR flow relation, and the development of anisotropic crystal fabrics compatible with the current deformation regime. In the embayed ice shelf test the most significant change in the deformation regime is clearly the transition to extensional flow on approach to the ice shelf front. The contours of $\lambda_S$ here are relatively well aligned with the ice flow so that flowing ice experiences gradual changes in stress regime, and the magnitudes of strain rates (e.g., Fig. 4b) and velocities (Fig. 3) indicate that in the region near the ice front, where the results for the Glen and ESTAR flow relations differ appreciably, a progression of

essentially compatible fabrics would be maintained. Indeed, under the prevailing deformation and flow conditions these would even develop from random fabrics over a few km.

The ISMIP-HOM experiments reveal potential violations of the tertiary flow assumption, although the significance for the flow field of these apparent short-comings needs to be assessed with regard to the somewhat artificial nature of the tests. Indeed as we saw, the difference between the results of the ESTAR and Glen flow relations (provided $E_G = E_S$) was small, except for ISMIPBp with $L = 5$ km, although of course the ESTAR flow relation makes no claim to correctly describe the transient deformation rates of ice with an evolving anisotropy.

The ISMIP-HOM experiments have a spatial periodicity, which could allow one portion of the repetitive basal conditions to dominate the overall flow. Also, there is no surface mass budget in these experiments so that, as remarked earlier, the ice surface is a streamline, whereas in a system with surface accumulation fresh snow is always being added and advected down into the ice sheet where it makes the transition to solid ice. Accordingly, in the flow regime of these prognostic experiments even the surface layers would be regarded as having developed some anisotropy just as the lower layers would, since they have in principle been deforming over an arbitrarily long time.

The main issue about the establishment of tertiary flow conditions in the periodic environment of our ISMIP-HOM experiments concerns the possible cycling of the flowing ice through a variety of stress regimes. This leads to transition regions where the stress regime and presumably the crystal anisotropy would be evolving, and the compatibility assumptions behind the ESTAR flow relation would locally be violated. Clearly the spatial extent of transitional flow and the delay in attaining any new tertiary state depends on the magnitudes of the strain rates and the velocity of the ice. By combining these with a threshold for accumulated strain as the criterion for development of a compatible (tertiary) fabric under a persistent flow regime, the extent of a transition zone can be estimated. This scale can then be compared to the horizontal variation of the stress regime. Selecting the 10% strain required to develop a compatible anisotropy from initially randomly oriented ice should provide a conservative yardstick, when applied to gradual changes in stress regime.

The patterns of stress regimes revealed by the distributions of $\lambda_S$ (Figs. 6e, 7e, 8g, and 9g) indicate where along-flow variations in stress regime might be too rapid to sustain the assumption that a compatible crystallographic anisotropy had evolved. For the ISMIPBp experiments this concern is essentially focussed to the near surface peaks in $\lambda_S$ around the two locations where longitudinal deformations vanish, since the anisotropy of deeper ice will be compatible with deformation dominated by simple shear. There may be some complications with a slow cycling of the upper levels between tensile and compressive flow. Very near the surface, the $\lambda_S$ peak intervals are narrow and the shear strain rates there are very small (corresponding to transition scales of several kilometres) so that there will be no appreciable development of a shear compatible fabric. Either side of the peaks, the main tensile and compressive flow domains for $L = 20$ km (see Fig. 6f) are $\approx 5$ km long and have transition scales of $< 1$ km, which suggests that the strongly normal stress dominated upper layer will be mainly in tertiary state. Turning to the transient shear intrusions into this layer: at 100 m depth over the bump the shear transition scale is $\sim 3$ km while at 200 m depth over the depression the shear transition scale is $\sim 5$ km, suggesting that the $\lambda_S$ peaks do represent a local failure of the ESTAR flow relation's tertiary assumption. Throughout the domain the lower half of the ice column has

transition scales of $\leq 300$ m which, given the gradual variations in $\lambda_S$ and the direction of ice flow, indicates that region is in the tertiary state.

For ISMIPBp with $L = 5$ km, which displays a generally deeper band of normal stress dominated regime (Fig. 7e), the transition scales for the compressive and tensile regions are $\sim 500$ m for regions of $\approx 1$ km in extent, while the shear transition scales at 100 m depth above the bump and 200 m depth above the depression are now $\sim 10$ km and $\sim 1$ km, respectively. For most of the domain the transition rates in the lower half of the ice column are $\leq 100$ m, although this rises to nearly $1000$ m above the bedrock bump.

In the ISMIPDp case, for $L = 20$ km, the pattern of $\lambda_S$ (Fig. 8g) shows there are also transitions between simple shear dominated and vertical extension dominated deformation associated with the slippery region, with varying abruptness at different depths, with some of the contours of $\lambda_S$ in this instance almost orthogonal to the ice flow. A complication is that there is very little longitudinal deformation (Fig. 8f) occurring over the slippery region because the overall flow is controlled by the periodic sticky spot. Accordingly, there would not be any significant fabric evolution across this $\sim 4$ km region (estimated transition scales there are $> 40$ km) so that the tertiary assumption and using $E_C$ (since $\lambda_S = 0$) would be inappropriate. Once again, the low strain rates here (Figs. 8h-i) translate into very stiff ice and might make the influence of ESTAR enhancement factors relatively unimportant. A factor of 100 in $\dot{\varepsilon}_e$ changes viscosity by a factor of 21.5, whereas the maximum viscosity contrast from $E(\lambda_S)$ is 1.39. The shear strain rates are also very low, with transition scales $> 1$ km except very close to the bed over the sticky spot. Accordingly, while a compatible fabric could be expected where the large $\lambda_S$ values are shown in Fig. 6a, its presence would be due to the periodic flow, and the inability of the $\lambda_S \sim 0$ region to modify it.

For the last test, ISMIPDp with $L = 5$ km, strain-rates are once again very low, and there is no simple structure to the picture of the stress regime portrayed by $\lambda_S$ in Fig. 9g. Below mid-depth there is a periodically continuous band of shear that might favour the development of crystal anisotropy, but clearly the tertiary flow assumption of the ESTAR flow relation would not be particularly useful here.

The focus of this study was to explore the effect on the dynamic response of ice sheets of using a constitutive relation appropriate to the tertiary flow regime, i.e. sensitive to the varying proportions of simple shear and normal stresses, compared to using the standard (Glen) flow relation. Our results, particularly with respect to the differences between the Glen and ESTAR simulations, are sensitive to the choice of $E_S$ and $E_C$. Experimental evidence (Treverrow et al., 2012) suggests that the ratio $E_S/E_C = 8/3$, rather than their overall magnitude, is the dominant control in the level of enhancement $E(\lambda_S)$ and corresponding dynamic response of grounded and floating ice sheets. Here, we used values of $E_S = 8$ and $E_C = 3$, which are based on laboratory experiments of tertiary creep (Treverrow et al., 2012), and which yield values for the overall enhancement that are compatible with estimates from borehole inclination measurements (e.g., Russell-Head and Budd, 1979) and modelling studies (e.g., Wang and Warner, 1999). Nevertheless, further investigation into suitable values of $E_S$ and $E_C$ to use in numerical modelling studies of grounded and floating ice sheets is warranted. Indeed, with the implementation of the ESTAR flow relation in ISSM, it might be possible to use inverse methods to search for values of $E_S$ and $E_C$ that improved the match between modelled and observed surface velocities.

In order to examine the impact of a flow relation appropriate to ice with a compatible flow-induced anisotropic crystal fabric on simulated ice dynamics, the ISMIP-HOM and embayed ice shelf experiments were carried out assuming isothermal conditions. However, as discussed earlier, real ice sheets and ice shelves typically have cold, upper layers and strong vertical gradients in temperature, and these will often be stronger controls on vertical contrasts in deformation rates, through $A(T')$

(Eqs. (2)-(3)), than a factor of 3 to 8 produced by enhancements for tertiary flow.

## 8 Conclusions

We have investigated some consequences of incorporating the flow properties of anisotropic ice into modelling flow in ice sheets and ice shelves. Specifically, we have investigated the flow response to prolonged deformation under a constant or slowly changing stress regime and the associated development of an anisotropic crystal orientation fabric compatible with

that deformation, as represented by the empirical, scalar, tertiary constitutive relation for ice with a compatible anisotropic crystal fabric of Budd et al. (2013) – the ESTAR flow relation. Having implemented this flow relation in ISSM, we made initial studies in the context of idealised experiments: for an embayed ice shelf, and in two-dimensional models of grounded ice flow over varying topography and variable basal friction previously explored by ice flow modellers (Pattyn et al., 2007, 2008). We have demonstrated that the ESTAR flow relation is computationally efficient for large-scale ice sheet models. We have

highlighted that it produces different flow responses compared with the prevailing Glen flow relation, in regions where simple shear and normal stresses, and combinations of these, are drivers of ice flow. We have also noted some possible limitations of this empirical treatment of the tertiary flow regime, although their significance and whether there is scope for developing the empirical approach to resolve them remain to be determined. It would also be interesting to compare the ESTAR flow relation with the predictions of modelling using microstructure-controlled constitutive relations, even if the comparisons were limited

to local domains or idealised cases.

Our embayed ice shelf results have significant implications for ice sheet model simulations that rely on the Glen flow relation to simulate past, present, and future ice flow, which are used to constrain uncertainty in reconstructions and projections of sea levels. In particular, the effect of unrealistically fast thinning ice near the calving front, as simulated with the Glen flow relation, is to deform the ice shelf, which could lead to unrealistic ice shelf geometries and affect buttressing if it were to spread beyond

the "passive ice" sector (Furst et al., 2016) near the ice front.

With the implementation of the ESTAR flow relation into ISSM completed, further investigation into its capacity to replicate real-world ice sheet flow in Antarctic outlet glaciers is currently underway.

*Code availability.* The results from this work are reproducible using ISSM (from version 4.11). The current version of ISSM is available for download at https://issm.jpl.nasa.gov. The ISMIP-HOM experiments are documented in Pattyn et al. (2007).

*Competing interests.* The authors have no competing interests.

*Acknowledgements.* The authors thank the editor, Oliver Gagliardini, and each of the three reviewers for their comments that resulted in an improved manuscript. This work was supported under the Australian Research Council's Special Research Initiative for Antarctic Gateway Partnership (Project ID SR140300001), and the Australian Government's Cooperative Research Centres Programme through the Antarctic Climate and Ecosystems Cooperative Research Centre (ACE CRC). The University of Tasmania supported the visit of MM to Hobart. This research was undertaken with the assistance of resources from the National Computational Infrastructure (NCI), which is supported by the Australian Government.

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

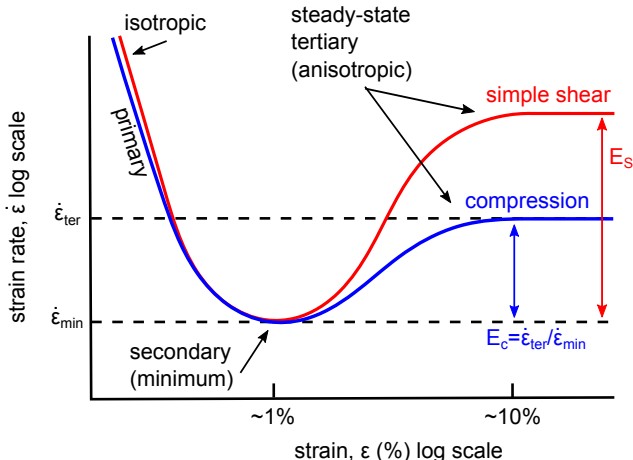

**Figure 1.** Schematic illustrating strain rate characteristics of polycrystalline ice undergoing deformation driven by single stresses as measured in laboratory experiments. The part of the curve corresponding to tertiary (steady-state) anisotropic creep is relevant to the deformation of ice masses in typical ice sheets and glaciers. The red (blue) curve illustrates the result of simple shear-alone (compression-alone) stress configurations. The ratio of the shear enhancement factor $E_S$ to the compression enhancement factor $E_C$ is approximately $8/3$ (Treverrow et al., 2012), and the enhancement due to compression-alone is approximately three times that of the secondary (minimum) creep rate.

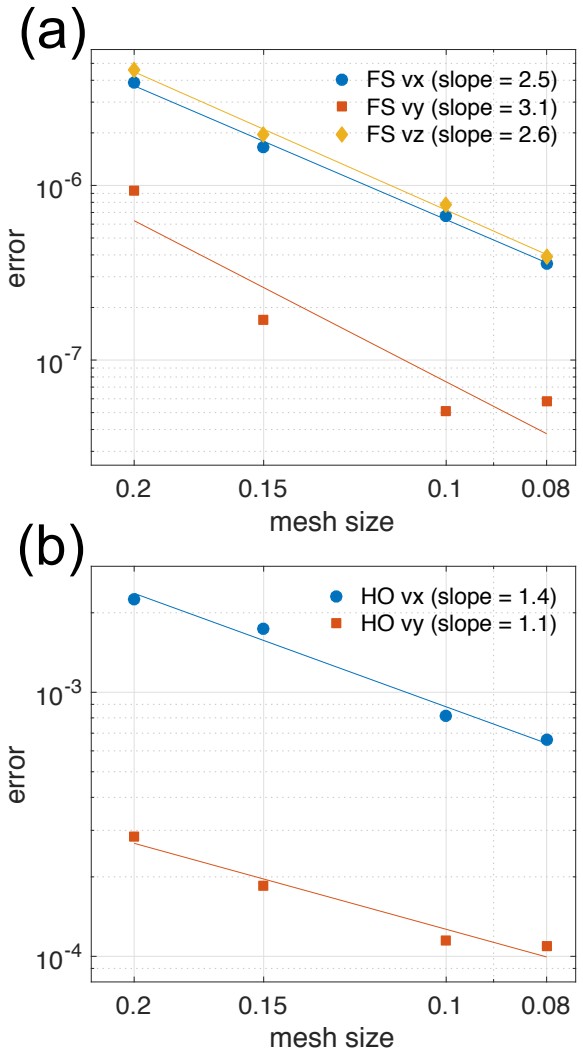

**Figure 2.** Convergence rates of the simulated (a) full-Stokes and (b) higher-order velocity fields $(v_x, v_y, v_z)$ to the analytical solutions in Eqs. (13)-(17) for increasing mesh resolutions.

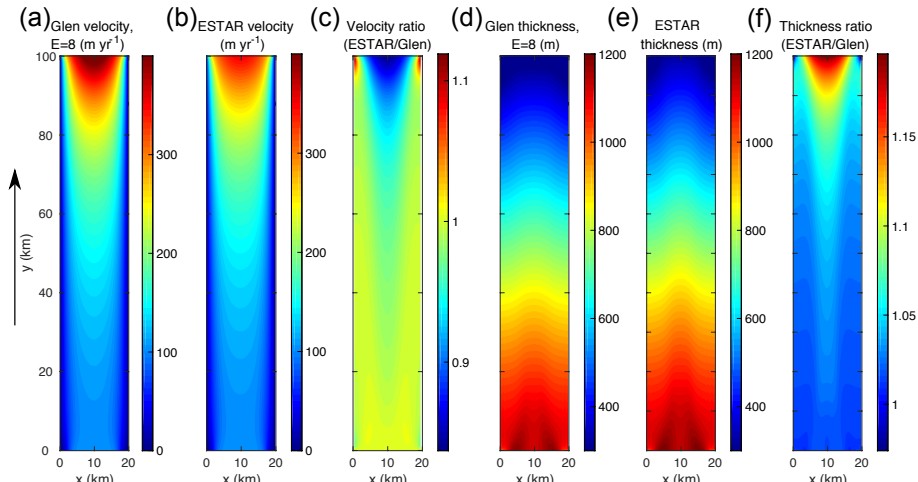

**Figure 3.** Rectangular ice shelf higher-order steady-state surface fields. **(a)** Velocity magnitude (m yr$^{-1}$) for the Glen flow relation ($E_G = 8$); **(b)** velocity magnitude (m yr$^{-1}$) for ESTAR; **(c)** ratios (i.e., ESTAR/Glen) of velocity magnitudes; **(d)** thickness (m) for the Glen flow relation ($E_G = 8$); **(e)** thickness (m) for ESTAR; and **(f)** ratios (i.e., ESTAR/Glen) thicknesses. The black arrow indicates the direction of flow.

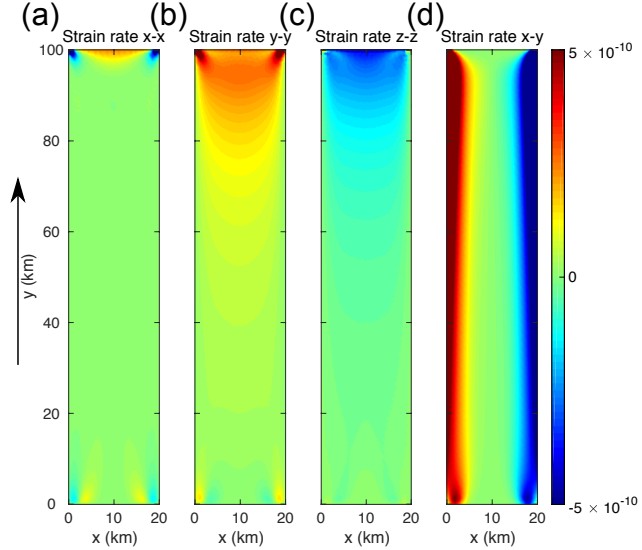

**Figure 4.** Rectangular ice shelf ESTAR higher-order steady-state surface strain rates (s$^{-1}$): **(a)** $\dot{\varepsilon}_{xx}$; **(b)** $\dot{\varepsilon}_{yy}$, **(c)** $\dot{\varepsilon}_{zz}$; and **(d)** $\dot{\varepsilon}_{xy}$.

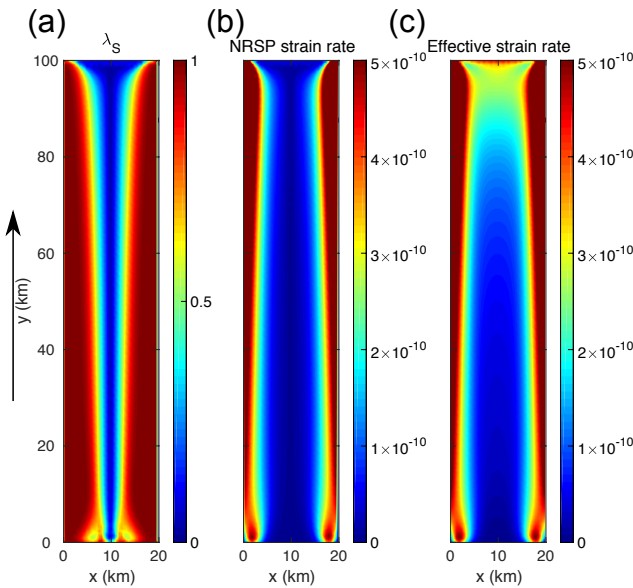

**Figure 5.** Rectangular ice shelf ESTAR higher-order steady-state surface fields. **(a)** ESTAR shear ratio $\lambda_S$; **(b)** shear strain rate resolved on the non-rotating shear plane (NRSP) $\dot{\varepsilon}'$ (s$^{-1}$); and **(c)** effective strain rate $\dot{\varepsilon}_e$ (s$^{-1}$). The black arrow indicates the direction of flow.

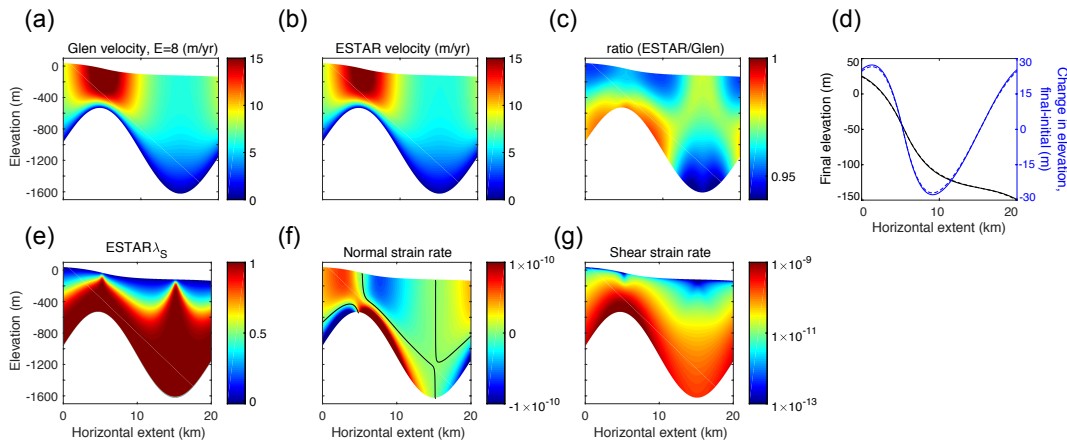

**Figure 6.** ISMIPBp full-Stokes steady-state results with horizontal extent $L = 20$ km. **(a)** Horizontal velocity $v_x$ (m yr$^{-1}$) for the Glen flow relation with $E_G = 8$; **(b)** $v_x$ (m yr$^{-1}$) for ESTAR with $E_S = 8$ and $E_C = 3$; and **(c)** ratio between the Glen and ESTAR $v_x$ fields; **(d)** steady-state surface elevation (black) and difference between steady-state and initial surface elevation (blue) for ESTAR (solid) and Glen (dashed); **(e)** ESTAR shear enhancement factor $\lambda_S$ (Eq. (6)); **(f)** ESTAR normal strain rate (i.e., $x-x$ strain rate; s$^{-1}$); and **(g)** ESTAR shear strain rate (i.e., $x-z$ strain rate; s$^{-1}$). The black contours in **(f)** correspond to the curves where $\dot{\epsilon}_{xx} = 0$. Note the log scale in **(g)**.

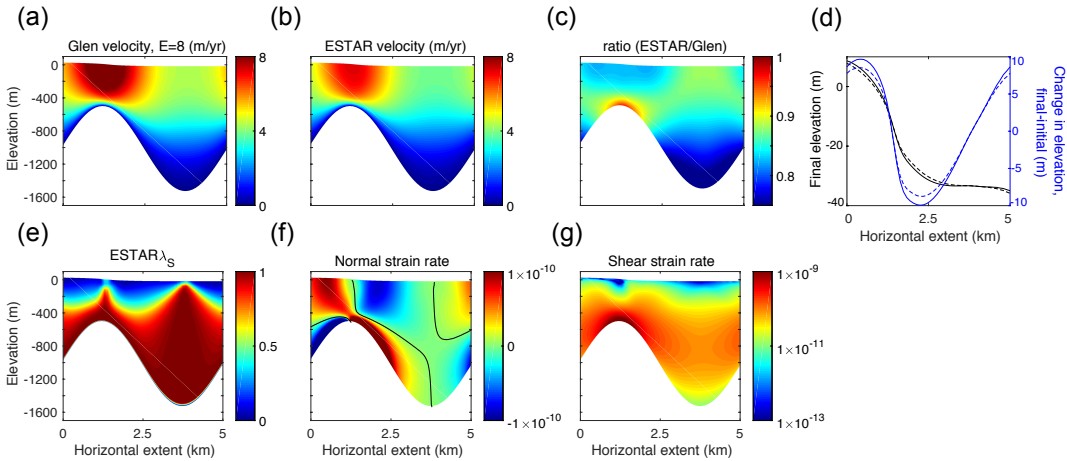

**Figure 7.** As for Fig. 6, but with $L = 5$ km.

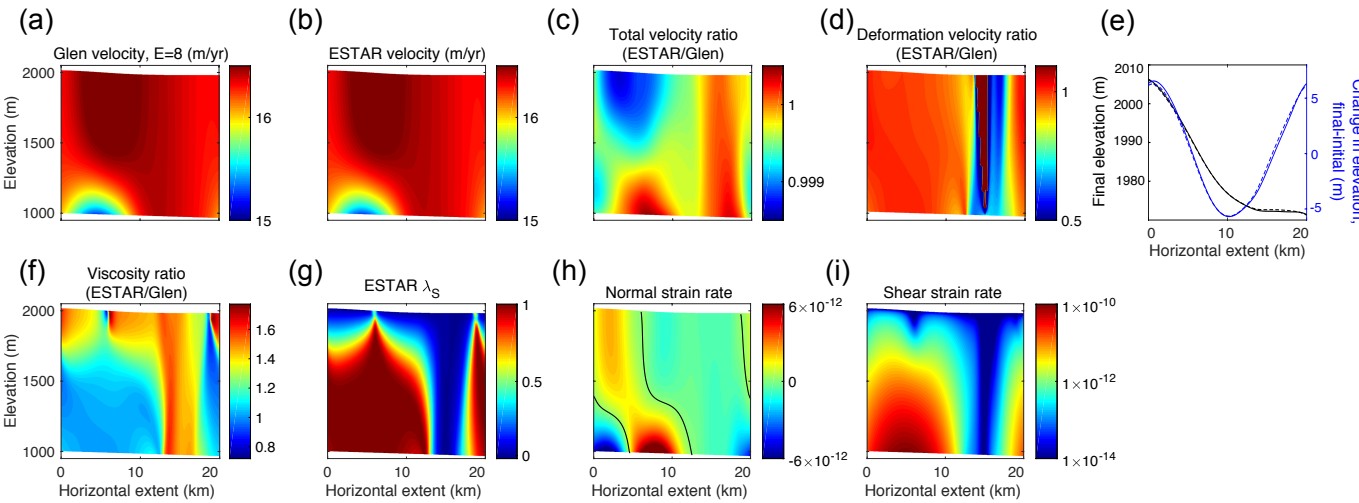

**Figure 8.** ISMIPDp full-Stokes steady-state results with horizontal extent $L = 20$ km. **(a)** Glen horizontal velocity $v_x$ (m yr$^{-1}$, with $E_G = 8$); **(b)** ESTAR $v_x$ (m yr$^{-1}$); **(c)** ratio of ESTAR/Glen $v_x$; **(d)** ratio of ESTAR/Glen deformation $v_x$; **(e)** steady-state surface elevation (black) and difference between steady-state and initial surface elevation (blue) for ESTAR (solid) and Glen (dashed); **(f)** ratio of ESTAR/Glen viscosity; **(g)** ESTAR $\lambda_S$; **(h)** ESTAR normal strain rate (s$^{-1}$); and **(i)** ESTAR shear strain rate (s$^{-1}$). The black contours in **(h)** correspond to the curves where $\dot{\varepsilon}_{xx} = 0$. Note the log scale in **(i)**.

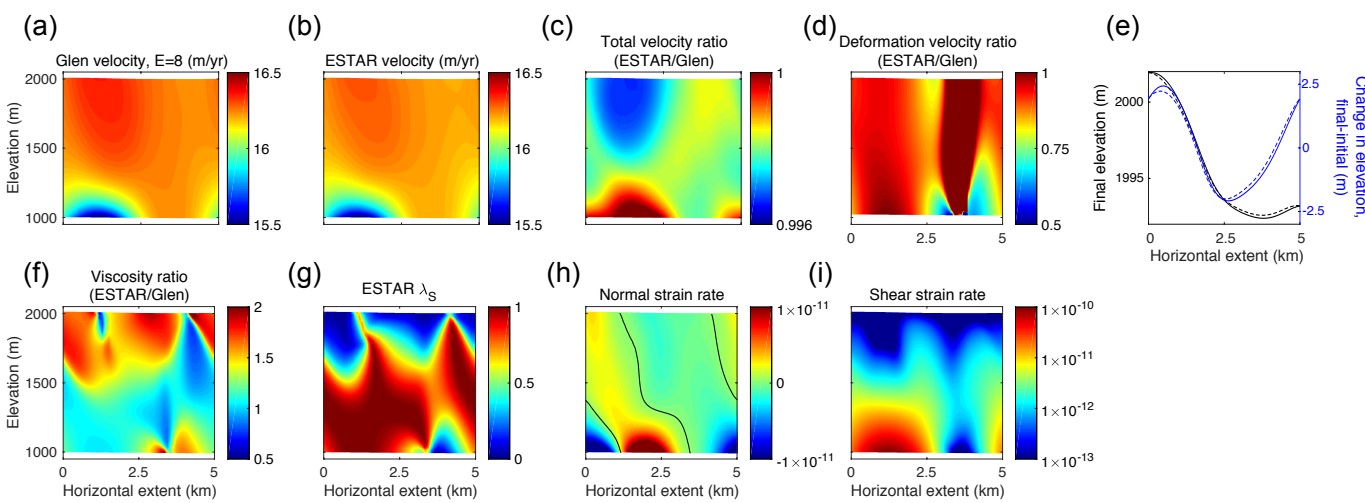

**Figure 9.** As for Fig. 8, but with $L = 5$ km.

**Table 1.** Computational times for simulations of the higher-order embayed ice shelf model (Sect. 6.1) using Glen and ESTAR flow relations. The model is simulated for 1000 years with 2 year time steps, and for a mesh of 80080 vertices over 10 vertical layers.

| CPUs | Glen walltime (s) | ESTAR walltime (s) |
|---|---|---|
| 1 | 27 568 | 32 794 |
| 4 | 8 083 | 8 521 |
| 8 | 5 164 | 5 568 |
| 16 | 3 457 | 3 639 |
| 32 | 2 721 | 2 821 |