# Peer review of "Implementing an empirical scalar constitutive relation for ice with flow-induced polycrystalline anisotropy into large-scale ice sheet models"

_The Cryosphere, 2017_

## Referee Comment (RC1) · Anonymous Referee #1 · 6 Jun 2017

The paper present the implementation of a scalar anisotropic flow law (ESTAR) into the large-scale model ISSM. The model is applied to 3 idealised configurations (embayed ice-shelf, ISMIP B and D experiments) and the results compared to the Glen flow law with a uniform enhancement factor.

My main concern with this paper is that ESTAR is presented as a physically-based alternative to represent the anisotropic rheological properties of polar ice. There is 2 problems with this presentation; first ESTAR is not an anisotropic flow relation; second, ESTAR is based from laboratory measurements for tertiary creep of polycrystalline ice and thus does not apply in large portions of the ice-sheet. I detail these two points

below.

Anisotropy: It can only be applied to properties that depend on the orientation of the material. Here the flow relation (3) is stress-configuration dependant, but is isotropic; i.e. for a given stress state, the mechanical response (the strain-rate state) does not change by rotation of the sample as, by definition of ESTAR, the fabric instantaneously adapt to the stress state, and thus to any rotation of the stress state. The dependance of the flow law to the stress state, results from the mechanical anisotropy of the ice crystal and the anisotropy of the crystal orientations (the fabric) that develop during tertiary creep; but the mechanical behaviour predicted by ESTAR is isotropic. Note that this property is not due to the scalar relationship, the CAFFE model (e.g. Seddik et al., J. Glaciol. 2008) is anisotropic as its scalar enhancement factor depends on the direction of the stress state with respect to the ice fabric, and thus the mechanical response depends on the orientation of the stress state relative to the ice sample.

Tertiary creep: In the paper tertiary creep is presented as "the predominant mode of deformation in ice sheets", under the justification that it is usually reached after few percent of deformation in laboratory creep experiments. For a detail review on the ductile deformation of ice and textures in polar ice masses one can refer to the book "creep and fracture of ice" by Schulson and Duval. Tertiary creep in laboratory experiments is explained by softening processes associated with migration recrystallization. Textures resulting from migration recrystallization are indeed stress-configuration depend and the argument that the microstructure evolves more rapidly than the flow configuration sufficiently robust to assume a scalar isotropic relationship in the context of large scale ice flow modelling. In the central part of the ice-sheets, temperature and strain energy can be too low to initiate migration recrystallization and there is no evidence that it occurs except in the warm ice near the bed (the last 100-200 meters usually). See texture measurements in all deep ice cores in the central regions of Greenland and Antarctica. There is a general agreement that in the regions of normal grain growth and rotation recrystallization (most of the ice thickness in the central regions) textures basically de-
velops as the results of lattice rotation due to intractrystalline slip; and thus reflect the deformation history. The combination of anisotropic textures and the high mechanical anisotropy of the ice mono-crystal makes the ice polycrystal highly anisotropic. This has been confirmed by laboratory experiments that have shown that the strain-rates for secondary creep of polycrystalline ice depends on the fabric and its orientation with respect to the test configuration; while the tertiary creep is independent of the initial fabric and depends only on the test configuration. All the studies presented in Sec. 2.1 as "microstructure approaches", where motivated by representing the mechanical anisotropy of polycrystalline ice that develops as a result of the development of strain-induced preferred orientations of the ice crystals. They can not be opposed or compared to the flow law presented here, as they don't represent the same physical processes.

Laboratory experiments for tertiary creep are certainly relevant for the regions where migration recrystallization occurs; i.e. in temperate glaciers, in the bottom part of the central regions of the ice-sheets and certainly in the margins where temperature and stresses are sufficiently high. These areas are potentially important for the flow dynamics of polar ice sheets. The flow law presented in this paper must be presented in this perspective, and not as an empirical approach to represent the strain-induced mechanical anisotropy that results from the development of ice fabrics as seen in all deep ice-cores. This implies to largely rewrite large parts of the abstract, introduction and second section, to discuss the validity of the assumptions of the flow law and their domain of applicability for large-scale ice flow modelling.

Concerning the rest of the paper, i.e. the implementation of the flow law in the ice sheet model and the applications I only have minor comments: ietmize

Application to the embayed ice shelf:
I think it would have been more interesting to test the model with larger transverse dimensions, i.e. to represent non-embayed ice-shelves, as I assume that simple shear

along the margin will be less important and the difference with Glen and a uniform enhancement factor for shear should increase ?

ISMIP-HOM experiments:
When discussing the results for Glen in the main text it is never clear for which value of the enhancement factor. The fact that the velocity scales with the enhencement factor is not a result, but as written in the manuscript follows from the definition of the experiment. So it would be more clear for the reader, to discuss the results for only one value of the enhencement factor or maybe better, find the value of the Glen enhancement factor that minimize the velocity difference with the tertiary creep flow law, and discuss this value and the results for this value.
The discussion in page 12 about the value of the shear fraction along the transition curve is not very clear.

---

## Referee Comment (RC2) · Anonymous Referee #2 · 4 Jul 2017

The paper presents the implementation of an isotropic rheology that accounts for the effects of ice fabric (ESTAR) and it is easy to implement in large-scale ice sheets models. The method, based on Budd 2013, estimates the local effect of ice fabric directly from the instantaneous flow field and incorporates it in a Glen flow law relation. The authors apply this method to three idealized scenarios and compare the results with those from an identical Glen flow relation where the enhancement factor has been keep uniform. They conclude that the results are different.

I find the paper clear and well written but I believe that the method presented is wrongly described and the discussion is lacking some interesting points. I give bellow a few

general comments that, I have just realized, incorporate all the specific comments I have pencilled in the manuscript.

General Comments I have had a chance to look at the other reviewer comments and I have to say that I fully agree with him/her in the main two points of his/her review. First, the authors always refer to the method as "anisotropic", it is the "A" in the acronym. That is not simply misleading, it is wrong. In an isotropic rheology the relation between stress and strain-rate is a scalar number, twice the viscosity, in a more general anisotropic rheology it is a tensor. The method presented in the paper is simply not anisotropic. Also in this point, the authors refer to Glen flow relation vs ESTAR throughout the manuscript. ESTAR as far as I can see is a Glen flow relation with a pre-exponential factor that varies spatially to account for ice fabric. A in Equation 1 is known to depend on ice fabric and a myriad of other things (not only on temperature as stated in the manuscript), the whole point of ESTAR is that it is giving a method to estimate the effect of ice fabric on A. And second, the empirical relation to extract the enhancement factor is based on laboratory experiments of tertiary flow, and it may well be that it can not reproduce the widespread observations of strain-induced anisotropy in polar ice. There is no discussion in the paper about how the proposed method explain observations or even expectations of the effect of ice fabric in polar flow. Do the vertical or horizontal variations of enhancement factor make sense according to observations of fabric or strain-rate fields?

I also miss an important point in the paper, what is the point of using ESTAR in a large-scale ice sheet model? I may be missing something here but large-scale models are capable to initialize the local depth-averaged hardness parameter in the Glen flow law using satellite observations. What is ESTAR adding? Could ESTAR inform the initialization of a large-scale model somehow? It could be applied to simulations were there is no satellite observations but then, as I suggested in my previous point, shouldn't we check how well does it do in a case where we have observations? Also, are there any significant vertical variations on enhancement factors that are captured by ESTAR but

can't be inverted by a large-scale model? In any case, the paper needs be clearer about what is ESTAR and what can it do.

Finally, I find really intriguing that the authors state that results are very similar with different aspect ratios in the experiments presented in Sections 5.1 (P9-L31) and 5.2 (P11-L25). What do they mean? In the embayed ice-shelf, I would expect the ice fabric induced by extension and shear at the margins to be different. The narrower the ice-shelf is I would expect that the overall effect of lateral shearing should be more important. In the ice flow over a bumpy bed, L is the wavelength of the bedrock undulations. I would expect the increase in basal roughness to have a strong control over the orientation of the fabric. What does it mean that aspect ratios don't affect ESTAR results? I would like to see some discussion about the results presented.

---

## Author Comment (AC1) · 10 Aug 2017

Manuscript ID: tc-2017-54

**Title: Implementing an empirical scalar tertiary anisotropic rheology (ESTAR) into large-scale ice sheet models**

Authors: Felicity Graham, Mathieu Morlighem, Roland Warner, and Adam Treverrow

**Overall response to the reviewers**

We thank the reviewers for their initial comments, which have led to a substantially improved manuscript.

The two main points highlighted by both reviewers concern the philosophy underlying ESTAR, specifically whether or not it can be regarded as a physically-based anisotropic rheology that is applicable for most of the ice sheet. We recognise that an understanding of the conceptual basis behind ESTAR, particularly how it relates to anisotropic properties of polar ice and tertiary creep, is important and deserves discussion.

However, the main aim of this paper is not to provide a comprehensive review of anisotropic flow relations, nor a justification of ESTAR or comparison of ESTAR with other anisotropic approaches (e.g., fabric based rheologies). Instead, the aim of this manuscript is to present the implementation and some initial testing of ESTAR within the context of a numerical ice sheet model. We are not introducing ESTAR for the first time: it was derived and discussed in detail in Budd et al. (2013) and a comparison of a simplified version of the generalised ESTAR from Budd et al. (2013) with some of the leading anisotropic flow relations was undertaken by Treverrow et al. (2015). A detailed reprise of Budd et al. (2013) is not feasible here, and would be inappropriate in the present context.

Nevertheless, we provide comments to the reviewer's main points below, and have amended the manuscript, where appropriate. In particular, we have expanded on the background to anisotropic flow relations (Sect. 2), modified the results section (Sect. 5), included ISMIP-HOM experiments investigating the impact of aspect ratios, as suggested. We have modified and reordered the discussion section (Sect. 6) to discuss more broadly the relevance and applicability of ESTAR, and have expanded on the conclusions (Sect. 7).

Our responses to each of the reviewers' specific comments are below, with text from each reviewer's comments presented in italics. Where necessary, we have segmented the reviewers' comments to deal with specific issues sequentially.

All line number and figure references are to the revised manuscript unless otherwise indicated.

Kind regards,

Dr Graham and coauthors

**REVIEWER 1**

**Main point**

*R1: My main concern with this paper is that ESTAR is presented as a physically-based alternative to represent the anisotropic rheological properties of polar ice. There is 2 problems with this presentation; first ESTAR is not an anisotropic flow relation; second, ESTAR is based from laboratory measurements for tertiary creep of polycrystalline ice and thus does not apply in large portions of the ice-sheet. I detail these two points below.*

**Point 1: Anisotropy**:
*R1: It can only be applied to properties that depend on the orientation of the material. Here the flow relation (3) is stress-configuration dependant, but is isotropic; i.e. for a given stress state, the*

*mechanical response (the strain-rate state) does not change by rotation of the sample as, by definition of ESTAR, the fabric instantaneously adapt to the stress state, and thus to any rotation of the stress state. The dependance of the flow law to the stress state, results from the mechanical anisotropy of the ice crystal and the anisotropy of the crystal orientations (the fabric) that develop during tertiary creep; but the mechanical behaviour predicted by ESTAR is isotropic. Note that this property is not due to the scalar relationship, the CAFFE model (e.g. Seddik et al., J. Glaciol. 2008) is anisotropic as its scalar enhancement factor depends on the direction of the stress state with respect to the ice fabric, and thus the mechanical response depends on the orientation of the stress state relative to the ice sample.*

We are certainly presenting ESTAR as a physically-based alternative to the standard Glen flow relation.

The reviewer is incorrect in stating that ESTAR is isotropic. The conceptual origin of the description of anisotropy in ESTAR is based on considerable observations from ice cores, boreholes and laboratory experiments (e.g. Jacka and Maccagnan, 1984; Dahl-Jensen and Gundestrup, 1987; Gao and Jacka, 1987; Budd and Jacka, 1989; Etheridge, 1989; Li and Jacka, 1998; Morgan et al., 1998; Wang and Warner, 1999; Treverrow et al., 2012). Observations such as these demonstrate that if the stress configuration is changed within a fixed reference frame then the compatible fabric that develops in response to that stress is correspondingly altered, i.e. specific fabric patterns are associated with specific stress configurations (note that in the following we use the term 'fabric' to describe the distribution of crystallographic $c$-axis orientations). This is valid provided that the stress configuration remains stable for a period that exceeds the time (or strain) required for the microstructure to adapt to that state of stress. Since polycrystalline anisotropy is fundamentally linked to microstructure (including fabric, but not exclusively so) the anisotropic response of an aggregate can be directly related to the present stress configuration, if that is a suitable indication of a persistent recent strain history.

Identification of the local non-rotating shear plane (normal, $n$), the simple shear stress, $\tau'$, acting on that plane and its magnitude relative to the total effective stress, $\tau_e$, summarise the computational framework for specifying the anisotropic enhancement factor, $E(\lambda_S)$, in ESTAR (Equations 4-6). The parameter $\lambda_S$ also characterises the fabric pattern, i.e., when $\lambda_S = 1$ there will be a strong single maximum, while when $\lambda_S = 0$, there will be $c$-axes concentrated around a conical surface centred on the compression axis – a small circle girdle. The anisotropy of $E(\lambda_S)$ is derived from the spatial variability in the flow field and its dependence on the deformational component of the vorticity.

The concepts of rheology and its description in a flow relation can sometimes become confused. ESTAR is the representation of the deformational response of ice with a crystal fabric anisotropy compatible with prolonged deformation under a stable stress regime – this is the 'tertiary' aspect of ESTAR. When the compatibility requirement is considered, it is clear that there is no sense in contemplating arbitrary rotations between the stress configuration and anisotropic crystal fabrics – this relates to the deformability of anisotropic ice. Regarded in this light, it seems to us that the suggestions of both reviewers that one could meaningfully consider an arbitrary rotation of spatially anisotropic material properties and the applied stresses breaks the intrinsic point of ESTAR. The point also seems semantic – the rheology of polycrystalline anisotropic ice versus an anisotropic rheological description for polycrystalline ice.

This reviewer notes correctly that the enhancement factor in the CAFFE flow relation is anisotropic since it is defined by the local stress field and its orientation relationship with respect to a polycrystalline aggregate, and rotation of one relative to the other produces a different effect. In CAFFE the explicit inclusion of fabric in defining the deformability of polycrystalline ice allows the instantaneous flow response for any arbitrary combination of stress and fabric to be calculated. CAFFE and ESTAR

are structurally similar in featuring a collinear relationship between stress and strain rate tensor components and hence each can be interpreted as an enhancement factor, defined as a function of some experimentally-based enhancement parameters and the local stress field. A key difference between the two relations is that for ESTAR fabric is not explicitly included in specification of the enhancement. By linking the enhancement directly to the relative magnitude of the components of $\sigma'$ via $E(\lambda_S)$, ESTAR is restricted to a subset of the fabric/stress combinations that can be contemplated with microstructure based approaches, such as CAFFE. Importantly, these combinations where ESTAR is applicable are those where fabric has evolved to be compatible with the local stress field. Put another way, ESTAR only allows contemplation of tertiary creep (addressed on P2L5-13; P5L5-18; P7L1-4). This is not a problem since tertiary creep is relevant to the vast majority of the most dynamically active regions of an ice sheet. In the revised manuscript, we have noted examples where this might not be the case (P5L21-P6L12):

"There are of course zones within an ice sheet where the assumption of compatible tertiary flow will not apply; however, we note that these zones will be restricted in their extent. We contend that ESTAR will apply to the vast majority of the dynamically active regions of an ice sheet, in particular those zones where creep deformation makes a significant contribution to the overall flow. Specific zones where the assumption of tertiary creep may be inappropriate can be summarised as those where fabric has not yet evolved compatibility with the flow, where there is a rapid transition in the flow configuration, or where creep deformation makes only a minor contribution to the overall dynamics.

For example, in very cold ice in a low stress setting, such as the uppermost layers of the polar ice sheets, the time required to accumulate the strain necessary to develop a compatible fabric may lead to a near-surface zone in which the assumption of tertiary creep is not valid. Since the development of anisotropic fabrics provides an indication of the existence of tertiary flow, their observation at modest depths, (e.g., $\lesssim 100 - 200$ m; Morgan et al., 1997; DiPrinzio et al., 2005; Treverrow et al., 2016) allows estimation of the maximum extent of the zone where tertiary creep is not occurring. The observation within polar firn of microdeformation processes that are necessary for the development of fabric throughout ice sheets (e.g. Kipfstuhl et al., 2009; Faria et al., 2014) suggests that it may be appropriate to even further restrict the extent of the near-surface zone for which the assumption of tertiary creep is not valid. Additionally, the nonlinear nature of polycrystalline ice rheology leads to very high viscosities in low temperature and stress environments, so that incorrectly estimating deformation rates due to the assumption of tertiary flow in such regions may be of limited importance to simulations of ice sheet evolution.

Regions where rapid transitions in dynamic conditions can lead to abrupt changes in the pattern of applied stresses and a potential breakdown in tertiary flow compatibility include ice shelf grounding zones and other locations where basal traction is lost or abruptly changes, e.g., where ice flows over a subglacial lake, or with the onset of basal sliding in ice streams. The convergence zones where tributary glaciers or ice streams merge with a larger flow unit at a high angle may also lead to a transition in dynamic conditions that is problematic for the assumption of tertiary compatibility. Of course the more highly dynamic the evolving flow regime, the more rapidly a new compatible anisotropy will be established, so that the spatial interval where the flow relations are inapplicable may be limited. While there is little guidance on how to extend empirical flow relations to parametrise ice rheology in these transition regions, we note that similar difficulties exist for a Glen-type flow relation, which unlike ESTAR does not have the benefit of being able to correctly describe anisotropic enhancement throughout the remainder

of the ice sheet."

That tertiary creep is relevant to the majority of the ice sheet is crucial and is dealt with further in responding to **Point 2**.

**Point 2: Tertiary creep**

*R1: In the paper tertiary creep is presented as "the predominant mode of deformation in ice sheets", under the justification that it is usually reached after few percent of deformation in laboratory creep experiments. For a detail review on the ductile deformation of ice and textures in polar ice masses one can refer to the book "creep and fracture of ice" by Schulson and Duval. Tertiary creep in laboratory experiments is explained by softening processes associated with migration recrystallization. Textures resulting from migration recrystallization are indeed stress-configuration depend and the argument that the microstructure evolves more rapidly than the flow configuration sufficiently robust to assume a scalar isotropic relationship in the context of large scale ice flow modelling. In the central part of the ice-sheets, temperature and strain energy can be too low to initiate migration recrystallization and there is no evidence that it occurs except in the warm ice near the bed (the last 100-200 meters usually). See texture measurements in all deep ice cores in the central regions of Greenland and Antarctica. There is a general agreement that in the regions of normal grain growth and rotation recrystallization (most of the ice thickness in the central regions) basically develops as the results of lattice rotation due to intractrystalline slip; and thus reflect the deformation history.*

The suggestion being made by the reviewer here seems to be that for laboratory experiments conducted at relatively high temperatures, tertiary creep is heavily influenced by migration recrystallisation, and that the results of such experiments are not representative of polar ice dynamics. The extension of this argument is that at lower temperatures boundary migration (a process which allows microstructural recovery during deformation) is less effective and does not contribute to fabric development. It seems this line of reasoning is then used to discount our expectation that appropriate fabrics, and perhaps tertiary creep conditions will be found in polar ice masses.

Following Schulson and Duval (2009), the reviewer states that fabric development at low temperatures is largely attributed to rotation recrystallisation – a deformation process, without the accompanying recovery afforded by boundary migration. Consequently for different polycrystalline aggregates, deformed under identical conditions of stress, it is suggested that the resultant fabric patterns will vary as a function of temperature due to the activity or otherwise of migration recrystallisation. As per Schulson and Duval (2009) the temperature at which migration recrystallisation is claimed to be sufficiently depressed, so that it no longer contributes significantly to fabric development, is $\sim -15°$C. Differences in the fabrics that are supposed to form in uniaxial compression at various temperatures are provided by Schulson and Duval (2009) as evidence to support this view. The small circle girdle (or cone-type) fabrics that form at higher temperatures are claimed to give way to single maximum (or cluster-type) fabrics at lower temperatures and stresses. The suggested implication of this for ESTAR is that the enhancement parameter derived for uniaxial compression from laboratory experiments is not valid for ice sheets. There are two problems with this argument and these appear to relate directly to the reviewer's concerns about how widespread tertiary creep is within an ice sheet. These issues are addressed in detail below, following a summary reiterating our view, before returning to the rest of the reviewer's comments.

More broadly, as described in Budd et al. (2013), tertiary creep describes a state of deformation where the microstructure within a polycrystalline aggregate has evolved to be compatible with the stress field, and this results in the development of polycrystalline anisotropy. A conspicuous feature of

tertiary creep is the development of a fabric and to an extent one can remain agnostic about the actual microdeformation and recovery mechanisms that contribute to the development of this state. The anisotropic nature of the macroscopic response, and its spatial variability, is of primary importance to modelling large-scale ice sheet dynamics, not the specific mechanisms of microdeformation.

In short, we consider tertiary creep to be far more widespread than reviewer 1 claims it to be. Of course there are zones within the ice sheet where tertiary creep is not relevant and the manuscript has been altered to more clearly highlight these (e.g., P5L21-P6L12 – as quoted directly in our response to **Point 1** above).

**Comment to the reviewer on fabric development and the extent of tertiary creep in ice sheets**

First we discuss the temperature range over which small circle girdle fabrics occur in response to uniaxial compression and then we address the issue of whether or not a single maximum fabrics can be associated with uniaxial compression.

**Small circle girdle/cone-type fabrics**

From laboratory experiments there is considerable evidence (e.g. Jacka and Maccagnan, 1984; Jacka, 1984; Gao and Jacka, 1987; Jacka and Li, 2000; Treverrow et al., 2012) supporting the formation of small circle girdle fabrics in response to uniaxial compression. That this occurs is not in question here, rather it is the issue of to what extent similar patterns have been observed in the polar ice sheets.

Law Dome in East Antarctica provides a good example of an approximately radial dome whose dynamics are mostly isolated from the remainder of the Antarctic Ice Sheet. Observations from the A001 core drilled at Law Dome summit, where the ice thickness is $\sim 1200\,\mathrm{m}$ show a distinct small circle girdle fabric at a depth of 318 m (see Fig. 3a, Budd and Jacka, 1989). At this depth uniaxial compression is expected to be dominant and the in situ temperature is $\sim -22°\mathrm{C}$. We highlight this observation since it demonstrates that small circle girdle fabrics can form in response to uniaxial compression at temperatures below the $\sim -15°\mathrm{C}$ threshold for the activity for migration recrystallisation that is suggested by Schulson and Duval (2009). Small circle girdle fabrics have been recorded at many other locations, e.g. cores drilled at Siple Dome (DiPrinzio et al., 2005), Byrd (Gow and Williamson, 1976), Dye 3, Greenland (Herron et al., 1985) and the Amery (Budd, 1972) and Ross ice shelves (Gow, 1963). Importantly, some of these observations correspond to locations where the in situ temperatures were even lower than the $\sim -22°\mathrm{C}$ described for the Law Dome, A001 example.

Perhaps one reason why small circle girdle fabrics are considered by some to be the preserve of uniaxial compression laboratory experiments conducted at high temperatures and/or stresses is that these fabrics are not frequently identified in ice cores since ideal dome summits are uncommon, i.e., the vertical shortening is typically associated with different horizontal strain rate components in radial directions, particularly at depth. Even at sites that provide a good approximation of a dome surface, the inevitable depth-evolution of the flow configuration leads to correspondingly different fabrics. Indeed, at ice divides a nearly 2D plane-strain flow can be expected.

Under conditions of plane-strain a restricted form of the cone-type fabric pattern is encountered. These are 2-pole fabrics where the individual maxima are aligned in the direction of horizontal extension and separated by the small circle girdle cone angle (e.g. Gow, 1963; Wakahama, 1974). In laboratory ice deformation experiments, similar levels of flow enhancement are observed for plane strain and unconfined uniaxial compression (e.g. Wilson, 1982; Wilson and Peternell, 2012; Budd et al., 2013).

For ESTAR this means $E_C$ has a constant value for both plane strain and unconfined compression. Assuming this equality holds for any combination of normal deformations allows the Budd et al. (2013) flow relation (ESTAR) to be generalised via specification of the $\zeta$-parameter.

**Single maximum/cluster-type fabrics**

As noted above, strongly clustered vertical single maximum fabrics are routinely observed at depth in polar ice sheets. Contrary to what we interpret as the opinion of reviewer 1, we do not suppose that these fabrics form in response to uniaxial compression at temperatures where the deformation regime is dominated by rotation recrystallisation and recovery by migration recrystallisation is negligible. Where strong single maximum fabrics have been observed in polar ice sheets these can be explained by the contribution of simple shear to the local dynamics – this includes ice-coring sites in central regions of Antarctica and Greenland (e.g. Gow and Williamson, 1976; Russell-Head and Budd, 1979; Azuma and Higashi, 1984; Tison et al., 1994; Gow et al., 1997; Morgan et al., 1997, 1998; Thorsteinsson et al., 1997; Azuma et al., 1999; DiPrinzio et al., 2005; Durand et al., 2007; Montagnat et al., 2012, 2014; Weikusat et al., 2016).

A primary concern when selecting a site to drill an ice core for a palaeoclimate record is that as far as possible it should be dynamically quiescent. Despite this aim it is clear that considerable shear can exist well above bedrock at the dome summit or ridge divide locations chosen for palaeoclimate studies. At drilling sites where there are measurements of borehole inclination to accompany the corresponding fabric data it is apparent that the development of distinct single maxima fabrics is associated with increasing levels of a simple shear deformation in the overall flow regime at depth (e.g. Dahl-Jensen and Gundestrup, 1987; Etheridge, 1989; Morgan et al., 1998; Jansen et al., 2017). No summit/divide is ideal, nor are they necessarily stationary over glacial cycles. Combined, these factors increase the likelihood of encountering single maximum fabrics (and simple shear) at depth at a palaeoclimate coring site. Even at small distances (1-2 ice thicknesses) from an ideal summit, simple shear is present at modest depths and makes a significant contribution to the overall deformation.

In Greenland the GRIP-NGRIP-NEEM ice cores were drilled along a $\sim 700\,\mathrm{km}$ section of flow line. The coring sites are located nearly equidistant along the flow line, with GRIP at the highest elevation and NEEM furthest along the flow line. For the GRIP, NGRIP and NEEM sites Montagnat et al. (2014) have estimated the depth at which the vertical and simple shear strain rates are equal. Their estimates are 79%, 73% and 65% of the total ice thickness, respectively. As noted by Montagnat et al. (2014), these estimates do not make any allowance for the effect of mechanical anisotropy on enhancing strain rates. It is reasonable to expect that doing so would lead to a proportionally greater increase in the shear strain rates relative to the vertical strain rates. This would shift estimates of the depth where the simple shear and vertical strain rates are equal to lower values, i.e. closer to the ice sheet surface. Results from Law Dome provide some interesting context for these estimates since the simple shear strain rates have been quantified by repeat measurements of borehole inclination at several locations and the vertical strain rate can be estimated by similar methods to those employed by Montagnat et al. (2014). At the Law Dome, Dome Summit South (DSS) site an ice core was drilled $4.7\,\mathrm{km}$ ($< 4$ ice thicknesses) SSW of the dome summit. Here the simple shear and vertical strain rates are equivalent at $\sim\!1/2$ the total ice thickness (Morgan et al., 1998; Treverrow et al., 2015), demonstrating that considerable simple shear can develop throughout the depth profile very close to a dome summit. In accord with the experimental observations of Budd et al. (2013), observations from ice cores reveal there is a progression towards increasingly strong single-maximum fabrics and relatively higher strain rates as the proportion of simple shear within the deformation regime increases (even

when the effect of temperature on flow rates is considered, e.g. Etheridge, 1989; Wang and Warner, 1999; Morgan et al., 1997, 1998). These observations are part of the fundamental basis of ESTAR.

Laboratory experiments also demonstrate the formation single maximum fabrics in simple shear (e.g. Bouchez and Duval, 1982; Li and Jacka, 1998; Li et al., 2000; Wilson and Sim, 2002; Budd et al., 2013; Wilson and Peternell, 2012). As described in Schulson and Duval (2009) a two-maxima fabric forms initially during simple shear experiments. The dominant maxima is normal to a plane referred to as the permanent (Bouchez and Duval, 1982) or non-rotating (Budd et al., 2013) shear plane. With increasing strain the second maxima rotates towards the dominant maxima and vanishes (e.g. Bouchez and Duval, 1982).

The only experiments we are aware of which demonstrate that a single maximum fabric can form in response to uniaxial compression have been conducted at very high deviatoric stresses in conjunction with a high confining pressure. Prior et al. (2015) present a diffuse single maximum fabric for a sample deformed in uniaxial compression to a strain of 0.37 with a deviatoric stress of 15 MPa at $-43°$C where the confining pressure was 50 MPa. Similar experiments by Qi et al. (2017) at $-10°$C with 10 MPa confining pressure show that as the deviatoric stress increases from 1.30 MPa to 4.31 MPa the resultant fabric undergoes a transition from a small circle girdle to a diffuse single maximum. This behaviour is attributed to the increased activity of rotation recrystallisation at higher stresses. Importantly, these experiments do not support for the concept of a single maximum forming in response to compression within an ice sheet, since they clearly demonstrate that a small circle girdle is expected for deviatoric normal stresses $\leq 1.30$ MPa.

To be clear, we are not suggesting that temperature and stress have no influence on the relative activity of microdeformation and recovery processes occurring within polar ice sheets and the formation of fabric. A detailed comparison of the fabrics encountered in Antarctic and Greenland ice cores and laboratory experiments is beyond the scope of the present response. However, it is worth noting that factors other than stress (both magnitude and configuration) can influence the development of fabric. For example, the presence of soluble and insoluble impurities is related to fluctuations in fabric and grain size over small spatial scales that are observed in ice cores (e.g. Morgan et al., 1997; DiPrinzio et al., 2005).

The series of uniaxial compression experiments of Jacka and Li (2000) provide some insight into the effects of temperature and stress magnitude on the rate at which fabric and strain rates evolve as a function of accumulated strain. Their results suggest that for temperatures below $-15°$C fabrics do not evolve into the distinct small circle girdles that are observed at higher temperatures. For accumulated strains of $\sim 10\%$ (or less) the observed fabrics are weakly anisotropic, displaying a minimal degree evolution from the initially isotropic distribution of $c$-axis orientations. Unfortunately these fabrics do not provide strong support for the development of either a small circle girdle or single maximum fabric.

Jacka and Li (2000) conducted a single experiment at $-45\,°$C, that unlike other experiments in the series, was conducted in two stages to allow higher strains to be accumulated (e.g. $> 20\%$, Treverrow et al., 2012). This experiment provides some support for reduced rates of fabric and strain rate evolution at low temperatures and/or stresses. This experiment was incomplete when Jacka and Li (2000) was published, but nevertheless was showing signs of a transition from secondary to tertiary creep (E>1). At the end of the first stage the sample was machined back to a cylindrical shape prior to recommencing deformation. The experiment was terminated some considerable time after the initial results were published. Analysis of the sample (T.H. Jacka pers. comm) demonstrated that while a weak small circle girdle fabric did eventually form with continued deformation, the strain required for

this at $-45\,°C$ was higher than the 8-10% observed in the experiments at higher temperatures and stresses.

That lower rates of fabric evolution may be expected at lower temperatures (or stresses) is consistent with the detailed assessment of dynamic recrystallisation within polar ice presented by Faria et al. (2014). They describe how multiple microdeformation and recovery processes are active during microstructural evolution, even in the very cold ice ($T < -20\,°C$) present in the upper regions of the polar ice sheets. Importantly, Faria et al. (2014) note that the relative contribution of these processes to microstructural evolution can vary according to temperature and strain rate.

Since small circle girdle fabrics can form due to uniaxial compression in ice sheets – even if they are not widespread – then allowances must be made for the activity of boundary migration processes at lower temperatures than the suggested threshold values of $\sim -10°C$ (Duval and Castelnau, 1995) or $\sim -15°C$ (Schulson and Duval, 2009). Observations from polar firn and ice cores, e.g., Kipfstuhl et al. (2009); Weikusat et al. (2009b,a) reveal microstructural features which clearly demonstrate that internal strain energy is sufficient to allow dynamic recrystallisation processes, including strain-induced boundary migration, to contribute to microstructural evolution at temperatures below $-20\,°C$. Modelling of the microstructural evolution of firn (Steinbach et al., 2016) demonstrates the crucial role of air bubbles in the development of the strain localisation and strain energies that are sufficient to drive dynamic recrystallisation at low temperatures in firn.

The point of the preceding discussion is to demonstrate that tertiary creep and the development of polycrystalline anisotropy is associated with the formation of a statistically steady state microstructure (including fabric) that persists under stable conditions of temperature and stress. Kipfstuhl et al. (2009) and Faria et al. (2014) demonstrate that tertiary creep cannot be simply defined according to activity (or otherwise) boundary migration. Consequently it is incorrect to invoke this single specific microdeformation process as the determinant of fabric variations observed with depth (or temperature) in ice sheets. Kipfstuhl et al. (2009) demonstrate that strain-induced boundary migration is active below the threshold temperatures proposed by Schulson and Duval (2009).

Regarding the issue of just how widespread tertiary creep is within polar ice sheets, Faria et al. (2014) describe (with specific reference to the EDML site) how the densification of firn, driven by a reduction in pore space, must be accommodated by ice deformation. Total accumulated strains within firn are high enough to support the conclusion that tertiary creep is occurring there. Faria et al. (2014) describe how the occurrence of tertiary creep, without the development of a corresponding pattern of preferred $c$-axis orientations is due to the complex geometry of the pore space in the two-phase ice-air system which produces localised heterogeneity in the stresses driving deformation. Thus, any localised development of preferred $c$-axis orientations would be masked by the spatial variability in local stress and strain.

There are some counter-intuitive aspects to the argument for single maximum fabrics forming in response to compression. Firstly, for deformation largely accommodated by basal slip, the effective viscosity will continually increase if a single maximum evolves in response to compression. Consequently high levels of deformation would have to be accommodated by non-basal processes. Secondly, if a single maximum forms in compression due to the inactivity of migration recrystallisation, then it is hard to see why single maximum fabrics become stronger with increasing depth in an ice sheet where the ice is is also warmer. If a single maximum did form in an ice sheet due to compression, as the ice becomes warmer we should see a transition to a cone-type fabric, since the activity of migration recrystallisation would be increased. The problem is that we don't see such a progression in fabrics.

The multiple maxima fabrics observed in zones closest to bed in an ice sheet don't provide the answer

here, since they are not the same as the cone-type small circle girdle fabrics associated with uniaxial compression. Multiple maxima fabrics form due to the high activity of migration recrystallisation in temperate ice. In polar ice sheets this often corresponds to warm ice near the bed that has undergone a stress relaxation due to the effect of underlying bedrock topography on disrupting the large-scale flow. The high levels of strain energy in relatively warm ice can drive migration recrystallisation. The existence of a topographic disturbance to the flow can be deduced from horizontal velocity profiles obtained from borehole inclination measurements. The inflection in these profiles close to the bed is consistent with boundary layer flows where there is an adverse pressure gradient, i.e. the reduction in velocity close to the stationary surface (the bed) exceeds expectations associated with either a neutral or favourable pressure gradient.

We now return to the latter part of the reviewer's comment about textures and tertiary creep, which we repeat for convenience:

*R1: Textures resulting from migration recrystallization are indeed stress-configuration depend and the argument that the microstructure evolves more rapidly than the flow configuration sufficiently robust to assume a scalar isotropic relationship in the context of large scale ice flow modelling. In the central part of the ice-sheets, temperature and strain energy can be too low to initiate migration recrystallization and there is no evidence that it occurs except in the warm ice near the bed (the last 100-200 meters usually). See texture measurements in all deep ice cores in the central regions of Greenland and Antarctica. There is a general agreement that in the regions of normal grain growth and rotation recrystallization (most of the ice thickness in the central regions) basically develops as the results of lattice rotation due to intractrystalline slip; and thus reflect the deformation history.*

We agree with the reviewer that all development of anisotropic fabric reflects the deformation history, more particularly, the recent deformation history. It is certainly true that within the lowest $100 - 200\,\mathrm{m}$ warm ice, high levels of strain energy and stress relaxation due to the effect of underlying bedrock topography on disrupting the large-scale flow can lead to significant migration recrystallisation and formation of multiple-maxima fabrics. The topographic disturbance to the flow can be deduced from horizontal velocity profiles obtained from borehole inclination measurements. Such profiles are consistent with flows where an adverse pressure gradient exists. However, as discussed above, it is incorrect to suggest that there is 'general agreement' on a depth-progression of specific zones of normal grain growth and rotation recrystallisation etc throughout an ice sheet. A more up-to-date review on microdeformation in ice sheets is provided by Faria et al. (2014). The observations of Kipfstuhl et al. (2009); Weikusat et al. (2009a,b) demonstrate that strain energy sufficient to drive strain-induced boundary migration occurs throughout ice sheets, including firn layers. This view is also supported by the numerical simulations of Steinbach et al. (2016).

*R1: The combination of anisotropic textures and the high mechanical anisotropy of the ice mono-crystal makes the ice polycrystal highly anisotropic. This has been confirmed by laboratory experiments that have shown that the strain-rates for secondary creep of polycrystalline ice depends on the fabric and its orientation with respect to the test configuration; while the tertiary creep is independent of the initial fabric and depends only on the test configuration.*

Yes, this final point is precisely the basic tenet of our theory, but only because the stress configuration has been applied for long enough to achieve the tertiary state. Tertiary creep of polycrystalline ice is dependent upon the stress configuration and whether or not it is occurring is not simply a matter of the activity of migration recrystallisation.

*R1: All the studies presented in Sec. 2.1 as 'microstructure approaches', where motivated by representing the mechanical anisotropy of polycrystalline ice that develops as a result of the development of strain-induced preferred orientations of the ice crystals. They can not be opposed or compared to the flow law presented here, as they don't represent the same physical processes.*

We disagree with the reviewer. What all flow relations aim to do is represent the bulk deformation of a polycrystalline aggregate. Microstructure based flow relations, where fabric is an input, can be applied equally to predictions of secondary or tertiary creep, or by extension to any arbitrary combination of stress and fabric. ESTAR only claims to represent tertiary creep – a subset of the cases that can be compared with microstructure based flow relations. Consequently comparisons with microstructure based approaches are entirely valid provided that the appropriate combinations of fabric and stress/deformations are considered. Laboratory experiments and ice cores for which corresponding strain rate data are available from repeat measurements of the borehole inclination are valuable sources of data that can be applied in such comparisons. This is particularly true for laboratory measurements of tertiary creep since the applied stresses, strain rates and microstructure data are all available.

*R1: Laboratory experiments for tertiary creep are certainly relevant for the regions where migration recrystallization occurs; i.e. in temperate glaciers, in the bottom part of the central regions of the ice-sheets and certainly in the margins where temperature and stresses are sufficiently high. These areas are potentially important for the flow dynamics of polar ice sheets. The flow law presented in this paper must be presented in this perspective, and not as an empirical approach to represent the strain-induced mechanical anisotropy that results from the development of ice fabrics as seen in all deep ice-cores. This implies to largely rewrite large parts of the abstract, introduction and second section, to discuss the validity of the assumptions of the flow law and their domain of applicability for large-scale ice flow modelling.*

Based on the preceding discussion, we do contend that ESTAR is a legitimate '...empirical approach to represent the strain-induced mechanical anisotropy that results from the development of ice fabric'. We have slightly modified the manuscript to more clearly highlight the 'domain of applicability' of ESTAR (P4L31-P5L18):

"A second approach comprising experimental and observational approaches (Li et al., 1996; Wang et al., 2002a,b), modelling (Wang and Warner, 1998, 1999; Hulbe et al., 2003; Wang et al., 2003, 2004; Breuer et al., 2006; Wang et al., 2012), and theoretical studies (Warner et al., 1999), has focussed on the development and assessment of an anisotropic flow relation for polycrystalline ice in which the nature of the crystal fabric and the magnitude of strain rate enhancement, $E$, are both regarded as determined by the stress regime. This assumption is supported by experimental observations for pure polycrystalline ice, which demonstrate that an accumulated strain of $\sim 10\%$ is required for the microstructure to evolve to a state that is compatible with the flow configuration, irrespective of its initial condition (Jacka and Maccagnan, 1984; Gao and Jacka, 1987; Li and Jacka, 1998; Treverrow et al., 2012). Specifically, this approach regards the fabric and the enhancement in tertiary flow as determined by the relative proportions of the simple shear and normal deviatoric stresses. For such flow relations, it is typically assumed that the spatial variation in dynamic conditions (e.g., flow configuration and temperature) only occur gradually in an ice sheet, so that the microstructure evolves to maintain compatibility with these conditions. Through most of an ice sheet we expect that the rate of microstructural evolution generally exceeds the rate at which the flow configuration varies, and that the distances travelled by a parcel of ice during the time taken to develop a compatible fabric are typically

small compared to the relevant ice sheet spatial scales.

The anisotropic flow relation proposed by Budd et al. (2013) represents a continuation of this strand. They found that a scalar anisotropic flow relation, i.e., one maintaining the collinear relationship between the components of $\vec{\dot{\varepsilon}}$ and $\vec{\sigma}'$ ($\tau_e$ is a scalar function of the second invariant of $\vec{\sigma}'$) provides a good fit to laboratory data from combined compression and shear experiments. Such a scalar anisotropic rheology also simplifies the requirements for implementation within ice sheet models that are already compatible with the (scalar) Glen rheological description. Budd et al. (2013) proposed what we term ESTAR as a suitable candidate scalar anisotropic rheology generalised to arbitrary stress configurations (i.e., not restricted in its application to the limited set of experimental stress configurations described in Li et al. (1996) and Budd et al. (2013))."

Rewriting 'large parts of the abstract, introduction and second section' is not required. It is important to bear in mind that this is a paper about the implementation of a flow relation, not its formulation (P3L12-17).

**Point 3**:

*Application to the embayed ice shelf*
*R1: I think it would have been more interesting to test the model with larger transverse dimensions, i.e. to represent non-embayed ice-shelves, as I assume that simple shear along the margin will be less important and the difference with Glen and a uniform enhancement factor for shear should increase?*

As mentioned in the manuscript (P11L17-22) we have carried out the embayed ice shelf experiment for the cases where the transverse dimensions are $L \in [20, 60, 100]$ km. Figures 1-3 below show the ESTAR to Glen velocity ratio, ESTAR to Glen thickness ratio, and $\lambda_S$ for each of the three cases. Note that the figures are plotted to approximately preserve their aspect ratio. It is clear from these figures that as the real aspect ratio increases, the ice shelves become flatter, so the proportion of the ice shelf that is shear dominated does not change markedly. Specifically, approximately 60% of the $L = 20$ km ice shelf is shear dominated (i.e., $\lambda_S > 0.5$), 63% of the $L = 60$ km ice shelf is shear dominated, and 67% of the $L = 100$ km ice shelf is shear dominated. This is an important result, demonstrating that, contrary to the reviewer's expectations, changing the aspect ratio does not result in an embayed ice shelf that approaches the unembayed situation (i.e., with free-slip at the side walls).

Furthermore, regardless of aspect ratio the pattern and magnitude of the velocity ratios is similar in all cases, the maximum difference being approximately 15% near the ice-ocean front where normal stresses dominate.

Hence, given the similarities in the influence of the rheology on the dynamics between the aspect ratios, we prefer to focus our discussion in the manuscript on only one transverse dimension, $L = 20$ km. We have updated the manuscript to be more explicit about the consequences of increasing the aspect ratios, as per P11L17-22:

"The experiment was carried out for model domains with transverse spans $x \in [0, L]$, for $L = 20$, 60, and 100 km and along-flow dimension $y \in [0, 100]$ km. The initial ice thickness decreases uniformly from 1000 m at the grounded zone to 300, 600, and 850 m at the ice front for the $L = 20$, 60, and 100 km cases, respectively. The main features of the anisotropic effects are similar regardless of aspect ratio. This is principally because wider embayed ice shelves are flatter so that the influence of simple shear stresses on the dynamics is not particularly sensitive to aspect ratio. Accordingly, we focus our discussion on one transverse length scale: $L = 20$ km."

[Figure]

Figure 1: HO steady-state surface fields for the embayed ice shelf with transverse dimension $L = 20$ km. From left: ratio of ESTAR/Glen velocity ($E_G = 8$); ESTAR $\lambda_S$; Glen thickness ($E_G = 8$; m); ESTAR thickness (m); ratio of ESTAR/Glen thickness.

[Figure]

Figure 2: HO steady-state surface fields for the embayed ice shelf with transverse dimension $L = 60$ km. From left: ratio of ESTAR/Glen velocity ($E_G = 8$); ESTAR $\lambda_S$; Glen thickness ($E_G = 8$; m); ESTAR thickness (m); ratio of ESTAR/Glen thickness.

[Figure]

Figure 3: HO steady-state surface fields for the embayed ice shelf with transverse dimension $L = 100$ km. From left: ratio of ESTAR/Glen velocity ($E_G = 8$); ESTAR $\lambda_S$; Glen thickness ($E_G = 8$; m); ESTAR thickness (m); ratio of ESTAR/Glen thickness.

**Point 4**:

*ISMIP-HOM experiments*

*R1: When discussing the results for Glen in the main text it is never clear for which value of the enhancement factor. The fact that the velocity scales with the enhancement factor is not a result, but as written in the manuscript follows from the definition of the experiment. So it would be more clear for the reader, to discuss the results for only one value of the enhancement factor or maybe better, find the value of the Glen enhancement factor that minimize the velocity difference with the tertiary creep flow law, and discuss this value and the results for this value.*

The ISMIP-HOM experiments are scenarios in which the bed-parallel shear is the main driver of the flow. Hence, it emerges as appropriate that we choose a value for the Glen enhancement factor equal to the shear enhancement factor, i.e., $E_G = E_S = 8$. Accordingly, we have adopted the reviewer's first suggestion and have deleted the paragraph from the manuscript that compares the results for different values of $E_G$. Since in response to Reviewer 2 we have introduced additional experiments (see details later), and as an interested observer would recognise the overall scaling properties, we have not considered it useful to take up the alternative suggestion of fine-tuning $E_G$ for each experiment. On P13L23-25 of the manuscript for ISMIPB, we clarify that our discussion of the results concentrates on the value $E_G = 8$, which is the most appropriate overall choice for this experiment:

> "In what follows, we consider the case when the Glen enhancement factor is equal to the ESTAR shear enhancement factor, i.e., $E_G = E_S = 8$. This is the most relevant case for the ISMIPB experiment as the dynamics here are driven by bed-parallel shear, as discussed below."

For ISMIPD, we include the following lines, echoing the above lines regarding ISMIPB that here we compare only the results from the case when $E_G = E_S = 8$ (P15L17-19):

> "Consistent with ISMIPB, the control of the final deformation flow in the ISMIPD experiment is bed-parallel shear, so we consider the case when the Glen enhancement factor is equal to the ESTAR shear enhancement factor, i.e., $E_G = E_S = 8$."

*R1: The discussion in page 12 about the value of the shear fraction along the transition curve is not very clear.*

We have amended the discussion about the shear fraction along the transition curve on P14L23-P15L5 to be clearer, as follows:

> "In order to examine the dynamics giving rise to the high shear-dominance peaks in Fig. 6d and Fig. 7d, we consider the following exact form of $\lambda_S^2$ (for these two-dimensional flow fields) expressible using the cartesian frame strain rate components
>
> $$\lambda_S^2 = \frac{\alpha \dot{\varepsilon}_{xx}^2 + \beta \dot{\varepsilon}_{xz}^2 + \gamma \dot{\varepsilon}_{xx} \dot{\varepsilon}_{xz}}{\dot{\varepsilon}_{xx}^2 + \dot{\varepsilon}_{xz}^2}, \tag{23}$$
>
> for some spatially varying coefficients $\alpha$, $\beta$, and $\gamma$. Since there is no surface accumulation, velocities and hence local non-rotating shear planes at the ice sheet surface are parallel to the surface. The traction free surface boundary condition implies that the numerator ($\dot{\varepsilon}'^2$) in Eq. 23, and accordingly $\lambda_S$, vanishes at the surface, except that if $\dot{\varepsilon}_e$ also vanishes, $\lambda_S$ is technically undefined. Our implementation sets $\lambda_S = 0$ for vanishing $\dot{\varepsilon}'$ in such situations. It is apparent from Eq. 23 that along the transition curves, i.e., where $\dot{\varepsilon}_{xx} = \dot{\varepsilon}_{zz} = 0$, $\lambda_S^2 = \beta$, independent of (non-zero) $\dot{\varepsilon}_{xz}$ strain rate. One can show that $\beta \to (1 - S_x^2)^2$ towards the surface (i.e., for surface slope in the $x$-direction $S_x$) along the transition curve, in order to satisfy the surface

boundary condition. This indicates that $\lambda_S$ would be finite along the transition curves all the way to the surface, except that we enforced its vanishing there. For these locations, the Glen and ESTAR viscosities corresponding to Eqs. 2-3 would tend to infinity as $\dot{\varepsilon}_e$ vanished approaching the surface, but are limited to a maximum value in the ISSM implementation.

Note that away from the transition curves $\lambda_S$ goes to zero as we approach the surface, associated with vanishing shear on the non-rotating shear plane and the corresponding dominance of normal deformations. We return to these near-surface spikes in $\lambda_S$ in the discussion."

**REVIEWER 2**

**Point 1**:

*Anisotropy*

*R2: I have had a chance to look at the other reviewer comments and I have to say that I fully agree with him/her in the main two points of his/her review. First, the authors always refer to the method as "anisotropic", it is the "A" in the acronym. That is not simply misleading, it is wrong. In an isotropic rheology the relation between stress and strain-rate is a scalar number, twice the viscosity, in a more general anisotropic rheology it is a tensor. The method presented in the paper is simply not anisotropic.*

The question of the anisotropic nature of ESTAR has already been addressed in our response to Reviewer 1. ESTAR is an anisotropic flow relation. The fact that ESTAR describes the rheology of anisotropic polycrystalline ice via a scalar enhancement factor does not preclude it from being an anisotropic flow relation. Reviewer 2 is incorrect in claiming that a scalar (collinear) relationship between the stress and strain rate tensor components is a necessary feature of an isotropic flow relation and that a tensor relationship between the stress and strain rate tensors is a necessary condition for an anisotropic rheological description. This issue has been addressed previously in the literature (Placidi et al., 2010).

The Continuum-mechanical, Anisotropic Flow model, based on an anisotropic Flow Enhancement factor (CAFFE) (Placidi et al., 2010) is an anisotropic flow relation that features an enhancement factor that is scalar function of a deformability parameter that is defined in terms the fabric and stress field. Faria (2008) provides a derivation of the anisotropy of CAFFE and Section 4 of (Placidi et al., 2010) provides further verification of its anisotropy, despite its collinear nature. Conceptually the same arguments apply to ESTAR, since it also features collinearity of the stress and strain rate tensors and a scalar enhancement that is an anisotropic function of the stress configuration.

As discussed in the response to Reviewer 1, the required ('tertiary') compatibility of the anisotropy with the stresses prevents any idea that stress can be rotated relative to the anisotropic material as the reviewer implies – this is the essence of anisotropy.

*R2: Also in this point, the authors refer to Glen flow relation vs ESTAR throughout the manuscript. ESTAR as far as I can see is a Glen flow relation with a pre-exponential factor that varies spatially to account for ice fabric. A in Equation 1 is known to depend on ice fabric and a myriad of other things (not only on temperature as stated in the manuscript), the whole point of ESTAR is that it is giving a method to estimate the effect of ice fabric on A.*

One could regard the ESTAR flow relation as it is presented, as a fabric-dependent enhancement factor for the Glen flow relation in tertiary flow. We regard this as essentially a semantic point since the task of implementing this factor is identical regardless of the viewpoint, and the enhancement function would still need a name that would encompass, empirical, tertiary and anisotropic. In terms of tertiary flow, it is not entirely certain (e.g. Treverrow et al., 2012) that tertiary flow does only differ from the Glen relation by this pre-factor. Regarding the "myriad of other things" known to influence $A$ in addition to temperature, we contend that these are usually persistent material properties, like dissolved or particulate impurities, rather than dynamically evolving features such as crystal anisotropy. We now make this distinction more clearly in the manuscript, as per P2L18-20:

> "$A(T')$ is a flow parameter ($Pa^{-n}$ $s^{-1}$), dependent on homologous temperature $T'$ and persistent material properties, for which various parameterisations exist based on laboratory tests and field measurements (e.g., Budd and Jacka, 1989; Cuffey and Paterson, 2010)."

**Point 2**:

*Tertiary creep*

*R2: And second, the empirical relation to extract the enhancement factor is based on laboratory experiments of tertiary flow, and it may well be that it can not reproduce the widespread observations of strain-induced anisotropy in polar ice. There is no discussion in the paper about how the proposed method explain observations or even expectations of the effect of ice fabric in polar flow. Do the vertical or horizontal variations of enhancement factor make sense according to observations of fabric or strain-rate fields?*

The aim of this paper is to describe the numerical implementation of a flow relation or ice rheology within an ice sheet model (P3L12-17). While ESTAR is not directly concerned with determining strain-induced anisotropy, there is a connection through the empirical observation that the flow-induced anisotropy is connected to the stress regime, so incorporation of ESTAR into an ice dynamics model is clearly an essential step in testing whether this can *"reproduce the widespread observations of strain-induced anisotropy in polar ice"*.

This flow relation, ESTAR, has been previously described (Budd et al., 2013) and its performance evaluated using ice core, borehole and laboratory data (Treverrow et al., 2015). It is outside the scope of this paper to present a comprehensive review of how ESTAR *"explain(s) observations or even expectations of the effect of ice fabric in polar flow"*, or to provide an even more general review on the effects of polycrystalline anisotropy on polar ice sheet dynamics. The background to ESTAR is described in Sects. 2.2-2.3. It is important to realise that implementation of ESTAR (or any other rheological description) in an ice sheet model provides an additional quantitative means by which that flow relation can be further tested and subsequently applied in simulating ice sheet dynamics. Regarding the validity of the assumption that tertiary creep is the predominant mechanism of deformation in the dynamically active region of polar ice sheets, we refer the reviewer to preceding comments in this response document. We also slightly modify the sections from the previous manuscript detailing conditions under which we expect ESTAR to be valid (P4L31-P5L18; quoted directly in the response to Reviewer 1 above), and provide new paragraphs discussing regions where it might not hold (P5L21-P6L12; again, quoted directly in the response to Reviewer 1 above).

In response to the last item above:

*R2: Do the vertical or horizontal variations of enhancement factor make sense according to observations of fabric or strain-rate fields?*

We refer the reviewer to Treverrow et al. (2012) and Budd et al. (2013) and references therein for a detailed discussion on the variation in strain rate enhancement according to the stress configuration. Wang and Warner (1999), Wang et al. (2002a), and Treverrow et al. (2015) each discuss the vertical variations in enhancement that are required to explain observed strain rate profiles within an ice sheet and also present fabric profile information.

**Point 3**:

*ESTAR vs tuning hardness parameter*

*R2: I also miss an important point in the paper, what is the point of using ESTAR in a large-scale ice sheet model?*

We have interpreted this question as 'what is the point of using any anisotropic flow relation in any large-scale ice sheet model?'.

There has long been recognition that a flow relation that realistically incorporates the anisotropic

rheology of polycrystalline ice is a necessary component of any ice sheet model. While a diverse range of anisotropic flow relations have been proposed, implementing such a relation within an ice sheet model in computationally efficient manner has proven challenging. Additionally, there has been a long standing awareness that using a Glen-type flow relation with a spatially-invariant enhancement factor does not provide an adequate description of polycrystalline ice rheology. What we are presenting is our attempt at incorporating a description of anisotropic rheology into an ice sheet model (P3L12-17).

*R2: I may be missing something here but large-scale models are capable to initialize the local depth-averaged hardness parameter in the Glen flow law using satellite observations. What is ESTAR adding?*

While it is possible to "tune" a "local depth-averaged hardness parameter" as well as a basal friction parameter using inverse methods to match modelled and observed surface velocities, such an approach provides no physical insight into the local controls on deformation rates and no indication how the "depth-averaged hardness" should change as the ice sheet evolves (note, we also do not want to invert for both ice rheology parameters and basal friction, as there would be an infinite number of solutions – they have "similar" effects on the surface velocities. This is why we only invert for B on floating ice, and friction on grounded ice, but never both at the same place). ESTAR is adding the ability to incorporate the spatial variability of large-scale anisotropy within ice sheets into simulations of ice sheet evolution, with a physical basis, in a fashion that will remain valid as the system evolves. It would be possible to explore using inverse methods to find appropriate global values for $E_C$ and $E_S$, since ESTAR describes the spatial variations, encoded in $\lambda_S$, e.g. (P19L32-34):

> "Indeed, with the implementation of ESTAR in ISSM, it might be possible to use inverse methods to search for values of $E_S$ and $E_C$ that improved the match between modelled and observed surface velocities."

However, there is clearly too little data in a 2D surface velocity field to determine a 3D varying "hardness parameter" throughout an ice sheet.

*R2: Could ESTAR inform the initialization of a large-scale model somehow? It could be applied to simulations were there is no satellite observations but then, as I suggested in my previous point, shouldn't we check how well does it do in a case where we have observations? Also, are there any significant vertical variations on enhancement factors that are captured by ESTAR but can't be inverted by a large-scale model? In any case, the paper needs be clearer about what is ESTAR and what can it do.*

The purpose of implementing and testing ESTAR in a large-scale ice sheet model is to explore what advantages it offers. We have shown that it is a computationally-efficient, physically based description of anisotropic flow (Sects. 5-7). We are reluctant to further expand Section 2.3 about the theory underlying ESTAR, as this paper is about the implementation of a flow relation that has already been discussed at length by Budd et al. (2013). In this manuscript, we have explained the implementation and explored what ESTAR can do in some preliminary test examples; its implementation is a necessary precursor to testing it in other more realistic situations, including the reviewer's suggestion to "check how well it does". Indeed, we are currently applying ESTAR to an Antarctic glacier system where observations are available to validate our simulations and these are forthcoming. However, it is outside the scope of this current manuscript to include full simulations of real glaciers.

As already discussed, large-scale models are capable of initialising the "local depth-averaged hardness parameter" in the Glen flow relation (relying, of course, on also tuning of the basal traction

parameter over grounded ice) using satellite observations. However, this sort of initialisation, without physical insights into the processes controlling the tuned parameter, is problematic. As the system evolves, the various physical processes that a "black box" parameter represents may change, rendering the "initialisation" values inappropriate. ESTAR could potentially inform the initialisation of an ice sheet model, through determining appropriate global values of enhancement factors $E_S$ and $E_C$, using the physics embodied in ESTAR through $\lambda_S$ to include spatial variations in 3D (P19L32-34). Indeed, in our ice shelf example, an essentially 2D flow problem in the absence of temperature effects, one could attempt to invert for the values of $E_S$ and $E_C$ to see if this improved agreement between modelled and observed velocities. However, the enhancement factor in ESTAR is not physically or dynamically equivalent to the overall flow parameter in the Glen flow relation. Furthermore, in this study, we have used the full-Stokes 3D implementation of ESTAR, which relies on more than just depth-averaged parameters. Indeed, there is insufficient data in satellite-derived surface velocities to derive a local, 3D ice flow parameter.

*R2: Also, are there any significant vertical variations on enhancement factors that are captured by ESTAR but can't be inverted by a large-scale model? In any case, the paper needs be clearer about what is ESTAR and what can it do.*

Wang et al. (2002a) demonstrate the need for a significant vertical variation in enhancement factor – simply derived from observations – together with a convincing connection to both an ESTAR-type prescription that yields that depth variation, and to the anisotropic crystal orientation fabrics observed. Treverrow et al. (2015) continues this study by exploring how ESTAR compares with other candidate flow relations. Both these studies advocate the need for a depth dependent enhancement factor, though in the general context of an assumed driving stress. This further drives the motivation for implementing ESTAR in an ice dynamics model that can solve for the full stress field to extend the testing of ESTAR against observations.

**Point 4**:

*Aspect ratios*

*R2: Finally, I find really intriguing that the authors state that results are very similar with different aspect ratios in the experiments presented in Sections 5.1 (P9-L31) and 5.2 (P11-L25). What do they mean? In the embayed ice-shelf, I would expect the ice fabric induced by extension and shear at the margins to be different. The narrower the ice-shelf is I would expect that the overall effect of lateral shearing should be more important. In the ice flow over a bumpy bed, L is the wavelength of the bedrock undulations. I would expect the increase in basal roughness to have a strong control over the orientation of the fabric. What does it mean that aspect ratios don't affect ESTAR results? I would like to see some discussion about the results presented.*

For the embayed ice shelf, as discussed above in response to reviewer 1, we have focussed our discussion in the manuscript on the case when the transverse dimension is $L = 20$ km. Figures 1-3 in this document show the Glen and ESTAR steady-state surface velocity ratio, the Glen to ESTAR thickness ratio, and $\lambda_S$, for the cases when the transverse dimensions are $L \in [20, 60, 100]$ km, respectively. The key point highlighted in these figures is that as the aspect ratio increases, the ice shelves become flatter, and the overall proportion of the ice shelf that is shear dominated does not change markedly (so the embayed ice shelf does not approach the unembayed situation). Hence, and in the interests of constraining the length of the manuscript, we have omitted the results of the cases when $L = 60$ and 100 km. We have amended the manuscript to be clearer on the impact of increasing aspect ratio, as

highlighted on P11L17-22:

> "The experiment was carried out for model domains with transverse spans $x \in [0, L]$, for $L = 20$, 60, and 100 km and along-flow dimension $y \in [0, 100]$ km. The initial ice thickness decreases uniformly from 1000 m at the grounded zone to 300, 600, and 850 m at the ice front for the $L = 20$, 60, and 100 km cases, respectively. The main features of the anisotropic effects are similar regardless of aspect ratio. This is principally because wider embayed ice shelves are flatter so that the influence of simple shear stresses on the dynamics is not particularly sensitive to aspect ratio. Accordingly, we focus our discussion on one transverse length scale: $L = 20$ km."

Regarding the ISMIP-HOM experiments (Sects. 5.2 and 5.3), we wish to thank the reviewer for encouraging us to explore more deeply the cases of bed variations with shorter wavelengths. We did observe greater variations in the anisotropic case, and we have also developed a more extensive discussion of the results (Sect. 6). The analysis also provides some indications of regions where the assumptions of the ESTAR rheology may not hold and this is now addressed in the discussion (Sect. 6), though the consequences of that type of lapse in realistic ice sheets requires further study.

Given that ESTAR does not directly calculate crystal orientations our interpretations do not directly address the reviewer's interest in fabrics, though we do provide some interpretations in the discussion (P17L31-P19L22):

[revised manuscript text omitted]

For the ISMIP-HOM experiments with larger horizontal extents (i.e., $L > 20$ km), the extent to which the longitudinal stresses impact flow and the differences between ESTAR and Glen results both decrease as $L$ increases, such that the dominant sensitivity of the flow to the bed aspect ratio is similar for both rheologies. This is generally consistent with the results concerning the longitudinal stresses presented by Pattyn et al. (2008), allowing for the fact that our simulations have evolved to steady-state whereas those tests were to find flows satisfying initial stress balance. By contrast, for smaller horizontal extents (e.g., $L = 5$ km), the increasing importance of longitudinal stresses leads to greater differences between Glen and ESTAR experiments. We overview the expanded ISMIPB and ISMIPD results in turn.

In ISMIPB the more rapid bed variation shows greater reductions in ESTAR velocities (stiffer ice) and higher spatial contrast in the velocity ratios. This is evident when comparing the $L = 5$ km results in new Fig. 4 below (Fig. 7 from the updated manuscript) with the case when $L = 20$ km in Fig. 6 from the updated manuscript. The smaller aspect ratio ($L = 5$ km) leads to a difference of up to 25% between the Glen and ESTAR velocities in the topographic depression, and a difference of almost 20% in the surface layers over the topographic bump. We have amended the manuscript in section 5.2 for ISMIPB to include discussion on this smaller aspect ratio, as follows (P14L14-22):

"In addition to the case where $L = 20$ km, we also investigated the impact of reducing the horizontal extent to $L = 5$ km. In this steeper bed scenario (Fig. 7), the ESTAR surface velocities are at least 11% slower than the Glen velocities in the surface layers across the whole domain, as much as 20% slower around the topographic bump, and up to 25% slower in the topographic depression (Fig. 7c). The much greater reductions in the magnitude of the ESTAR velocities for $L = 5$ km are a consequence of the increasing importance of longitudinal stresses in the stress balance equations for the smaller aspect ratio (Fig. 7e), and also in some areas the lower strain rates, which lead to correspondingly stiffer ice. Indeed we see a clear decline in the shear strain rate in the lower part of the bed depression in Fig. 7f, in contrast to Fig. 6f. The qualitative pattern of the longitudinal strain-rates in Fig. 7e is similar to the $L = 20$ km case, although the horizontal gradients are naturally accentuated, and the "transition curves" are displaced."

Note that while the contrast due to the ESTAR incorporation of anisotropy is greater for the $L = 5$ km case, the structure and patterns of the results are broadly similar, e.g. compare Figs. 6 and 7 from the updated manuscript. We have extended the discussion section (as suggested) to cover these additional results (P17L20-P17L30):

"For more rapidly varying bed topography in ISMIPB, with $L = 5$ km, the differences in velocity for the two flow relations reached 25%, with surface variations of 11%...These results suggest if major bed topography only varied on scales much longer than the ice thickness, close agreement between ESTAR and the Glen flow relation might be achieved more generally by choosing the

[Figure]

Figure 4: Steady-state results for the ISMIPB experiment with horizontal extent $L = 5$ km. **(a)** Glen ($E_G = 8$) along-flow velocity $v_x$ (m yr$^{-1}$); **(b)** ESTAR $v_x$ (m yr$^{-1}$); **(c)** ratio of ESTAR/Glen $v_x$ ($E_G = 8$); **(d)** ESTAR $\lambda_S$, **(e)** ESTAR normal strain (s$^{-1}$), and **(f)** ESTAR shear strain rate (s$^{-1}$). The black contours in **(e)** correspond to the curves where $\dot{\varepsilon}_{xx} = 0$.

tertiary shear enhancement factor as the Glen enhancement factor ($E_G = E_S$). This might provide a physical rationale to replace the *ad hoc* enhancement factors typically used in large-scale grounded ice sheet modelling with the value appropriate to flow dominated by simple shear. However, larger differences between velocities and vertical shear profiles emerged for the more rapid bedrock variation, where the importance of including longitudinal stresses in the momentum balance is already recognised (Pattyn et al., 2008), suggesting that adopting ESTAR would be preferable."

We do not observe similar dramatic changes in flow for ISMIPD when the horizontal extent is reduced (Fig. 5 below). In the ISMIPD experiments, sliding dominates over deformation to an extreme extent for both length scales. For $L = 20$ km (new Fig. 8 of the manuscript) the only difference between the velocities for the Glen and ESTAR cases are observed over the sticky spot. The differences between the horizontal velocities in ISMIPD tests are less than 1% in both cases, which already had prompted us to examine the ratio of the deformational contributions to the velocities as well. While these are clearly very small contributors to the flow, we observe that there is both a marked spatial contrast in the ratio, and a considerable change in that contrast between the two spatial scales.

For $L = 20$ km the aspect ratio appears to favour a simple correlation between the variation in bed friction coefficient and the deformational regime. Over the slippery spot the stiffer ice of the ESTAR case correlates with stiffer ice a reduced horizontal deformational velocity, but for $L = 5$ km a different, more complex picture emerges. The structure of the longitudinal strain rates is similar for both aspect ratios, but the pattern of shear strain rates is more complex for $L = 5$ km. We note the resulting interesting pattern in $\lambda_S$ (Fig. 5 below) compared with $\lambda_S$ for the $L = 20$ km case (Fig. 9

from the manuscript). This is commented on in the explanation of the surface spike structures (see below). Again, and as remarked in the updated manuscript (P16L4-13), the only region for which $\lambda_S$ is relevant and plays a role in the dynamics is the region over the sticky spot, where deformation is the dominant control of flow near the bed. In this region, the proportion of shear-dominated deformational flow does not increase markedly over the $L = 20$ km case. We have included discussion on the impact of the smaller aspect ratio (P16L14-29):

"The results of a FS prognostic run to steady-state for the ISMIPD experiment for $L = 5$ km are presented in Fig. 10. The $v_x$ velocity ratio (Fig. 10c) shows that very little difference is seen between results for the two flow relations, unlike the ISMIPB experiments in the previous section, where differences up to 25% between ESTAR and Glen cases emerged for the shorter bedrock periodicity. The tiny differences in overall velocities are enhanced but the patterns in Fig. 9c and Fig. 10c are similar. However, there is a significantly different picture in the ratio of deformation velocities seen in Fig. 10d, compared to Fig. 9e. The largest differences are now limited to the lower portion of the ice column and for much of the region over the slippery bed the ratio is almost unity.

The pattern of deformation regimes mapped by $\lambda_S$ (Fig. 10f) is also more complex than that seen in the preceding $L = 20$ km case (see Fig. 9d). The general structure of the normal strain rates is similar to previous experiments, but here the persistence of a band of shear (Fig. 10h) above the slippery spot at intermediate depths prevents the establishment of a vertical block of flow dominated by normal stresses. The shear profile above the sticky spot is also much weaker in the upper layers. Accordingly, $\lambda_S$ reveals a band of unevenly shear-dominated deformation which is continuous across the periodic domain. Once again, shear dominated spikes extend towards the surface in association with the vanishing of the normal strain rates.

The spatial variations in the viscosity ratio (Fig. 10e) depart significantly from those of $\lambda_S$, reflecting more strikingly than for $L = 20$ km (Fig. 9e) the combined influence of the pattern of enhancement (controlled by $E(\lambda_S)$) and the effect of different strain rates, with values both above and below the range (1.0-1.39) directly controlled by $E(\lambda_S)/E_G$."

[Figure]

Figure 5: Steady-state results for the ISMIPD experiment with horizontal extent $L = 5$ km. **(a)** Glen ($E_G = 8$) along-flow velocity $v_x$ (m yr$^{-1}$); **(b)** ESTAR $v_x$ (m yr$^{-1}$); **(c)** ratio of ESTAR/Glen $v_x$ ($E_G = 8$); **(d)** ratio of ESTAR/Glen deformation $v_x$; **(e)** ratio of ESTAR/Glen viscosity; **(f)** ESTAR $\lambda_S$; **(g)** ESTAR normal strain (s$^{-1}$); and **(h)** ESTAR shear strain rate (s$^{-1}$). The black contours in **(g)** correspond to the curves where $\dot{\varepsilon}_{xx} = 0$.

**Other modifications**

As indicated at the start of this response document, we have considerably expanded the discussion section as suggested (Sect. 6). The additional observations led to some alterations in ordering in that section. We also introduced additional diagnostics in the results section (Sect. 5) – including ratios of total and deformational velocities and of viscosities to assist our interpretation of the influence of ESTAR versus Glen rheology. We changed the presentation of shear strain rates to use a logarithmic colour scale to better capture the contrast and assist in the interpretation. The conclusion (Sect. 7) has also been expanded to cover the additional insights and to respond to reviewers' concerns.

---

## Referee Report (RR1)

I already reviewed the first version of the manuscript.

My main concern was on the presentation of the flow law (ESTAR) which is presented as a physically based alternative to describe the anisotropic rheological properties of polar ice.
I noticed 2 problems with this presentation: the anisotropic character of the flow law and the applicability of laboratory results on tertiary creep of polycrystalline ice to large portions of polar ice sheets. The same kind of remarks where raised by the second reviewer, however they are contested by the authors in their reply and changes in the manuscript for these points have been minimalist.

The authors recall that the main aim of the paper is to present the implementation of ESTAR in an ice flow model and not to provide a comprehensive review of anisotropic flow models or a justification of ESTAR as it is presented in Budd et al. (2013).
**However as we may anticipate that this paper will mostly interest ice flow modellers that are not all specialist of ice rheology, I think that it is particularly important to correctly discuss the hypothesises of the flow law and what it can do or not.**

**Anisotropic or not?**

I think there is a semantic problem with the word "anisotropy" that is used in the paper to describe different things: the "fabric", i.e. the orientation of the ice crystals, and the rheological flow law. "Anisotropy" in a polycrystal comes from the combination between *(i)* anisotropic properties at the crystal scale and *(ii)* a non-random orientation of the ice crystals. "Anisotropic" polycrystalline ice could either mean *(i)* ice with a non random crystal orientation or *(ii)* ice with a mechanical anisotropy. In general there is no need to distinguish between the two meanings as the flow laws that integrate the description of the fabric (referred in the paper as "microstructure approaches") will predict an anisotropic behaviour when the fabric is non-random. However, this is not the case with ESTAR as, because of the hypothesis that the fabric instantaneously adapts to the stress state, the fabric does not enter the constitutive relation so that there is no information about the material structure and orientation in the flow law. So we have to precise the meaning of "anisotropy" and see if it can be applied to describe the flow law ESTAR, i.e. the "A" in the acronym.

I looked for a definition of "anisotropy" in the book "Texture and anisotropy: orientation in polycrystals and their effect on material properties" by Kocks, Tomé and Wenk (1998):
- page 11:
*"A material property is isotropic when it does not depend on how the sample is 'turned' [...]; it is anisotropic when it does depend on the orientation of the sample with some external frame. Such an anisotropy is due to the arrangement of the building block of the material: its structure"*
- p14, footnote:
*"The word anisotropy will be used [Nye 1957] in connection with properties that depend on direction [..], rather than in relation to structural elements [...]. In early works, the word 'anisotropy' was used for what we call 'texture' [...]"*
Following these definitions, in the context of ice flow modelling, by "anisotropic polycrystalline ice" we should understand "ice with an anisotropic mechanical behaviour" and not "ice with a non random fabric".

In general, the anisotropic ice viscosity should be a rank-four tensor as it relates two field variables, stress and strain-rate, that are rank-two tensors. This point gave rise to the polemic around the anisotropic character of the CAFFE model.
Faria (2008) gives the following definition for anisotropy:
*"Succinctly, in continuum physics (cf. Truesdell and Noll, 1965; Hutter, 1983; Nye, 1985; Liu, 2002, and references therein) a material is said to be isotropic in a given reference configuration if*

*its response is invariant with respect to any orthogonal transformation (viz. rotation, reflection or inversion) of the body, otherwise it is called anisotropic. (When defining isotropy, some authors consider only rotations (i.e. proper orthogonal transformations), since only these are usually feasible in practice. However, this approach would be unsuitable, for example, for optically active materials (hemitropic media) and is therefore not adopted here. As remarked by Nye (1985): 'if we are to link physics to the mathematical theory of symmetry it is difficult to avoid the use of such unperformable operations'.) The set of transformations that render the material response invariant is called the 'symmetry group' of the material in the given configuration"*

We find the same two elements as in Kocks, Tomé and Wenk (1998): *(i)* anisotropy is used to describe a material property, i.e. the "material response", and *(ii)* the property depends on the orientation of the body. Faria (2008) then shows that the scalar-valued enhancement factor in CAFFE is an anisotropic function of the deviatoric stress.

Because ESTAR has no information about the material orientation, it does not enter this definition of anisotropy, as for a given solicitation (e.g. compression, simple shear) the material response will be invariant by any orthogonal transformation as, by definition, ESTAR does not include information about a material orientation. Contrary to what is claimed in response to reviewer 2, the scalar enhancement factor in ESTAR is an **isotropic** function of the deformability which itself is an **isotropic** function of the forcing represented by the deviatoric stress tensor and **n** (the normal to the non-rotating shear plane)

**ESTAR is then not an "anisotropic" flow relation, the fact that it predicts a different behaviour between compression and simple shear is not sufficient to comply with the definition of anisotropy.**

The difference between compression and simple shear comes (mainly) from the fact that, in tertiary creep, the polycrystal has different fabrics depending on the stress state, however that does not make of ESTAR an anisotropic relation.

Again I refer to Kocks, Tomé and Wenk (1998), p421:

*"Texture itself evolves with straining […]. This fact contributes, for example, to differences in stress/strain curves between tension and compression, since texture evolves differently in the two cases. It also leads to a difference between tension and torsion stress/strain curves; merely on the basis of the differently evolving textures […]. These topics are not properly labeled as consequence of 'anisotropy'; but they are macroscopic effects that can be explained on the basis of texture and the single crystal yield surface."*

To summarise, I am not contesting that laboratory results shows that in tertiary creep the fabric depends only on the stress configuration and that the mechanical response depends on the fabric and thus on the stress configuration. ESTAR captures these properties and, integrated in an ice flow model, will give a spatially varying mechanical behaviour depending on the flow configuration. However, this is not the definition of anisotropy.

**In consequence the manuscript must be revised to use the term "anisotropy" where appropriate, especially it can not be included in the acronym to describe the flow law.**

**Tertiary creep:**

In their reply, the authors suggest that we may remain agnostic about the processes that control the material response. The idea is that we can use empirical laboratory results in different situations without a proper understanding of the detailed physical processes; **this is only true as far as the "physical processes" remain the same between the empirical results and the situation where it is applied.**

Tertiary creep is usually defined from the creep curves obtained in laboratory experiments that are performed at high stresses and temperature. According to the review by Faria et al. (2014) at low temperatures and stresses (that would results in strain-rates below 10^-10 s^-1, that are not unusual in the central parts of polar ice sheets), observations are inconclusive about tertiary creep as this would require to run experiments for centuries (or more). There is then a circular logic in defining tertiary creep only from the cumulative strain that is observed in the creep curves: i.e. lab experiments shows a tertiary creep stage after 10-20% deformation, so if ice has been deformed by more than 10-20% it must be in tertiary creep and the physical processes must be the same.

The microstructure evolution in laboratory creep experiments is described by Budd et al. (2013): *"These results show negligible change from the initial isotropic structure up to about 1–2% strain. From 2% to 10% strain there is relatively rapid recrystallization giving clear well-established fabric patterns by 10%. Extending to 20% strain strengthens the fabrics, which tend to a steady state in orientation and crystal size with the continuing tertiary flow. These changes have been well studied in the laboratory to temperatures below –15C."*
This is also summarised, e.g., in the review by Faria et al. (2014), *"The accelerating part of tertiary creep is accompanied by the development of lattice preferred orientations (LPOs) and an increase in the mean grain size [...]. It should be noticed that the rapid LPO formation in such "fast" experiments is not caused by slip-driven lattice rotation, since strains of only a few percent are not sufficient to produce noticeable LPOs by lattice rotation alone (Azuma and Higashi, 1985; Jacka and Li, 2000). Rather, this early LPO formation must be related to the nucleation of new grains."*

From this I draw two conclusions:
- migration recrystallisation is an important process in the laboratory experiments against which ESTAR is calibrated. Using ESTAR in configurations where migration recrystallisation is not important is an extrapolation of the laboratory experiments.
- If tertiary creep is defined as a stage where the microstructure is in steady state, then if nucleation is not important and fabric evolution if mainly driven by slip-driven lattice rotation and e.g. rotation recrystallisation, then it could take much more than 10-20% deformation to reach a steady state and this steady state may be different from the steady state where migration recrystallisation is important.

**For the applicability of ESTAR, the question is then not too much *"is tertiary creep occurring?"* but *"is tertiary creep as seen in laboratory experiments where migration recrystallisation is important occurring?"*. The answer is clearly related to the activity of migration recrystallisation in-situ.**

The authors mention in their reply that there is evidence that migration recrystallisation is occurring at low temperature (below -15C). However, according to Faria et al. (2014), *"From the microstructural analyses of ice cores, we conclude that the formation of many and diverse subgrain boundaries and the splitting of grains by rotation recrystallization are the most fundamental mechanisms of dynamic recovery and strain accommodation in polar ice. [...] Evidence of nucleation of new grains is also observed at various depths, provided that the local concentration of strain energy is high enough (which is not seldom the case). [...] Nucleation is not predominant in polar ice, but newly nucleated grains can be found regularly in ice-core samples from any depth, and are specially frequent in samples from the lower firn."* and Diprinzio et al. (2005) note that their observations of recrystallised microstructures at shallow depth and low temperature is "unusual" compared to observations in other ice cores.

In their review, Faria et al. (2014) mainly discuss observations of the grain shape and size, and their main critics are about previous views on the occurrence of normal grain growth. I don't see any

consideration about the fact that previous views on fabric evolution should be reconsider. On the contrary we can read in their part I: *"For the upper 1500 m of EDC, Wang et al. (2003) could show that the gradual clustering of c-axes towards the vertical (which is expected for an ice dome undergoing uniaxial compression) agrees well with equivalent datasets from GRIP and Dome F (cf. Sects. 4.1 and 6.2), when plotted together with respect to a common normalized depth (i.e. depth/total ice thickness). Furthermore, a simple model of strain-induced c-axis rotation based on the assumption that basal dislocation glide is the dominant deformation mechanism (Azuma, 1994) satisfactorily reproduces the anisotropy evolution with depth in all these cores."* Note that according to the definition of "anisotropy" given by Kocks, Tomé and Wenk (1998), in the previous sentence, "anisotropy" should be replaced by "fabric". This could be extended to the upper parts of NGRIP and NEEM as Montagnat et al. (2014) show that the fabric evolution is similar down to a depth where shear becomes dominant. Note that observed fabrics in shear dominated areas and their evolution with depth are also consistent with a fabric evolution mainly driven by slip-induced lattice rotation.

In their reply, the authors seem to suggest that the development of fabrics is a proof of the existence of tertiary flow. I don't see the causality, as the plastic deformation by slip induce an evolution of the fabric, so there is no requirement to reach the tertiary creep to have a fabric. I am also surprised by this sentence :*"The observation within polar firn of microdeformation processes that are necessary for the development of fabric throughout ice sheets (e.g. Kipfstuhl et al., 2009; Faria et al., 2014)"*. Kipfstuhl et al. (2009) and Faria et al. (2014) discuss the occurrence of dynamic recrystallisation in polar firn including nucleation, but these processes are not "necessary" to explain the development of a fabric; again plastic deformation by intra-crystalline slip also induce a fabric development.

In a creep test, a steady state for the fabric (i.e. what I take as definition for tertiary creep according to the authors reply) will then only be achieved if all crystals reach a steady state orientation compared to the stress configuration or if a recrystallisation process can balance the slip-induced fabric evolution. That may well require much more than the 10-20% strain threshold used to define the tertiary creep. Jacka an Jun (Physics of Ice Core Records, 2000) present the results of laboratory experiments at low temperature and stresses; they show that migration recrystallisation, which is the dominant mechanism at high temperature and stresses, degenerates to the point where it becomes insignificant. For these cases, the experiments show negligible fabric evolution between 1 and 10% deformation so that strain-rates are nearly constant after the secondary creep minimum at 1%.

**I then maintain that the occurrence of migration recrystallisation and its importance in controlling the microstructure evolution, is an important observation to assess areas where in-situ conditions could be compared to the laboratory tests that have been used to calibrate ESTAR.**

Looking at microstructure observations in all deep ice cores where temperature and stresses are very low, there is no evidence that migration recrystallisation is dominant, so there is no clear evidence to which extent results of laboratory experiments on tertiary creep can be used to asses the in-situ ice mechanical properties. Most of the previous works on ice polycrystalline mechanical anisotropy have used these observations for calibration and or validation and they had success in explaining fabric evolution with depth. Tertiary creep observations, as used to calibrate ESTAR, can not explain such observations. **So I think it is better not to present ESTAR as an alternative to previous approaches but as complementary as they have been calibrated against observations where different conditions and thus different mico-deformation processes prevail**.

**This do not prevent to implement ESTAR in a large-scale ice-sheet model and test its performances. But the different hypotheses and the character of the flow law must be**

**described more carefully so ice flow modellers can discuss the choice of the flow law depending on the targeted applications. The authors should clarify and revise their use of both "anisotropy" and "tertiary creep" all along the manuscript.**

There was only few comments about the implementation and application parts. The authors have expanded the discussion on the ISMIP test. I have no specific comment except that again the authors should be careful in their use of the words "anisotropy" and "tertiary creep" used for the justification of the assumptions of ESTAR (the compatibility of the fabric with the flow).

---

## Referee Report (RR2)

**Journal:** *The Cryosphere*
**Manuscript:** tc-2017-54: Implementing an empirical scalar tertiary anisotropic rheology (ESTAR) into large-scale ice sheet models
**Authors:** F.S. Graham, M. Morlighem, R.C. Warner, and A. Treverrow
**Reviewer:** E.D. Waddington
**Date:** October 20, 2017.

**1 Overview**

I was invited by the editor to review this manuscript late in the process. After reading the 30 March version of the paper, and the reports by the two anonymous reviewer, I wrote the review in Section 3 below. Then, later, after seeing the response from the authors to the initial two reviewers, I wrote the additional comments following immediately as Section 2. I hope the points in both reviews will be considered.

I think the revised manuscript does a good job of demonstrating that the ESTAR flow relation of Budd et al. (2013) is an effective and low-cost constitutive relation for representing the flow of ice sheets and ice shelves. Because its prefactor varies depending on the local stress state, ESTAR appears to capture features of the flow with demonstrable advantages over the Glen flow law, which is limited by its constant prefactor.

Many years ago, in a Glaciology group discussion about anisotropy and enhancement factors here at UW, Charlie Raymond noted that a scalar enhancement factor in Glen's Flow Law should be a fine to incorporate anisotropic-fabric effects in a flow model, as long as it accurately reproduced the strain rate that was the most important or dominant strain-rate component at each position and at each time. Whatever the other strain-rate components did was less important. This manuscript shows that the ESTAR flow relation appears to be a rigorous way to do exactly that.

**2 Comments on revised submission (July, 2017)**

**Page numbers and line numbers refer to the author-annotated version of the revised manuscript, which was attached to the authors' responses to the reviewers.**

**2.1 Scientific points**

With some justification, the authors have pointed out to the reviewers that the message of their manuscript is *not* about whether the Budd et al. (2013) ESTAR flow law is anisotropic; the point of their manuscript is to test how well ESTAR works in a variety of situations in comparison with the Glen flow law.

- While I am sympathetic to their point, life also doesn't always work out the way we hope it will, and I fear that the semantics of anisotropy may continue to hound any discussion of the paper among readers (as it did among all the reviewers, including me), to the detriment of focus on their primary message. So if the authors want the paper to be widely read, and cited for the right reasons, I think it might be helpful to include a short section in the **Introduction** to address this concern head-on, as outlined in the following item. This could come at the expense of removing some to the other introductory material about anisotropy, which I felt didn't always make the point that is needed here.

- In their response to reviewers, the authors have written a long description and review of anisotropy and its treatment, which perhaps merits publication in its own right, but after reading it, I was left feeling that they have still not clearly pin-pointed the issue of contention, which is a lack of agreement among glaciologists as to what it means to be an *anisotropic flow law*. I admit to having felt ambivalent over the past few years myself, with particular reference to CAFFE and ESTAR. However, after re-reading Placidi et al. (2010) and Budd at al. (2013), and after reviewing this manuscript, I no longer feel ambivalent.

  Perhaps if these authors can clarify exactly what they mean by an *anisotropic flow relation* in this paper with a few sentences or paragraphs, it will convince readers to focus on their results, and their paper will be more widely cited.

  I expect that glaciologists agree that ice develops a non-uniform preferred crystal orientation fabric as a result of deformation, and that because an ice crystal has essentially just one slip system, this preferred crystal orientation fabric affects the deformation rate.

  **(1) First definition: prefactor must be tensorial**
  Many glaciologists would express the view that in order to be *anisotropic*, a constitutive relation between the deviatoric-stress tensor and the strain-rate tensor should be able to produce different effective viscosities associated with different stress components. For example, with an anisotropic ice fabric such as a strong vertical single pole, that ice parcel should be much stiffer to

vertical compression than to shear along horizontal planes. With this definition of anisotropy, a constitutive relation should have a tensorial prefactor. (Experimental results with multiple stress components in Budd et al. (2013) suggested that perhaps a tensor prefactor isn't needed after all, but having a tensor prefactor is still one working definition of an anisotropic flow relation.)

**(2) Second definition: response changes when sample is rotated**
Placidi et al. (2010) (Section 4.1) defined an isotropic constitutive relation as one in which the strain-rate response does not change if the sample were to be rotated in a steady stress field. They went on to show that with this definition, the CAFFE constitutive relation is anisotropic, even though it has a scalar prefactor.

They also pointed out that although many glaciologists assume that a tensorial prefactor is a necessary requirement for anisotropy (i.e. (1) above), continuum mechanics experts view definition (1) as an overly restrictive requirement for anisotropy. In general, Placidi et al. stated that (a) anisotropy, and (b) lack of collinearity between strain rate and deviatoric stress are independent properties of flow laws.

They also remarked (end of section 4.2) that their scalar prefactor in CAFFE (collinearity) is a shortcoming, because it precludes associating different effective viscosities for different components of the deviatoric stress, even though ice crystals have only one slip system.

**(3) Third definition: Relation replicates strain rates for material with anisotropic crystal fabric**
Other researchers, including the authors of this manuscript, may take the view that a constitutive relation that can realistically describe the behavior of ice with a non-uniform preferred crystal orientation (i.e. anisotropic ice) is an anisotropic flow relation, regardless of its mathematical form.

ESTAR appears to do a good job of modeling the flow of ice in tertiary creep, and has a scalar prefactor, which makes it easy to apply (that's the point of this manuscript). But, by the widely used (but restrictive) glaciolocicial definition (1) above, ESTAR is not anisotropic, because it has a scalar prefactor.

By the less-restrictive Placidi et al. definition (2), the ESTAR constitutive relation is also isotropic, but *not* because of the scalar prefactor. The ESTAR constitutive relation is isotropic because if an ice sample (which by assumption has developed a tertiary fabric in equilibrium with the current stress field) is then rotated, the ice fabric in the sample cannot rotate with the sample, because by the ESTAR assumptions, the fabric is locked to the stress field, which does not rotate in this thought experiment. There is no

change in the strain-rate response according to ESTAR, and so the flow law is not anisotropic.

Since the authors are apparently assuming a definition similar to (3), they should state this clearly (and early). However, I personally would prefer to see it clearly stated that ESTAR is a *mathematically isotropic* flow relation that works well for ice with fully developed *tertiary anisotropic* fabric.

With this perspective, I am also concerned about the places in the text that compare "*the ESTAR relation*" with "*the "isotropic Glen relation*", because they are both isotropic relations. I think the text would be more effective to focus on the differences between "*ESTAR with its stress-dependent prefactor*", and "*the Glen relation with its constant prefactor*". That's where the important new science that makes this paper valuable lies.

Perhaps the authors are moving in this direction with their text on page 6, line 30: " ... *and could be regarded as providing a variable enhancement factor for the Glen relation that incorporates the effect of anisotropy.*"

- I am puzzled by Equation (21), which represents the incoming boundary condition for flow. Equation (21) does not go to zero at the lateral boundaries ($x = 0$ and $x = L$), yet the boundary condition along the lateral margins is zero flow, i.e. $v_y(0, y) = v_y(L, y) = 0$. Since $x_{mid} = L/2$, these two conditions are incompatible in the corners where they meet. Admittedly $\exp(-5.96) = 0.003$ is small but it is not zero. Does this create some of the fine structure there, for example in Figure 4a?

- It is a pity that the ISMIP experiments used only sinusoidal bed variations, because, as the authors point out, they are unable to examine the ESTAR response in what we might call *the far field*, away form the influence of an isolated bump. Perhaps in future work, experiments with isolated bumps in a longer model domain would shed more light on the effectiveness of the ESTAR formulation?

**2.2   Editorial points and clarity**

- Page 9, line 10:
  What is a *binormal to a flow-line*?
  Also see comment about *flow line* on page 20.

  Be consistent with terminology. Do you want a hyphen in flow line (as here) or not (as on page 20, line 26)?

- Page 11, line 11:
  Consistent with what theory?

- Page 21, Line 25:
  Should be Fig. 3f.

- **Can times be slow?**
  Page 13, Line 21:
  *Rates* can be slower, but the corresponding times are *longer*.
  ". . . no more than 3% *longer* than . . ."

- **Misplaced *only***

  Page 4, line 21:
  " . . . are appropriate only for highly localized . . ."

  Page 5, line 11:
  " . . . occur only gradually . . ."

  Page 6, line 29:
  " . . . Eq.2 differs from the Glen relation only . . ." Page 20, line 1:
  " . . . varied only on scales . . ."

- Page 18, Line 1:
  ". . . between *extensile* and . . .
  Better to avoid *extensive* which can also mean *over a broad area.*
  Page 20, line 13: Same.

- Page 20, line 26:
  ". . . *the ice surface is a flow line,* . . ."
  Most glaciologists would describe the surface as a *particle path*, or equivalently in a steady state, as a *streamline*. A *flowline* would be a line on the surface that was always tangential to the ice flow at the surface, or it would be the projection of a particle path or streamline onto the free surface (assuming the velocity azimuth did not rotate with depth).

- **Hyphens**

  Page 13, line 22:
  Generally, adverb-noun pairs and adjective-noun pairs are not hyphenated when the adverb or adjective ends in a *y*.
  ". . . as computationally efficient as . . ."

  Page 18, line 4:
  ". . . leading to near-surface spikes . . ."

- **Figure 1 axes**
  The tic marks on the horizontal axis are given in units of strain, but the axis label says the units are log strain. There is a big difference. The tic labels and the axis label need to be consistent.
  The vertical axis has no magnitudes indicated. So what is the point of attributing units of inverse seconds to strain rate $\dot{\epsilon}$? See note in earlier review about the factor of 8/3.

- **Use of $e$ notation on graphs**
  Many of the figures (4, 5, 6, 7, 9, 10) use $e$ notation on graph axes. In computer code, $e$ notation is the way to indicate powers of ten, but in text and publications I think explicitly writing the powers of ten is the more appropriate way to report large and small numbers (i.e. $5.2 \times 10^{-6}$, not 5.2e-6).

**3 Review of initial submission (March 30, 2017)**

The manuscript demonstrates that the ESTAR constitutive relation from Budd et al. (2013) for ice-sheet flow can be used in a large ice-sheet model, with a negligible increase in computation time relative to computation time using the standard Glen relation with a constant enhancement factor, while producing more-realistic flow fields in the presence of anisotropy in crystal fabric orientation distributions. This is a worthy contribution meriting publication in *The Cryosphere*. However, the points raised by the two earlier reviews should be addressed first.

**3.1 Scientific points**

- Both anonymous reviewers questioned whether ESTAR is really an anisotropic flow law. I agree with their conclusion that it is not. ESTAR is a clever way to adjust the scalar Glen enhancement factor $E$ at each point to judiciously reflect the local stress state and its impact on the local fluidity, but locally it is still an isotropic flow law. As reviewer #1 stated it,
  "*for a given stress state, the mechanical response (the strain-rate state) does not change by rotation of the sample . . .*"
  A *changing* response when a sample is rotated is the essence of an anisotropic constitutive relation.

  It would be fair to say that ESTAR offers an improved flow law for ice that is anisotropic, but it is incorrect to say that ESTAR is an anisotropic flow law.

- Page 1, Line 17 -
  " *Data from laboratory ice deformation experiments can be used to define flow relations suitable for implementation in numerical ice sheet models.*"
  This may be correct, but at such dramatically different rates of strain, isn't there a strong possibility that the dominant micro-scale processes are different, and if that's the case, why should the lab experiments offer very much insight? Some more discussion might be in order.

- Both previous reviewers expressed concern that ESTAR is based on the assumption that tertiary creep has been reached everywhere. This is equivalent to the expectation (Page 4, line 24) that there is
  "*an anisotropic flow relation for polycrystalline ice in which the nature of the crystal fabric and the magnitude of strain rate enhancement, E, are both determined by the stress regime.*"
  I think we all understand that fabric actually evolves in response to strain, so in order for these two different views to be compatible, the time required for ice to undergo $\approx 10\%$ strain must be significantly less than the time required for it to move into a regime with a significantly different stress.

  Perhaps the paper by Thorsteinsson et al. (2003) would offer some ideas and discussion points about where this might be justified and where not.

  Thorsteinsson, T., E.D. Waddington and R.C. Fletcher. 2003. Spatial and temporal scales of anisotropic effects in ice-sheet flow. *Annals of Glaciology*, 37, 40-48.

- Figure 1 capton -
  "Note that the ratio . . . is approximately 8/3, . . . "
  How is a reader supposed to note this when there are no numbers on the vertical axis? The figure is just a cartoon.

**3.2 Editorial points and clarity**

- Some purists would say that *rheology* denotes a field of study, like *geology*, and a better term in the title would be *constitutive relation*.

- Page 2, line 13 -
  Cuffey and Paterson is a big book. It helps readers when you include a page number.
  Page 13, line 17 - Same comment for Paterson (1994).

- Page 3, line 9
  Section 4 is not mentioned in the outline, although all other sections are mentioned.

- Page 4, line 7 -
  Text is generally easier to read if long strings of adjectives can be avoided, or if not avoided, then correctly hyphenated to make the groupings clear.
  Suggestion:
  "a discrete vector-based description of fabric based on c-axes ..."

- Misplaced "only"s.
  Page 4, line 8 - ... appropriate *only* for highly ...
  Page 4, line 26 - ... occur *only* gradually ...
  Page 5, Line 15 - ... Eq. 3 differs from the Glen flow relation *only* in the form ...

- Page 5, Equation (5) and below -
  The idea of a simple shear $\tau'$ requires an explanation of the plane on which is measured. However, the definition of that plane follows only after $\tau'$ has been used in $\lambda_s$. This backwards order can leave a reader puzzled and unnecessarily confused for half a page or so.

- Page 8, line 12 -
  Readers would probably like at least a brief description of Taylor Hood and P1×P1 finite elements, and why they are the elements of choice. Otherwise, why mention those types of elements specifically?

- Personally, I think that authors do themselves no favours when they introduce unnecessary acronyms into their papers. The acronyms can become stumbling blocks to readers who may not read the paper front to back in a single sitting, and may abandon it, rather than bothering to dig through the paper to find the definitions of unusual acronyms. In this case, eliminating the acronyms HO (higher order) and FS (full Stokes) and just writing those terms out in full for the perhaps 2 dozen times they are used would make the paper more readable, and would add perhaps 100 words, which I (and probably $TC$ editors) would view as a small price to pay for improved clarity.

---

## Author Response (AR2)

Manuscript ID: tc-2017-54

**Title: Implementing an empirical scalar tertiary anisotropy regime (ESTAR) flow relation into large-scale ice sheet models**

Authors: Felicity Graham, Mathieu Morlighem, Roland Warner, and Adam Treverrow

**Overall response to the reviewers**

We thank the reviewers for their comments on our manuscript.

Both reviewers raise the important point of the need for further clarification in the manuscript on the definitions of anisotropy and tertiary creep. Specifically, reviewer 1 said:

*The authors recall that the main aim of the paper is to present the implementation of ESTAR in an ice flow model and not to provide a comprehensive review of anisotropic flow models or a justification of ESTAR as it is presented in Budd et al. (2013). However as we may anticipate that this paper will mostly interest ice flow modellers that are not all specialist of ice rheology, I think that it is particularly important to correctly discuss the hypothesises of the flow law and what it can do or not.*

And later on:

*This do not prevent to implement ESTAR in a large-scale ice-sheet model and test its performances. But the different hypotheses and the character of the flow law must be described more carefully so ice flow modellers can discuss the choice of the flow law depending on the targeted applications. The authors should clarify and revise their use of both "anisotropy" and "tertiary creep" all along the manuscript.*

We agree about the likely audience for this paper, and the importance of providing clear definitions about our use of the terms "anisotropy" and "tertiary creep". Accordingly, we have rearranged and expanded the manuscript to clarify the discussion on these. We have introduced a new Sect. 2 that precisely defines anisotropy and the tertiary flow regime, and their relevance to polar ice sheets. We have introduced a new Sect. 3.4 that outlines those regions where our assumptions of anisotropy and tertiary creep, as they apply to the ESTAR flow relation, apply in polar ice sheets.

We have – as detailed in the specific responses to the reviewers below – changed our perspective on isotropy and anisotropy regarding flow relations. We have introduced a new Sect. 3.5 that differentiates between the anisotropic and isotropic flow relations that are used to describe the deformation of anisotropic ice. We acknowledge that the ESTAR flow relation is a mathematically isotropic flow relation that describes the flow of anisotropic ice. Accordingly, the acronym ESTAR now stands for Empirical Scalar Tertiary Anisotropy Regime (ESTAR).

We are glad that the reviewers agree that our work is of value. Indeed, we would be concerned if anyone regarded the use of the Glen flow relation as intrinsically superior to the ESTAR flow relation.

The modifications to address the reviewers' concerns have been applied throughout the manuscript and we encourage the reviewers to consult the "diff" file we provide at the end of this response document.

We are confident that the substantial modifications we have made address the main concerns of the reviewers and provide sufficient background for readers.

Below, we have provided specific comments on the reviewer responses. In each, the reviewer comment is italicised, and our response is in normal font. We have provided direct quotations of the additions/modifications to the manuscript, where appropriate.

Kind regards,

Dr Graham and coauthors

**REVIEWER 1**

The primary concerns of reviewer 1 remain twofold. They are associated with:

1. the usage of the term anisotropy in connection with the ESTAR flow relation,

2. tertiary creep of ice in polar ice sheets, its connection with anisotropic crystal fabrics, and the validity of using observations from ice deformation experiments conducted in the laboratory to define a constitutive relation that is widely applicable to polar ice.

In the following we provide a response to the comments of reviewer 1, addressing specific points from the review where necessary.

**Point 1: Anisotropic or not?**
*R1: ESTAR is then not an "anisotropic" flow relation, the fact that it predicts a different behaviour between compression and simple shear is not sufficient to comply with the definition of anisotropy.*

We raised the question in our previous response as to whether there was more than a semantic difference between an anisotropic flow relation and a constitutive relation for ice with an induced anisotropy. After considering the reviewer's latest comments we agree that the ESTAR flow relation is not an anisotropic flow relation, in the sense cited from various authorities, since it provides a constitutive relation that is unaffected by any hypothetical local rotation of the material.

We note that the new reviewer appears to consider this a suitable shift of perspective.

To avoid any confusion by readers we have modified the acronym, recasting it to describe the flow relation – Empirical Scalar Tertiary Anisotropy Regime (ESTAR) flow relation – to emphasise that this is a constitutive relation that describes the flow of anisotropic ice, not an anisotropic flow relation. We have also introduced a new Sect. 3.5 that discusses the differences between isotropic and anisotropic flow relations.

*R1: Because ESTAR has no information about the material orientation, it does not enter this definition of anisotropy, as for a given solicitation (e.g. compression, simple shear) the material response will be invariant by any orthogonal transformation as, by definition, ESTAR does not include information about a material orientation. Contrary to what is claimed in response to reviewer 2, the scalar enhancement factor in ESTAR is an isotropic function of the deformability which itself is an isotropic function of the forcing represented by the deviatoric stress tensor and n (the normal to the non-rotating shear plane).*

We agree with the reviewer's terminology regarding absence of anisotropy in the constitutive relation, and in new Sect. 3.5 we have explained, for the benefit of "ice flow modellers that are not all specialists of ice rheology" that we are using an an isotropic constitutive relation to describe the deformation rates for ice with a deformation-induced anisotropic crystal fabric, as occurs during tertiary creep.

It is precisely to keep this restriction on the applicability of the ESTAR constitutive relation that we have carefully modified our acronym describing the constitutive relation to attach "anisotropy' to the ice in tertiary stage deformation rather than the flow relation (P2L34-P3L5):

> "We refer to the generalised flow relation proposed by Budd et al. (2013) as ESTAR (Empirical Scalar Tertiary Anisotropy Regime), since it is based on steady-state (tertiary) creep rates describing the deformation of ice with a flow-compatible induced anisotropy and features a scalar (collinear) relationship between the strain rate and deviatoric stress tensor components.

As discussed below, the ESTAR relation is a mathematically isotropic flow relation for ice with a fully developed anisotropic fabric compatible with the deformation regime."

As the new Sect. 3.5 makes clear, it is incorrect to assert that the ESTAR flow relation has no information about the material orientation. It is simply that we can only say (due to our foundations on actual experimental situations in tertiary flow) what the material properties are when the microstructure has evolved to be compatible with the stress configuration.

Our assumption that we are dealing with a fully developed deformation-induced anisotropy simply means that the material anisotropy has evolved to have a characteristic direction ($\hat{n}$) which can be predicted from the normal to the non-rotating shear plane. This is not conceptually different to measuring the axis of symmetry of a collection of crystal $c$-axes. The limitation is that unlike a microstructure based model, we cannot say what the deformations would be in response to an arbitrary applied stress – or equivalently an arbitrary rotation of the material.

In view of the public nature of The Cryosphere reviewing process we feel it worth making two further points here.

First, we suggest that if the flow relation had been presented with a unit vector $\hat{n}$ simply declared as the axis of some material anisotropy, then the constitutive relation would be regarded as anisotropic. It is the immediate substitution into that flow relation of the normal to the non-rotating shear plane as the indicator of this axis of anisotropy (within the tertiary flow assumption) that collapses the anisotropic character of our constitutive relation.

Second, given the remark that "in general the anisotropic ice viscosity should be a rank-four tensor..." we feel it worth pointing out that if such a tensor description were developed as a similarly empirical constitutive relation for ice with a flow-induced anisotropy, that also would not constitute an anisotropic constitutive relation, since there would still be no sensitivity to material rotations. Note that such a tensor viscosity could have been constructed by more speculative extrapolations from the non-scalar analyses of the experiments on tertiary deformation rates under combined stresses presented by Budd et al (2013), or Warner et al (1999). Indeed, as we mention in the paper, the most general form of flow relation proposed by Glen (1958) on the basis of isotropy was not a scalar (collinear) flow relation.

*R1: To summarise, I am not contesting that laboratory results shows that in tertiary creep the fabric depends only on the stress configuration and that the mechanical response depends on the fabric and thus on the stress configuration. ESTAR captures these properties and, integrated in an ice flow model, will give a spatially varying mechanical behaviour depending on the flow configuration. However, this is not the definition of anisotropy.*

We have conceded that our usage of anisotropy with regard to the "rheology" or more specifically, the form of the ESTAR flow relation was potentially misleading, and we agree with the new reviewer that the usage of "rheology" should be restricted to the field of study and ought not be regarded as a synonym for a constitutive relation. However, we find it difficult to follow the reviewer's logic that even though "... the mechanical response depends on the fabric ..." the reviewer does not regard this as having any connection with "anisotropy" of the deforming ice.

As the survey by the new reviewer shows, there are a variety of perspectives about the scope of the term "anisotropy". We have included a new Sect. 2 in the updated manuscript that explicitly defines what we mean by anisotropy and polar ice sheets, and a new Sect. 3.5 that discusses the distinction between anisotropic flow relations and the flow of anisotropic ice in the context of tertiary flow.

*R1: In consequence the manuscript must be revised to use the term "anisotropy" where appropriate, especially it can not be included in the acronym to describe the flow law.*

Precisely because "this paper will mostly interest ice flow modellers that are not all specialist of ice rheology", we consider it is important to keep the fact of the deformation-induced anisotropic fabric of the polycrystalline ice in our revised acronym to emphasise that the effects we are treating and the domain of applicability of our constitutive relation only concerns the tertiary flow of ice with a flow-induced anisotropy (new Sects. 2 and 3.5). In relation to the ESTAR flow relation – our descriptive characterisation of the flow relation from Budd et al. (2013) – we emphasise that this is an empirical scalar flow relation, describing the deformation of ice in the state of tertiary anisotropy. Hence the ESTAR flow relation – Empirical Scalar Tertiary Anisotropy Regime.

**Point 2: Tertiary creep and anisotropy**

Main points:

- To improve clarity, a new Sect. 2 has been added in which we articulate our usage of anisotropy and describe tertiary flow.

- We have clarified our remarks about tertiary creep being commonly encountered in polar ice sheets, particularly in regions that strongly control the large-scale dynamics, and we have also expanded our remarks about regions where our model assumptions would not apply (Sect. 3.4). Indeed, in the previous revision we already addressed this – highlighting problematic regions arising in the simulations (e.g., throughout the Discussion).

**Comments on the activity of migration recrystallisation**

It is worth noting that, like the reviewer, we consider that multiple microdeformation and recovery mechanisms contribute to the creep deformation of polar ice, and that the relative contribution of individual processes may be expected to vary spatially throughout an ice sheet due to the influence of temperature and/or stress on their activity.

Differences of opinion emerge when it comes to the activity of migration recrystallisation. The reviewer suggests that tertiary creep rates from laboratory experiments have only limited relevance to in-situ conditions because migration recrystallisation is active under laboratory conditions, but not under in-situ conditions. In contrast to this, our view is that recovery processes enabling grain boundary migration do make a contribution to microstructural development within ice sheets.

Reviewer 1 states:

*For the applicability of ESTAR, the question is then not too much 'is tertiary creep occurring?' but 'is tertiary creep as seen in laboratory experiments where migration recrystallisation is important occurring?'. The answer is clearly related to the activity of migration recrystallisation in-situ.*

And later on:

*I then maintain that the occurrence of migration recrystallisation and its importance in controlling the microstructure evolution, is an important observation to assess areas where in- situ conditions could be compared to the laboratory tests that have been used to calibrate ESTAR.*

If, as suggested, the activity of migration recrystallisation is key to determining the validity of using laboratory observations to specify a constitutive relation for polar ice sheets, then it is necessary to assess where any such threshold lies. If all microdeformation and recovery processes in polycrystalline

ice were adequately understood it would be possible to describe any variability or thresholds in their activity. In general this is not the case, and in the absence of a generally agreed threshold for the activity of migration recrystallisation, we return to the example from Law Dome, described in our initial response to the reviews as a guide (also discussed in Sect. 2 of the updated manuscript). We reiterate that for the A001 core drilled at the dome summit, a distinct small circle girdle (cone-type) fabric is observed at a depth of 318 m, where the total accumulated strain is $\sim 30\%$ and the in-situ flow regime is compression dominated, with approximately radial symmetry in the transverse rates (Fig. 3a, Budd and Jacka, 1989). The latter is supported by observations from the Dome Summit South drill site and borehole, $\sim 4$ ice thicknesses downstream of A001. The in-situ temperature within this zone is $\sim -22°$C. The important points from this are that:

- a cone-type fabric such as this cannot form by lattice rotation alone in a compression dominated setting, therefore boundary migration processes must be contributing to microstructural development in a relatively low temperature setting

- similar fabrics cone-type fabrics are also observed in uniaxial compression laboratory experiments that are conducted at higher temperatures and stresses than those encountered in-situ.

From these observations we can infer that migration recrystallisation is both active and influential under in-situ conditions – at least for temperatures as low as $\sim -22°$C. As such, a crude approximation for the activity of migration recrystallisation and a limit for extrapolating laboratory observations down to in-situ conditions might be $\sim -22°$C. This temperature is almost certainly an upper limit. Cone-type fabrics have been observed in what are expected to be compression dominated settings at other locations, e.g. cores drilled at Siple Dome (DiPrinzio et al., 2005), Byrd (Gow and Williamson, 1976), Dye 3, Greenland (Herron et al., 1985) and the Amery (Budd, 1972) and Ross ice shelves (Gow, 1963). The corresponding in-situ temperatures are even lower for some of these sites, suggesting that migration recrystallisation remains sufficiently active to influence microstructure at temperatures below $\sim -22°$C.

From the examples given above and those presented within Sects. 2 and 3.4 of the manuscript, there is sufficient evidence to support our view that the ESTAR flow relation is likely to be applicable in the dynamically active regions of the ice sheet where creep deformation makes a significant contribution to overall flow. The task of identifying the cutoffs where the ESTAR flow relation may cease to be applicable e.g., in less dynamically active zones of the ice sheet, remains an active area of research.

**Further comments on using laboratory data in the development of constitutive relations**

When it comes to the implementation of the ESTAR flow relation, a detailed discussion in response to the reviewers question 'is tertiary creep as seen in laboratory experiments where migration recrystallisation is important occurring?' while relevant, is of secondary importance. Of greater relevance is whether or not the values of enhancement prescribed by the ESTAR flow relation – which are derived from experiments – are applicable to in-situ conditions.

The reviewer states:

*In their reply, the authors seem to suggest that the development of fabrics is a proof of the existence of tertiary flow. I don't see the causality, as the plastic deformation by slip induce an evolution of the fabric, so there is no requirement to reach the tertiary creep to have a fabric.*

Some clarifying remarks on anisotropy and tertiary creep are required here. We did not intend to suggest that one had to reach tertiary creep to have anisotropic fabric. It is true, we do not have

to reach tertiary creep to have an anisotropic fabric. However, the strain induced development of an anisotropic fabric is an integral part of reaching tertiary creep (where tertiary creep corresponds to the point where the microstructure (and strain rate) have evolved to be compatible with the imposed stresses). We assume that microstructural anisotropy is the cause of the different deformation responses observed for different stress regimes. If and when the stress configuration changes the microstructure will evolve in order to establish a new tertiary creep state.

Since the ESTAR flow relation is an empirical relationship used to define an enhancement factor dependent on the stress configuration, it will work provided the nature of mechanical anisotropy identified in experiments is consistent with in-situ conditions. It would not be possible to sensibly apply the ESTAR flow relation to simulating ice sheet dynamics if it was possible to point to widespread regions in an ice sheet where the fabric has evolved in such a way as to reduce the bulk deformation rate below that which would be expected from ice with an isotropic fabric. To our knowledge there are no ice core (fabric) and corresponding borehole deformation (other in situ measurements) that support the concept of widespread hardening in response to fabric evolution.

We do not suggest that the ESTAR flow relation is the final word on simulating anisotropic polar ice (Introduction, P3L5-7). We use the ESTAR flow relation on the basis that it provides better overall performance in simulating ice sheet dynamics than the Glen flow relation. There will be regions where the ESTAR flow relation is not applicable; however, we make it clear in the manuscript where we think it will and will not work, and comment on the relative importance of these zones to the overall dynamics of an ice sheet (Sect. 3.4). For example, the ESTAR flow relation may overestimate strain rates in the coldest, near surface layers of the deep interior of a polar ice sheet where accumulated strains are low (P10L19-26). Since strain rates are correspondingly low in such regions the overall impact of 'getting it wrong' here will be minimal. We discuss the significance of 'getting it wrong' in the new Sect. 3.4.

Lastly, the use of laboratory data to constrain constitutive relations is not unique to the ESTAR flow relation. At some level, most if not all flow relations are based on, calibrated by, or validated using experimental results. In many cases (including the ESTAR flow relation) specification of the limiting values of strain rate enhancement associated with the development of fabrics are obtained from laboratory results, so the ESTAR flow relation sets no new precedent here. Furthermore, the prescription of the temperature dependence of ice flow rates that is commonly used in many models is based on experimental secondary creep rates.

**The strain required for the development of tertiary creep**

It is clear from many laboratory studies that tertiary creep occurs at strains of $\sim 10\%$ – we regard this as a lower limit. Since this was not already clear in the manuscript we now indicate that tertiary creep occurs at $\geq 10\%$ strain (Sect. 2, e.g., P4L18-32).

There are suggestions that the strain required to develop steady-state strain rates and compatible fabrics at lower temperatures and stresses may be higher than values obtained from laboratory experiments. In the example from the Law Dome A001 ice core described above, a distinct cone-type fabric is observed for a strain of $\sim 30\%$, providing an upper limit for the strain required to develop a compatible fabric under in-situ conditions. Since the fabric is already well developed at this point, the actual strain required to achieve tertiary creep was probably less. This idea is further supported by experimental observations (Jacka and Maccagnan, 1984) that show how a steady-state (tertiary) creep rate is achieved prior to corresponding anisotropic fabric reaching a steady-state. The suggestion here is that the strain rate does not continue to evolve so much once the most obstructive grains have been

removed by fabric development, i.e. the continued strengthening of an already anisotropic fabric has a minor influence on the strain rates. We discuss this result in new Sect. 2.

While the experiments of Jacka and Li (2000) cited by the reviewer provide some support for the requirement of higher accumulated strains to develop tertiary creep at low temperatures and stresses, the results of these experiments are somewhat inconclusive and should be interpreted with care. These experiments were conducted at constant applied loads, not constant applied stresses. Due to an increase in the cross sectional area of the sample with increasing strain the stress effectively decreases throughout the experiments. This effect can be observed by the decrease in creep rates at strains $\geq 10\%$ for the higher temperature and/or stress experiments (see their Figures 7 & 8). At a strain of 10% the corresponding decrease in stress will lead to strain rates $\sim 30\%$ lower than the value expected for the stress that was applied at the start of the experiment. From this perspective the nearly constant strain rates referred to by the reviewer actually indicate a modest (30%) enhancement in the flow relation by 10%. Accordingly, we suggest that the cone-type fabric from A001 at $\sim 30\%$ and $\sim -22°$C provides a more robust indication of the levels of strain required for fabric development (Sect. 2, P4L33-P5L8).

Throughout much of an ice sheet the evolution in the stress configuration and corresponding compatible microstructure along ice streamlines is gradual, hence the strain required for the microstructure to adapt to changes is substantially less than observed in laboratory experiments that commence on samples with an initially isotropic fabric.

**REVIEWER 2**

**From Section 2.1: Scientific points (July 2017)**:

*R2: I think it might be helpful to include a short section in the Introduction to address [the semantics of anisotropy] head-on, as outlined in the following item. This could come at the expense of removing some to the other introductory material about anisotropy, which I felt didn't always make the point that is needed here.*

*Perhaps if the authors can clarify exactly what they mean by an anisotropic flow relation in this paper with a few sentences or paragraphs, it will convince readers to focus on their results, and their paper will be more widely cited. I expect that glaciologists agree that ice develops a non-uniform preferred crystal orientation fabric as a result of deformation, and that because an ice crystal has essentially just one slip system, this preferred crystal orientation fabric affects the deformation rate.*

We greatly appreciate the considerable effort expended by the reviewer in detailing each of the different concepts of what might be meant by an "anisotropic flow relation". In the revised manuscript we have adopted this reviewer's original perspective that (Introduction, P3L4-5):

> "...the ESTAR relation is a mathematically isotropic flow relation for ice with a fully developed anisotropic fabric compatible with the deformation regime."

Accordingly, we have amended the acronym to now stand for: Empirical Scalar Tertiary Anisotropy Regime, intending the usage "ESTAR flow relation" to correspond to "Glen flow relation". We have added a new Sect. 2 to discuss anisotropy and polar ice sheets, both the experimental and observational evidence, and the importance of its consideration in constitutive relations for use in numerical ice sheet models. For completeness, and in line with the reviewer's recommendation from both reviews, we have also added a new Section 3.5 to discuss the seeming paradox of using an isotropic constitutive relation to describe the flow of ice with a deformation-compatible tertiary anisotropic crystal fabric. We have added this at the end of the section on constitutive relations, rather than in the introduction. Given that we have adopted the reviewer's suggested perspective of presenting the ESTAR flow relation as a mathematically isotropic flow relation for anisotropic ice, we consider that this was the appropriate place to put this discussion.

*R2: I am puzzled by Equation (21), which represents the incoming boundary condition for flow. Equation (21) does not go to zero at the lateral boundaries ($x = 0$ and $x = L$), yet the boundary condition along the lateral margins is zero flow, i.e., $v_y(0, y) = v_y(L, y) = 0$. Since $x_{mid} = L/2$, these two conditions are incompatible in the corners where they meet. Admittedly, exp(-5.96)=0.003 is small, but it is not zero. Does this create some of the fine structure there, for example in Figure 4a?*

In the original set of experiments, we subtracted the small value of $v_y$ at $x = 0$ and $x = L$ to ensure that $v_y$ satisfies the no-slip boundary condition at the margin. We have revised the manuscript to be clear on this, as follows (P15L9-13):

> "At the inflow boundary, the $y$-component of velocity is set by

$$V(x) = V_0 e^{-\left[\frac{5(x-x_{\mathrm{mid}})}{2L}\right]^8},\tag{1}$$

$$v_y(x, 0) = V(x) - V(0),\tag{2}$$

> where $V_0 = 100 \text{ m yr}^{-1}$ and $x_{\mathrm{mid}} = L/2$. This ensures that $v_y(x, 0)$ satisfies the no-slip boundary condition on the margins."

*R2: It is a pity that the ISMIP experiments used only sinusoidal bed variations, because, as the authors point out, they are unable to examine the ESTAR response in what we might call the far field,*

*away from the influence of an isolated bump. Perhaps in future work, experiments with isolated bumps in a longer model domain would shed more light on the effectiveness of the ESTAR formulation?*

We agree with the reviewer and are planning further tests of the ESTAR flow relation over synthetic and realistic domains. It would certainly be desirable to decouple the spatial scale of a bump from any effects of periodic boundary conditions.

**From Section 2.2: Editorial points and clarity (July 2017)**:

*R2: Page 9, line 10: binormal to a flow line*

A binormal to a flow line has been defined (P12L17) as "the unit vector orthogonal to both the tangent vector and the normal vector". Throughout the manuscript, we have amended flow-line/flowline to flow line, unless where the word streamline is more appropriate.

*R2: Page 11, line 11: Consistent with what theory?*

Our results are consistent with those predicted from Ern and Guermond (2004), as discussed also on P12L22-26. This reference has been added on P14L18.

*R2: Page 21, line 25: Should be Fig. 3f.*
Amended (P15L28).

*R2: Can times be slow?*
Amended (P16L23).

*R2: Misplaced only*
Amended.

*R2: Extensive/extensile/tensile*

We have removed all occurrences of *extensive*, instead adopting *tensile* or *extensional*, where appropriate. E.g., see P15L23-24 and P16L4-5.

*R2: Flow line/streamline/particle path*

We have replaced flow line with streamline throughout the manuscript.

*R2: Hyphens*

We have removed hyphens on computationally efficient, and added a hyphen to near-surface (P10L21).

*R2: Figure 1 axes*

We have amended Fig. 1 axes labels as suggested. The x-axis is strain, but plotted on a log scale. The y-axis (strain rate) is also plotted on a log scale and the units have been removed from the label. We have specified that this figure is intended to be a cartoon to illustrate the relationships between strain and strain rates for different stress regimes and at different stages of deformation.

*R2: Use of e notation on graphs*

We have removed all occurrences of the *e* notation on graph axes.

**From Section 3.1: Scientific points (March 2017)**:

*R2: It would be fair to say that ESTAR offers an improved flow law for ice that is anisotropic, but it is incorrect to say that ESTAR is an anisotropic flow law*

We agree with the reviewer. As mentioned above in response to the scientific points from the reviewer's section 2.1, we have removed all reference to the ESTAR flow relation as an anisotropic flow

relation. Rather, we clarify that the ESTAR flow relation is a mathematically isotropic flow relation for describing the influence of induced anisotropy (P3L4-5).

*R2: "Data from laboratory ice deformation experiments can be used to define flow relations suitable for implementation in numerical ice sheet models." This may be correct, but at such dramatically different rates of strain, isn't there a strong possibility that the dominant micro-scale processes are different, and if that's the case, why should the lab experiments offer very much insight? Some more discussion might be in order.*

We have made detailed comments related to this same issue in our response to reviewer 1 above: "Further comments on using laboratory data in the development of constitutive relations". To briefly summarise: laboratory results are broadly transferable to polar ice sheets, particularly the dynamically active zones, but identifying thresholds in the activity of specific microdeformation and recovery processes (in order to determine when laboratory observations are no longer representative of in-situ conditions) remains an area of active research. In the updated manuscript we have added discussion about the rates of strain observed in laboratory experiments of tertiary creep and how they compare with observations from the field (e.g., from the A001 core drilled at the Law Dome summit; Budd and Jacka, 1989; Sect. 2 of the updated manuscript), as well as the applicability of laboratory experiments in constraining and validating constitutive relations (Sects. 3.2 and 3.4).

*R2: Both previous reviewers expressed concern that ESTAR is based on the assumption that tertiary creep has been reached everywhere. This is equivalent to the expectation that there is "an anisotropic flow relation for polycrystalline ice in which the nature of the crystal fabric and the magnitude of strain rate enhancement, E, are both determined by the stress regime." I think we all understand that fabric actually evolves in response to strain, so in order for these two different views to be compatible, the time required for ice to undergo $\approx 10\%$ strain must be significantly less than the time required for it to move into a regime with a significantly different stress. Perhaps the paper by Thorsteinsson et al. (2003) would offer some ideas and discussion points about where this might be justified and where not.*

We thank the reviewer for the insightful reference (Thorsteinsson et al., 2003) that highlights situations where the ESTAR flow relation might not apply. We have incorporated this reference in new Sect. 3.4, which outlines the domain of applicability of the ESTAR flow relation, and where the assumptions underlying the ESTAR flow relation are not expected to hold.

In the first revision of the paper we had already made efforts to indicate the type of region where this compatibility of anisotropy and stress regime was unlikely to be justified, and to address the connection between the time required to develop a compatible fabric (in terms of accumulating strain), the distance travelled in that time and the spatial scales over which the stress regime might change significantly. In the Discussion section we explicitly introduced the concept of a transition scale – the distance travelled in accumulating $\sim 10\%$ strain and used this to draw attention to regions in the simulations where the assumption of tertiary flow with a compatible fabric was expected to fail.

In the current version we have further expanded on this matter in the Sects. 2 and 3.4, as well as retaining the remarks in the Discussion section – e.g., P22-23.

*R2: Figure 1 caption. "Note that the ratio...is approximately 8/3..." How is a reader suppose to note this when there are no numbers on the vertical axis? The figure is just a cartoon.*

We have amended the caption on Fig. 1 and made the intent of the figure more explicit.

**From Section 3.2: Editorial points (March 2017)**:
*R2: Some purists would say that rheology denotes a field of study, like geology, and a better term in*

*the title would be constitutive relation.*

We agree with the reviewer and have replaced "rheology" with "constitutive relation" or "flow relation" throughout the manuscript.

*R2: Cuffey and Paterson (2010) and Paterson (1994) references need page numbers*

We have added a page number to both the Cuffey and Paterson (2010) and Paterson (1994) references.

*R2: Section 4 not mentioned in outline, although all other sections are mentioned.*
Amended (P3L13-20).

*R2: Avoid long strings of adjectives*
Amended as suggested.

*R2: Misplaced "only"s*
Amended.

*R2: Definition of NRSP moved before idea of simple shear*

We have reordered the material in section 3.3 so that the discussion of the non-rotating shear plane and the shear acting on that plane is introduced before the shear fraction is formally defined (P8-9).

*R2: Taylor Hood and P1×P1 elements*

We have specified the type of finite elements used in the verification process as it is a relevant detail for replicability, especially for modellers hoping to implement the ESTAR flow relation into their own models. Should readers require more information, details on the finite element method are provided in the reference (Ern and Guermond, 2004), and the ISSM code is available online.

*R2: ...In this case, eliminating the acronyms HO (higher order) and FS (full Stokes) and just writing those terms out in full for the perhaps 2 dozen times they are used would make the paper more readable...*

We agree with the reviewer and have removed the acronyms HO and FS.

Jacka, T. and Li, J.: Flow rate and crystal orientation fabrics in compression of polycrystalline ice at low temperatures and stresses, in: Physics of Ice Core Records, edited by Hondoh, T., pp. 83–101, The 
[revised manuscript text omitted]

---

## Author Response (AR3)

**Manuscript ID: tc-2017-54**

**Title: Implementing an empirical scalar tertiary anisotropy regime (ESTAR) flow relation into large-scale ice sheet models**

**Authors: Felicity Graham, Mathieu Morlighem, Roland Warner, and Adam Treverrow**

Dear Olivier,

Thank you very much for your comments. We have taken into consideration your suggestions, modifying the manuscript accordingly.

Your comments are presented in italics below, followed by our response. A version of the manuscript with marked-up changes is appended to this document.

We're uncertain which of the reviewers we can thank by name in the acknowledgements, but would like to do so in the final version.

Kind regards,

Dr Graham and coauthors

**ISMIP-HOM experiments**

*P1: I am confused by the presentation of these experiments as you are speaking of "steady-state", "timestep", "initial surface", etc... whereas these experiments are diagnostic and should not imply time and surface evolution: the velocity field should be computed for given (fixed) geometry and boundary condition. If you are solving a modified version of these tests, which I don't believe is the case, it should be mentioned. Else, why do you need a time step to solve these diagnostic experiments? This should be corrected in many places (text and figure captions), the terms "steady-state", "timestep", "initial surface", "prognostic", etc... should be removed/replaced.*

Apologies for any confusion. We have indeed modified the ISMIP-HOM experiments by considering *prognostic*, rather than diagnostic, simulations of each. We now clarify this in the outline of the paper at the end of the Introduction P3L18-19:

> "... including prognostic (evolving) simulations based on selected experiments from the Ice Sheet Model Intercomparison Project for Higher Order Models ..."

and on P14L24-26 of the text:

> "The ISMIP-HOM experiments were diagnostic. In contrast, we have taken the same geometries and boundary conditions, but allowed the velocity, surface, and thickness fields to evolve to steady-state, as defined in the corresponding sections below. We append a "p" for prognostic to experiment names, where appropriate."

And again in each of the relevant subsections, namely P17L6-9 for the ISMIPB experiment:

> "In each case we used the full-Stokes version of ISSM to carry out a *prognostic* simulation of the ISMIPB experiment (the original ISMIPB was diagnostic). Prognostic steady-state is regarded as reached when the mean velocity change over the surface mesh points is less than $1 \times 10^{-2}$ m yr$^{-1}$ between two consecutive time steps of $\triangle t = 1$ yr for this and the following ISMIP-HOM experiment."

And P18L25-26 for the ISMIPD experiment:

> "Once again, we employ the full-Stokes solver in ISSM for *prognostic* simulations evolved to steady-state (as defined in Sect. 6.2)."

Also, when referring to our simulations, rather than the original experiments we have appended a lower-case "p" to the experiment names to keep the prognostic character in the reader's mind, e.g. P17L10 ISMIPBp, and P18L22 ISMIPDp.

*P2: Also, as I understand, you have modified the Glen's flow law parameter such that the new value (:8) times the Glen's enhancement factor (=8) is equal to the ISMIP-HOM one (8:8=1). If this is correct, then I would expect that your results for the Glen's law are comparable with the ISMIP full-Stokes results, which seems not to be the case, especially for experiments B. This should be commented and discussed. On the same line, the sentence "to ensure our results are as close as possible to the original ISMIP-HOM experiments" page 17, line 3 should write "to ensure the solution with the Glen law does correspond to the original ISMIP-HOM experiments."*

Our results will differ from the original ISMIP-HOM experiments presented in Pattyn et al. (2008) because ours are prognostic, as mentioned above. We have modified the sentence as suggested (P17L2-3).

You may recall that the first version of our paper also included diagnostic ISMIPD experiments, but these were removed to shorten the paper, as we felt the prognostic cases were more relevant. We discussed this in the response to the reviewers at the time.

*P3: Sections 6.2 and 6.3 are quite long and could be shortened by discussing in parallel the two cases for L.*

In both Sect. 6.2 and 6.3, we felt the logic and flow of the presentation of results required separation of discussion between the two aspect ratio cases. This is particularly the case in Sect. 6.3 where the change in aspect ratio leads to a marked difference in the results. Nevertheless, we have edited and shortened both sections considerably (see marked-up version of the manuscript attached).

**Other remarks**

*page 1, line 4: scalar, constitutive or flow relation -> scalar, constitutive relation (specified below, not necessary here)*

Modified

*page 5, 2: (Fig. 3a Budd and Jacka, 1989) -> (Fig. 3a in Budd and Jacka, 1989)*

Modified

*page 8, line 21: in Eq. 2, Eq. 3 differs -> in Eq. (2), Eq. (3) (and everywhere in the manuscript, refer to `https: // www. the-cryosphere. net/ for_ authors/ manuscript_ preparation. html`)*

Thank you for the reminder. Modified

*Eq. (8): what is $w_D$, a vector? Not clear from above where it is said it is the deformation (tensor). Its expression should be given.*

The quantity $\omega_{\mathbf{D}}$ is a vector. We have added "vector" where this is defined on P11L31-32:

"...$\hat{\omega}_{\mathbf{D}}$ is the unit vector parallel to that part of the vorticity vector that is associated solely with deformation ($\omega_{\mathbf{D}}$)..."

As highlighted on P12L17-21, it is not a trivial exercise to decompose the total vorticity into its rotational and deformational components. Instead, we use Eq. (11) to derive the expression in Eq. (12) for the perpendicular component of deformational vorticity vector $\omega_{\mathbf{D}}^{\perp}$ that is used to calculate the shear strain rate on the non-rotating shear plane in Eq. (7).

*page 12, line 6: define what is v.*

Defined on P12L1:

> "... is defined as the normalised cross product of the velocity ($\mathbf{v}$) and the deformational vorticity..."

*page 13, line 8: stress (bridging effects van der Veen and Whillans, 1989) -> (bridging effects, van der Veen and Whillans, 1989)*

Modified

*page 15, line 14: about the boundary condition below the ice-shelf, it is not mentioned if you are prescribing a normal friction accounting for change in water pressure when the basal surface elevation changes (see Durand et al 2009). If not, it might explain why you need such small timestep (< 1 day) to have a stable ice-shelf. This should be discussed.*

The ice shelf basal boundary conditions for momentum balance are (as standard) vanishing tangential (shear) stress, reflecting the absence of basal drag and a normal stress balanced by the pressure of the sea water. Our ice shelf simulations do not contain any grounded-to-floating transitions so the question of a contact treatment of ungrounding as presented by Durand et al. (2009) is not directly relevant here.

For the embayed ice shelf results presented in Sect. 6.1 we use the higher-order version of ISSM. With the neglect of bridging stresses in higher-order models this provides a simple basal boundary condition due to the form of the ice pressure in the higher-order model, i.e.,

$$P = \sigma'_{zz} + \rho_i g(s - z).$$

Due to the neglect of bridging stresses, the vertical component of the force balance at the ice shelf base ($z = b$) requires the ice lithostatic pressure term ($\rho_i g(s - b)$) to balance the sea water pressure ($-\rho_w g b$), allowing the horizontal components of basal force balance, which now only involve deviatoric stresses, to be rearranged to form homogeneous natural boundary conditions for the 3D equations corresponding to momentum balance in the two horizontal directions as in Pattyn (2003).

The small time step (< 1 day) reflected the fact that the initial geometry was far from a compatible geometry leading to stability problems with the thickness evolution for the full-Stokes model of the embayed ice shelf. This full-Stokes model was mentioned in the manuscript only for comparison of computational time (i.e., corresponding to table 1). As discussed below, in the updated manuscript we have modified the presentation of computational efficiency of the ESTAR flow relation (table 1) to correspond to the higher-order experiment discussed in the results Sect. 6.1, shifting the focus to the question of parallel efficiency, thus removing what was a passing reference to the full-Stokes case.

*page 19, line 23: the free surface. in the next section -> the free surface. In the next section*

Modified

*page 19, line 24: flow relation . Shear dominates -> flow relation. Shear dominates*

Modified

*page 19, line 33: The results of a full-Stokes prognostic run to steady-state for the ISMIPD -> ISMIPD is not a prognostic simulation so that there is no steady state to look at.*

As explained earlier, we did modify these IMSIP-HOM experiments to be prognostic simulations (P14L24-26). So, we have carried out the simulation to steady-state. In revising this section, we hope that this is now clear throughout.

*page 21, line 32: Accordingly, in the flow regime of these prognostic experiments -> again, ISMIP B*

*and D experiments are diagnostic, not prognostic*

The original ISMIPB and ISMIPD experiments were diagnostic; here we have modified them to be prognostic.

*Figure 8: the size of the subplots should be similar than in the previous figures*

Modified

*Table 1: It is not clear from the text that you are also increasing the number of CPUs used to run this comparison between Glen and ESTAR walltime. Are you doing parallel simulations? Then you should also discuss this results in term of scalability? Why do you need the CPUs to increase much faster than the DOFs? I don't also understand why the number of DOFs is smaller than the Vertices. With the FEM, for 3D Stokes, the DOF per node is 4, so that I would expect that the total number of DOFs is 4 x the Vertices.*

The aim was to indicate that the computational overhead of using the ESTAR flow relation was small, which can be made equally well using the higher-order model. Accordingly, to simplify the discussion, we now present results for the higher-order ice shelf model, modifying table 1 and our discussion of the computational efficiency of the ESTAR flow relation to correspond directly with the simulations presented in Sect. 6.1, rather than the full-Stokes model (the results of which were/are not discussed in Sect. 6.1). Table 1 presents computation times for the higher-order model of the embayed ice shelf for each flow relation. To address your question, these are parallel simulations, and we now focus on the issue of scalability. We fix the number of vertices (a total of 80080 horizontal vertices over 10 vertical layers), and run simulations for 1000 years, for an increasing number of processors. The parallel efficiency of the ESTAR model is marginally better than the Glen flow relation for each parallel simulation, despite having longer computation times. See updated table 1, the results of which are discussed on P16L15-18.

**Other changes**

We have made some further minor changes to the manuscript:

- replaced word "spike" with "peak" in reference to the high shear-dominance peaks in $\lambda_S$ (e.g., P18L9)

- Sects. 6.2 and 6.3 have been shortened

- some grammar has been fixed (e.g., parameterise)

**References**

Durand, G., Gagliardini, O., de Fleurian, B., Zwinger, T., and Le Meur, E.: Marine ice sheet dynamics: Hysteresis and neutral equilibrium, J. Geophys. Res., 114, 1–10, doi:10.1029/2008JF001170, 2009.

[revised manuscript text omitted]

---

## Author Response (AR4)

Manuscript ID: tc-2017-54

**Title: Implementing an empirical scalar tertiary anisotropic rheology (ESTAR) into large-scale ice sheet models**

**Authors: Felicity Graham, Mathieu Morlighem, Roland Warner, and Adam Treverrow**

Dear Olivier,

Thank you for your most recent comments.

In relation to the ISMIP-HOM experiments, we have decided to present only the prognostic results. Our reasoning behind this is detailed below, and in the manuscript.

In the updated manuscript, we have included discussion of the steady-state surface elevations for each of ISMIPBp and ISMIPDp, as you suggest.

Our responses to your specific comments are below, with your comments presented in italics. All line number and figure references are to the revised manuscript unless otherwise indicated. A version of the manuscript with all changes marked-up is provided at the end of this response document.

Kind regards,

Dr Graham and coauthors

**The choice of prognostic ISMIP-HOM**

*The fact that you were performing prognostic simulations adapted from the ISMIP-HOM experiments was indeed not clear. I am still a bit confuse about this choice as it doesn't allow any comparison with previously published ISMIP-HOM results. Also, if you keep this choice, I would suggest to make clearer in the manuscript that your results should not be compared to the ISMIP-HOM ones and clearly justify the choice of running these initially diagnostic tests in transient. I think the paper can now be accepted in TC, but I let you decide if you want to keep the prognostic experiments from ISMIP-HOM (and therefore justify why and make more clear this choice) or change to the initial ones. There are few more points listed below that should be corrected before publication.*

We chose to present prognostic runs because they are more in keeping with the idea of the flow field and stress regime being steady in time. That the stress regime in the prognostic experiments is steady in time makes for a more meaningful comparison between the Glen and ESTAR flow relations for each case. A set of diagnostic Glen relation cases here would be a trivial revisit of the ISSM results (Larour et al., 2012) for the original ISMIP-HOM (Pattyn et al., 2008) experiments. We have clarified this point in the manuscript, to make our reasons clear, when first introducing the prognostic ISMIP-HOM experiments (P14L24-P15L3):

> "The ISMIP-HOM experiments were diagnostic. In contrast, we have taken the same geometries and boundary conditions, which are already familiar to the modelling community, but allowed the velocity, surface, and thickness fields to evolve to steady-state, as defined in the corresponding sections below. We choose to present steady-state results from prognostic simulations based on ISMIP-HOM experiments (supplemented by prescribing zero accumulation or loss of ice) using the Glen and ESTAR flow relations because in this situation the ice sheets, the flow fields, and stress regimes are steady in time. This is more in keeping with the assumptions underlying the ESTAR flow relation than a simple diagnostic experiment for a prescribed geometry. It is also of interest to see the differences in the dynamic evolution of the systems resulting from the different material constitutive relations. While our focus is on the differences between the results for the ESTAR and Glen flow relations, the latter results provide a direct extension

to the original ISMIP-HOM experiments presented in Pattyn et al. (2008), which may be of some interest, rather than being directly comparable with them. We append a "p" for prognostic to experiment names, where appropriate. The ice sheet is isothermal, as in the original ISMIP-HOM experiments."

**Other points**:

*As you are running the ISMIP-HOM tests in transient, you should specify which accumulation/ablation function is applied to solve the free surface evolution. I guess it is null, but any other function fulfilling periodicity and having a null integral over the domain should work?*

We agree that we should have added the specification that there is zero mass input (accumulation/ablation) when introducing each ISMIP-HOM based experiment rather than simply mentioning it in passing during the interpretation.

In the updated manuscript we have now specified that zero accumulation is applied for both ISMIP-HOM experiments. That is, for the ISMIPBp experiment (P17L6-7):

"Prognostic experiments require specification of the local mass balance – there is no surface or basal melting or accumulation at any point in the domain."

and for the ISMIPDp experiment (P19L8):

"There is no surface or basal melting or accumulation at any point in the domain."

That there is zero accumulation in the ISMIPBp experiment is also mentioned on P18L14, where we discuss the fact that the surface here is both a streamline and a non-rotating shear plane.

*For the results, the comparison should not only be done on the velocity but also on the surface elevation, which should be different from Glen and ESTAR? These surface elevation differences should be quantified.*

The differences in the steady-state surface elevations for each of the prognostic ISMIP-HOM experiments for the Glen and ESTAR flow relations are now discussed in the updated manuscript, in conjunction with the corresponding new figure panels in each of Figs. 6-9, which show the final surface elevations and elevation changes from the initial geometry.

For ISMIPBp $L = 20$ km (P17L32-P18L2):

"The steady-state surface elevations for the Glen and ESTAR flow relations are everywhere within 1 m (Fig. 6d). The differences between initial and final elevations show that the Glen steady-state surface is slightly lower than the ESTAR surface between 0-5 km and 15-20 km and slightly higher between 5 and 15 km. This reflects differing contrasts in the horizontal velocities for the two flow relations. Greater differences are seen in the next example."

and $L = 5$ km (P18L29-31):

"The patterns in the steady-state surface elevations for the $L = 5$ km case (Fig. 7d) are consistent with the $L = 20$ km case (Fig. 6d), although here the differences between the steady-state surface elevations for each flow relation are greater ($\pm 1.25$ m)."

For ISMIPDp $L = 20$ km (P19L24-28):

"The ISMIPDp steady-state surface elevations for each flow relation are shown in Fig. 8e. The Glen steady-state surface is lower than ESTAR over the sticky spot and higher than the ESTAR surface over the slippery spot. The absolute differences between the two steady-state surfaces are everywhere less than 0.42 m. The evolution of the ice thickness and surface by approximately 5 m above and below the initial linear profile produces a marked change in the surface slope,

given the 0.1° slope of the bedrock ramp, which corresponds to a fall of 34.9 m over the 20 km domain."

and $L = 5$ km (P20L13-18):

"In the $L = 5$ km case, we see that a slight reverse slope develops downstream of the slippery spot for both flow relations, while the highest point occurs approaching the peak of the basal drag coefficient. Again, the steady-state surface elevations differences (Fig. 9e) show similar patterns in the $L = 5$ km case as in the $L = 20$ km case, with smaller differences between the two final surface elevations (i.e., 0.28 m).The deviations from the initial profile are approximately half that observed for the $L = 20$ km, but the fall in the bed (and the initial profile) over 5 km is only 8.73 m."

Now that we have examined the surface elevations in greater detail, it is clear that the differences in the final geometry of the ice surface between the two simulations are very slight and smoothly varying, so that we now consider that these geometric differences are not influential for the spatial differences in the velocity ratios discussed on P17L30 – and accordingly we have deleted the phrase "and a slightly different final geometry of the ice surface".

*Figures 8 and 9 should be merged and made similar to Fig. 10 as it seems that the 5+3 panels are identical to the 8 panels of Fig. 10. Also, you should avoid to repeat the caption of a figure when it is similar to a previous one. For example, caption of Fig. 7 should simply writes: "Same as Fig. 6 but for L=5 km".*

Figs. 7-9 and/or their captions have been modified accordingly. The new Fig. 8 is a merger of previous Figs. 8 and 9.

*For all figures of modified ISMIP-HOM results, the surface elevation should clearly be shown by increasing the maximal elevation on the vertical axis. For ISMIPDp, it seems from Fig. 10 that the surface is everywhere higher than the 2000m, which should not be possible if the mass conservation is correctly fulfilled (and assuming a null surface accumulation/ablation, which still as to be specified).*

We hope that the new axes and the addition of the surface elevation profiles has resolved these concerns.

The $y$-axis limits have been increased for each of Figs. 6-9.

For ISMIPDp Figs. 8 and 9 (previous Figs. 9 and 10), the surface is not everywhere higher than 2000m. Rather, it is only higher than 2000m in the first ∼1km of the domain (for $L = 20$km) and in the first ∼0.5km (for $L = 5$km) for each of the Glen and ESTAR flow relations (see Figs. 8e and 9e).

As you indicate, the real ingredients here are the ice thickness and mass conservation. The thicknesses are necessarily identical at each end of the domain (forcing the corresponding elevation difference), while the peak thickness occurs where the depth averaged $v_x$ is minimised, since at steady state the horizontal mass flux is constant across the domain. Thickness and the bedrock gradient combine to determine the location of the highest surface elevation.

**Other changes to the manuscript**:

- P23L9: added "vertical" to clarify that it is the transitions between simple shear dominated and *vertical* extension dominated deformation associated with the slippery region that we are interested in

- P23L13: $\lambda_S = 0$ (was previously a typographical error: $\lambda_S = 1$)

- other minor grammatical errors fixed. See the included manuscript diff.pdf file

**References**

[revised manuscript text omitted]